# A symmetry algebra in double-scaled SYK

**Henry W. Lin and Douglas Stanford**

Stanford Institute for Theoretical Physics, Stanford University, Stanford, CA 94305, USA

## Abstract

The double-scaled limit of the Sachdev-Ye-Kitaev (SYK) model takes the number of fermions and their interaction number to infinity in a coordinated way. In this limit, two entangled copies of the SYK model have a bulk description of sorts known as the "chord Hilbert space." We analyze a symmetry algebra acting on this Hilbert space, generated by the two Hamiltonians together with a two-sided operator known as the chord number. This algebra is a deformation of the JT gravitational algebra, and it contains a subalgebra that is a deformation of the $\mathfrak{sl}_2$ near-horizon symmetries. The subalgebra has finite-dimensional unitary representations corresponding to matter moving around in a discrete Einstein-Rosen bridge. In a semiclassical limit the discreteness disappears and the subalgebra simplifies to $\mathfrak{sl}_2$, but with a non-standard action on the boundary time coordinate. One can make the action of $\mathfrak{sl}_2$ algebra more standard at the cost of extending the boundary circle to include some "fake" portions. Such fake portions also accommodate certain subtle states that survive the semi-classical limit, despite oscillating on the scale of discreteness. We discuss applications of this algebra, including sub-maximal chaos, the traversable wormhole protocol, and a two-sided OPE.



## Glossary

$N$:       the number of SYK fermions (6).

$p$:       the number of fermions that $p$articipate in each SYK interaction (6) .

$\lambda$       $\equiv 2p^2/N$ is a useful parameter in the double-scaled limit (6).

$q$       $\equiv e^{-\lambda}$ gives the penalty factor for crossing Hamiltonian chords (8).

$\Delta$ :    characterizes matter operator insertions consisting of products of $p\Delta$ fermions (9).

$r$       $\equiv q^{\Delta}$ (9).

$n_L, n_R$:  the number of chords to the left and right of a matter chord, see below (10).

$\bar{n}$:       the total number of chords plus matter dimensions e.g. $\bar{n} = n_L + n_R + \Delta$ for single-particle states (16).

$[A, B]_q$ $\equiv AB - qBA$ (21).

$[n]$    $\equiv \frac{1-q^n}{1-q}$ is the $q$-deformed integer, see (14).

$(a; q)_n$ $\equiv (1-a)(1-aq)\cdots(1-aq^{n-1})$ is the $q$-Pochhammer symbol.

$[n]!$   $\equiv (q;q)_n/(1-q)^n$ is the $q$ factorial.

$\mathfrak{a}_L^{\dagger}$:    creates a Hamiltonian chord at the left side, $\mathfrak{a}_L^{\dagger}|n_L, n_R\rangle = |n_L + 1, n_R\rangle$, see (11).

$\alpha_L$:    left inverse of $\mathfrak{a}_L^{\dagger}$, it annihilates a Hamiltonian chord $\alpha_L|n_L, n_R\rangle = |n_L - 1, n_R\rangle$, see (12).

$\mathfrak{a}_L$:    the Hermitian conjugate of $\mathfrak{a}_L^{\dagger}$ with respect to the chord inner product (18).

$D(\cdot)$:  the coproduct, mapping an element of algebra $\mathcal{A}$ to an element of $\mathcal{A} \otimes \mathcal{A}$, see (32).

$\ell$       $\equiv \lambda\bar{n}$ see above (30).

$c$       $\equiv q^{\bar{n}/2} = e^{-\ell/2}$ (46).

$J_{AB}$:   elements of the chord algebra that commute with $\bar{n}$, see (45) .

$U(J)$:   the algebra generated by the $J_{AB}$.

$U_{\sqrt{q}}(\mathfrak{sl}_2)$:  a deformation of the universal enveloping algebra of $\mathfrak{sl}_2$ (with deformation parameter $\sqrt{q}$) It is related to the $U(J)$ algebra in various ways. See appendix I.

$B, E, P$:  linear combinations of the $J_{AB}$ operators, defined in (48).

$y$:       acting on single-particle states $|n_L, n_R\rangle$ the $y$ operator is defined as $(n_L - n_R)/2$.

$x$       $\equiv \lambda y$ see (121).

$v$:       parametrizes the inverse temperature $\beta$ by $\pi v/\beta = \cos\frac{\pi v}{2}$. When $\lambda \to 0$ we have $c = \cos\frac{\pi v}{2}$ and therefore $\sqrt{1-c^2} = \sin\frac{\pi v}{2}$. The Lyapunov exponent of large $p$ SYK is $2\pi v/\beta = 2c$. See (116).

$\hat{\Omega}$:    the Casimir of the chord algebra, see (50).

$|\Omega\rangle$:    the maximally entangled state with no open chords. .

B, E, P: rescaled versions of $B, E, P$ see (126) that satisfy an ordinary $\mathfrak{sl}_2$ algebra in the $\lambda \to 0$ limit.

$\theta$: an angular coordinate on the thermal circle (135).

$\phi$: an angular coordinate on the fake circle where the B, E, P operators are realized simply (137).

OTOC: an out-of-time-order or "crossed" correlator of the form $\mathrm{tr}\big(e^{-\tau_1 H}\mathsf{V}e^{-\tau_2 H}\mathsf{W}e^{-\tau_3 H}\mathsf{V}e^{-\tau_4 H}\mathsf{W}\big)$. We will have occasion to consider $\tau_i < 0$. We will continue to refer to these kinds of correlators as OTOCs, even if $\mathsf{V}, \mathsf{V}$ are closer in Euclidean time to each other than $\mathsf{V}, \mathsf{W}$.

TOC: a time-ordered or "uncrossed" correlator of the form $\mathrm{tr}\big(e^{-\tau_1 H}\mathsf{V}e^{-\tau_2 H}\mathsf{V}e^{-\tau_3 H}\mathsf{W}e^{-\tau_4 H}\mathsf{W}\big)$, for any values of $\tau_i$.

## 1 Introduction

Gravity has a lot to say about maximally chaotic systems [1,2]. However, sub-maximal chaos is more generic in interacting quantum systems. It is believed that the holographic explanation of sub-maximal chaos involves going beyond Einstein gravity. For a system like $\mathcal{N} = 4$ SYM in the 't Hooft limit, string theory [3] predicts a leading correction to the chaos exponent of order $\sim (\ell_{\text{string}}/\ell_{\text{AdS}})^2$. Checking this prediction with a direct boundary computation of the chaos exponent in $\mathcal{N} = 4$ seems futuristic,[1] but a simple model where the many-body Lyapunov exponent has already been computed directly from the boundary is the large $p$ SYK model. It is given by [7]:[2]

$$\text{many-body Lyapunov exponent} = \frac{2\pi v}{\beta}, \qquad \frac{\pi v}{\beta \mathcal{J}} = \cos\Big(\frac{\pi v}{2}\Big). \qquad (1)$$

At low temperatures $\beta \mathcal{J} \gg 1$, $v \to 1$ and the model is maximally chaotic. At high temperatures, $v$ is small, and the Lyapunov exponent goes to $2\mathcal{J}$. So it is natural to search for the bulk mechanism that explains the sub-maximal exponent in this model.

Guiding our search will be certain emergent symmetries of the model. At low temperatures, the SYK model possesses an emergent $\mathfrak{sl}_2$ symmetry[3] that explains the maximal Lyapunov exponent and is related to the near horizon symmetries of the black hole. To find the appropriate finite temperature generalization of these symmetries, we will take a scenic route by considering a more general limit, where $N \to \infty$, $p \to \infty$ holding fixed $\lambda \equiv 2p^2/N$ [11–14]. The advantage of this limit is that there is an emergent "chord Hilbert space" [14–17] that has a bulk flavor with an explicit description. This Hilbert space describes excitations of the thermofield double state of two copies of the double-scaled theory. We identify a symmetry algebra acting on this Hilbert space, generated by the two Hamiltonians, together with a "chord number" operator that generalizes the length between the boundaries in JT gravity.

This "chord algebra" contains an interesting subalgebra that commutes with the length and that generalizes the $\mathfrak{sl}_2$ in JT gravity. When one takes $\lambda \to 0$ to relax from the double-scaled theory to the ordinary large $p$ SYK model, this subalgebra reduces to $\mathfrak{sl}_2$. This was initially surprising to us, as we believed that an $\mathfrak{sl}_2$ algebra would imply a maximal chaos exponent.

---

[1]For the special case of $\mathcal{N} = 4$ SYM in Rindler space, the chaos exponent is determined by the ordinary Regge intercept, and the futuristic calculation has been done using integrability [4] and matched to string theory [5]. See Fig. 7 of [6] for a plot of the analog of (1) for this case, $v(\beta \mathcal{J}) + 1 \to j(\lambda)$.

[2]See also [8,9] for the full 4-pt functions.

[3]We emphasize that we are *not* referring to the $\mathfrak{sl}_2$ gauge symmetry of the Schwarzian theory. The $\mathfrak{sl}_2$ symmetries of [10] are gauge-invariant operators that act on the 2-sided Hilbert space of JT gravity + matter.

The heuristic argument is that the $\mathfrak{sl}_2$ algebra determines the rate of exponential growth of perturbations with respect to the "rotation" generator, which should be related to the time evolution generator by a factor of $2\pi/\beta$, where $\beta$ is the circumference of the thermal circle. The loophole in this argument is that the geometry where the $\mathfrak{sl}_2$ acts turns out to be partly "fake," in the sense that it includes a Euclidean timefold. The correct conversion factor is $2\pi/\beta_{\text{fake}}$, where $\beta_{\text{fake}} = \beta/v$ is the larger circumference of the fake thermal circle. So, even though the chaos is submaximal, it is determined by a symmetry algebra. The "scramblon" or "Pomeron" operator that grows exponentially is a particular $\mathfrak{sl}_2$ generator $\mathsf{P}^-$. In this way, the fake disk gives a geometrization of the Pomeron as in the case of maximal chaos.[4]

A somewhat separate motivation for understanding the symmetries in the double scaled limit is the following. It was argued in [17] that the chord number operator should be interpreted as the two-sided length of the wormhole. Since the chord number operator has discrete eigenvalues, this would imply that the bulk dual of double scaled SYK has a discretized geometry of sorts. Naively a model of quantum gravity with discrete geometry should fail to preserve local Poincaré invariance. In $3 + 1$ dimensions and higher, it has been argued that such a breaking would be catastrophic in the infrared [19]. In this setting, we will show that the breaking of Poincaré invariance is controlled by some deformation of the naive $\mathfrak{sl}_2$ symmetry. This deformation of $\mathfrak{sl}_2$ will have the unusual property that there are finite dimensional unitary representations of the algebra, which arise naturally in describing wormholes with a finite (and integer-valued) chord number.

## 1.1 Different regimes of SYK

This paper will make use of a variety of scaling limits of SYK; to avoid dizziness we will explain them all here. The SYK model has 3 dimensionless parameters. They are: $p$ be the number of fermions that participate in each SYK interaction, $N$ the number of fermions, and $\beta\mathcal{J}$ the dimensionless coupling. We will always consider the limit $N \to \infty$, $p \to \infty$. This leaves two dimensionless parameters, $\lambda = 2p^2/N$ and $\beta\mathcal{J}$, which we depict below:

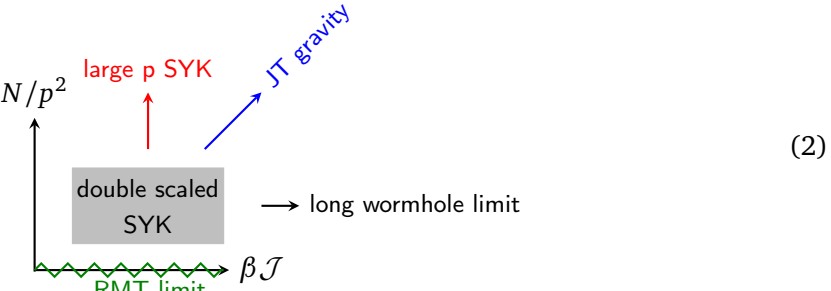

$$(2)$$

In the double scaling limit, $\lambda$ is held fixed, whereas in the conventional large $p$ limit, one takes $N \to \infty$ followed by $p \to \infty$, which is equivalent to first performing the double scaling limit and then taking $\lambda \to 0$. The quantum JT limit can be obtained by scaling $\beta\mathcal{J} \to \infty, N/p^2 \to \infty$, holding fixed $C \propto (\lambda\beta\mathcal{J})^{-1}$. This is sometimes referred to as the "triple scaling limit," and is perhaps the best understood regime of SYK. Note that the semi-classical JT gravity regime $C = \frac{N}{2p^2\beta\mathcal{J}} \gg 1$ smoothly matches on to the low temperature, large $p$ SYK limit. We can also study the limit $\beta\mathcal{J} \to \infty$ with $\lambda$ fixed. This is the $U_{\sqrt{q}}(\mathfrak{sl}_2)$ limit discussed in Appendix A.2. One can approach the JT gravity limit from below by taking this limit and sending $\lambda \to 0$. Finally, if we take $p^2 \gg N$, there are so many terms in the Hamiltonian that $H$ can be treated as a random matrix drawn from a Gaussian ensemble; the density of states in this limit is a semi-circle.

---

[4]Note that although we refer to (1) as submaximal chaos according to [2], a stronger bound has been proposed [18] which resembles the original chaos bound applied to the fake thermal circle. Large $p$ SYK saturates this one.

To use the language of von Neumann algebras (see [20] for a review), we expect that the algebra of 1-sided observables to be Type III$_1$ in the large $p$ limit [21, 22], Type II$_\infty$ in the triple scaling/JT gravity limit [23, 24], and Type II$_1$ in the finite $\lambda$, finite $\beta J$ limit [15, 17]. Of course, the SYK model at finite $N$ and $p$ has a type I algebra. So SYK is a neat example of a single system which realizes many different algebras in different limits.

## 1.2 Review of the JT gravitational algebra

The Hilbert space of JT gravity (+matter) on a disk is naturally 2-sided. Acting on this space are the operators $H_L, H_R$. Furthermore, a well-defined observable in semi-classical gravity is the 2-sided length $\ell$ or its renormalized version $\tilde{\ell}$. In a theory with matter, one can imagine extracting this from two-sided correlators $\mathcal{O}_L\mathcal{O}_R \sim e^{-\Delta\tilde{\ell}}$. We write the algebra in Appendix A.1, see (A.4). Although the algebra looks complicated, it can be presented as

$$\text{JT algebra} = \text{U}(\text{Heisenberg} \times \mathfrak{sl}_2). \tag{3}$$

Here $\text{U}(\cdot)$ means the universal enveloping algebra, e.g., the algebra formed by not just taking commutators but also products. The Heisenberg part of the algebra describes the "length mode" of the wormhole and its conjugate momentum, whereas the $\mathfrak{sl}_2$ describes the matter Hilbert space. The left and right Hamiltonians $H_L$ and $H_R$ can be expressed in terms of these variables [23], see (A.17) and (A.18). In other words, when we consider quantum fields on rigid AdS$_2$ the space of states organizes into $\mathfrak{sl}_2$ representations. Because the off-shell geometries in JT gravity (ignoring higher topologies) are always a cutout of the hyperbolic disk, and because the matter fields are not coupled to the Schwarzian mode (except via the gauge constraints), this $\mathfrak{sl}_2$ symmetry persists even when the Schwarzian mode is strongly coupled.

The $\mathfrak{sl}_2$ symmetry acts as the isometries of the Euclidean hyperbolic disk. Following the conventions of [10], we label them by how they act in the near horizon region, e.g., as boost, momentum, and global energy. The arrows show the direction that an insertion $O$ moves after conjugation $O \to e^{\tau G}Oe^{-\tau G}$:

$$B \;=\; \qquad -iP \;=\; \qquad E \;=\; \tag{4}$$

$$[B, P] = iE, \qquad [E, P] = iB, \qquad [B, E] = iP. \tag{5}$$

These generators move matter relative to the boundary, e.g., they do not change the length of the wormhole $\tilde{\ell}$ or its conjugate momentum $\tilde{k}$. The E generator can be viewed as variant of the coupled Hamiltonian in [25] that is singled out by the property that it commutes with both $\tilde{\ell}$ and $\tilde{k}$. In this way, the symmetries are related to the Gao-Jafferis-Wall/traversable wormhole protocol [26].

## 1.3 Summary

- In Section 2, we briefly review double-scaled SYK and introduce the chord algebra. We identify a subalgebra U($J$) that commutes with the total chord number. U($J$) plays the role of $\mathfrak{sl}_2$ in the JT limit. We also find a Casimir operator that commutes with the entire chord algebra.

- In Section 3, we discuss some representations of the chord algebra, including empty thermofield double states and also states with matter insertions.

- In Section 4, we take the $\lambda \to 0$ limit of the chord algebra. We find that U($J$) contracts to a finite temperature $\mathfrak{sl}_2$ algebra, which is different than the $\mathfrak{sl}_2$ algebra that is known to exist in the JT limit. We discuss some applications of this algebra.

- In the appendices, more details and $q$omputations are provided, including two other limits of the chord algebra (A) and a sketch of an independent derivation of the algebra in the semi-classical limit using the Liouville action (H). A hypothetical reader that is only interested in the strict large $p$ SYK model (no double scaling) could read Appendix H and then jump directly to Section 4.

## 2 The chord algebra

### 2.1 Review of the chord Hilbert space

We adopt the following conventions for the SYK model:

$$
\begin{aligned}
&\{\psi_i, \psi_j\} = 2\delta_{ij}\,, \\
&H = \mathrm{i}^{p/2} \sum_{1 \leq i_1 < \cdots < i_p \leq N} J_{i_1 \ldots i_q} \psi_{i_1} \cdots \psi_{i_p}\,, \quad \left\langle J_{i_1 \ldots i_p}^2 \right\rangle = \frac{\mathcal{J}^2}{\lambda \binom{N}{p}}\,, \\
&\lambda \equiv 2p^2/N\,, \quad q \equiv \exp(-\lambda)\,.
\end{aligned}
\tag{6}
$$

From now on, we will work in units where $\mathcal{J} = 1$.

The limit $N \to \infty$, $p \to \infty$, $\lambda =$ fixed is called the double-scaled limit. This is a solvable limit, in which the thermal partition function [11,12] and correlation functions [14] of the SYK model can be computed by summing chord diagrams. The chord diagrams represent the Gaussian Wick contractions from the disorder average of the $J_{i_1 \cdots i_p}$ and similar random couplings used to define operator insertions. As an example, consider the thermal 2-pt function of a "matter" operator

$$
\mathcal{O}_s = \mathrm{i}^{s/2} \sum_{1 \leq i_1 < \cdots < i_s \leq N} K_{i_1 \ldots i_s} \psi_{i_1} \cdots \psi_{i_s}\,, \qquad \left\langle K_{i_1 \ldots i_s}^2 \right\rangle = \frac{1}{\binom{N}{s}}\,.
\tag{7}
$$

A sample chord diagram that contributes is:

$$
\mathrm{tr}\, H^3 \mathcal{O}_s H^3 \mathcal{O}_s \quad \supset \quad \vcenter{\hbox{\includegraphics{diagram}}} \quad = \quad \frac{q^2 r^3}{\lambda^3}\,.
\tag{8}
$$

Here we are using the normalized trace (appropriate for Type II$_1$ algebras[5]), e.g., $\mathrm{tr}\,\mathbf{1} = 2^{-N}\, \mathrm{Tr}\,\mathbf{1}$. The rule for computing correlators in the double-scaled limit is to sum over chord diagrams with simple weighting factors: there is a factor of $\lambda^{-1}$ for each pair of Hamiltonian factors, a factor of $q$ for each intersection between the black Hamiltonian chords, and a factor of

$$
r = q^\Delta\,, \qquad \Delta = ps\,,
\tag{9}
$$

for each intersection between the matter chord and the Hamiltonian chords. By slicing open the chord diagram, one gets a two sided state [17]. In the above example, the (ket) state

---

[5]The trace can be written as $\mathrm{tr}(\cdot) = \langle\Omega| \cdot |\Omega\rangle$ where $|\Omega\rangle$ is the maximally entangled state. By acting with $H_L, H_R$ and matter operators on $|\Omega\rangle$ one generates a basis for the chord Hilbert space.

obtained by slicing at the gray points is

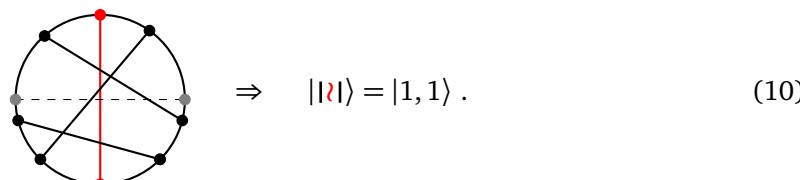

$$\Rightarrow \qquad |\text{〰}\rangle = |1,1\rangle \, . \tag{10}$$

We have indicated two different ways of labeling the chord state, first pictorially and second by writing the number of $H$ chords to the left and right of the matter chord, $|n_L, n_R\rangle$. There are multiple curves that connect the two gray boundary points. We chose the slice appropriate for defining ket vectors [17] – chords that cross the slice do not intersect in their past, see Appendix B.[6]

Acting on these chord states, we can define creation operators

$$\mathfrak{a}_L^\dagger |n_L, n_R\rangle = |n_L + 1, n_R\rangle, \qquad \mathfrak{a}_R^\dagger |n_L, n_R\rangle = |n_L, n_R + 1\rangle, \tag{11}$$

and annihilation operators

$$\alpha_L |n_L, n_R\rangle = |n_L - 1, n_R\rangle, \qquad \alpha_R |n_L, n_R\rangle = |n_L, n_R - 1\rangle \, . \tag{12}$$

We use different notation $\mathfrak{a}^\dagger$ and $\alpha$ because with respect to the chord inner product, these operators are not Hermitian conjugates. Acting on these chord states $|n_L, n_R\rangle$, the left and right Hamiltonians of the SYK model are represented by [17]:

$$\begin{aligned}
\lambda^{1/2} H_L &= \mathfrak{a}_L^\dagger + \alpha_L[n_L] + \alpha_R r q^{n_L}[n_R], \\
\lambda^{1/2} H_R &= \mathfrak{a}_R^\dagger + \alpha_R[n_R] + \alpha_L r q^{n_R}[n_L] \, .
\end{aligned} \tag{13}$$

In this expression, we defined the $q$-deformed integer

$$[n] \equiv 1 + q + q^2 + \cdots + q^n = (1 - q^n)/(1 - q) \, . \tag{14}$$

One can also consider a state with $m$ matter particles. We label such a state by $|n_0, n_1, \ldots, n_m\rangle$, where $n_0$ is the number of $H$ chords to the left of all matter chords, $n_1$ is the number between the first two, and so on. An intersection of the $i$-th matter chord with a Hamiltonian chord comes with a factor of $r_i = q^{\Delta_i}$. The left and right Hamiltonians act on such states by the representation [17]

$$\begin{aligned}
\lambda^{1/2} H_L &= \mathfrak{a}_0^\dagger + \sum_{i=0}^m \alpha_i[n_i] q^{n_i^<}, \qquad & \lambda^{1/2} H_R &= \mathfrak{a}_m^\dagger + \sum_{i=0}^m \alpha_i[n_i] q^{n_i^>}, \\
n_i^< &= \sum_{0 \le j < i} n_j + \Delta_{j+1}, & n_i^> &= \sum_{m \ge j > i} (\Delta_j + n_j) \, .
\end{aligned} \tag{15}$$

In section (2.3) we will write this formula in a more illuminating way.

In addition to the Hamiltonians, it is important to consider a 2-sided operator $\bar{n}$ which measures the total chord number weighted by the size of each chord:

$$\bar{n} = n_0 + \Delta_1 + n_1 + \cdots + \Delta_m + n_m \, . \tag{16}$$

In terms of the microscopic variables, we may realize $\bar{n}$ as the operator size [27]:

$$\bar{n} = \frac{\text{size}}{p} = \frac{1}{2p} \sum_{\alpha=1}^N \left( 1 + i \psi_\alpha^L \psi_\alpha^R \right) \, . \tag{17}$$

---

[6]This is a bit different from cutting the trace open and labeling states by $H, \mathcal{O}_s$ insertions in the ket or bra parts of the boundary. The chord Hilbert space allows some insertions to contract so that they do not appear in the state. This leads to a different basis for the same Hilbert space, with the advantage that the "chord number" is simple.

## 2.2 The chord algebra

The chord algbera is simply the algebra generated by $H_L, H_R$ and $\bar{n}$. As we see from (15), the left and right Hamiltonians are sums of terms that create and annihilate chords. It is useful to separate these terms, and write

$$\lambda^{1/2} H_{L/R} = \mathfrak{a}_{L/R}^{\dagger} + \mathfrak{a}_{L/R}, \tag{18}$$

where $\mathfrak{a}_{L/R}$ is the sum of all of the $\alpha$ terms appearing in (15), e.g.

$$\mathfrak{a}_L = \alpha_0 \frac{1-q^{n_0}}{1-q} + \alpha_1 q^{n_0+\Delta_1} \frac{1-q^{n_1}}{1-q} + \cdots + \alpha_m q^{\bar{n}-n_m} \frac{1-q^{n_m}}{1-q}. \tag{19}$$

This formula represents the fact that $\mathfrak{a}_L$ can annihilate any of the Hamiltonian chords, with a weighting factor given by the penalty of moving this chord all the way to the left. This characterization will help us show that $\mathfrak{a}_L$ is the adjoint of $\mathfrak{a}_L^{\dagger}$ with respect to the chord inner product [16]. This inner product is defined by summing over all of the ways of pairing up the chords in the ket with those in the bra, weighted by the penalty factor associated to whatever crossings take place, see Appendix B. To show that

$$\langle \mathfrak{a}_L \psi, \chi \rangle = \langle \psi, \mathfrak{a}_L^{\dagger} \chi \rangle, \tag{20}$$

with respect to this inner product, consider first the RHS. The $\mathfrak{a}_L^{\dagger}$ operator introduces a new Hamiltonian ket chord to the left of all ket chords. In the inner product computation, this new chord can pair up with any Hamiltonian chord from the bra, with a penalty factor given by the penalty associated to moving that bra chord all the way to the left (on its way to pair up with our new ket chord). So we can regard the $\mathfrak{a}_L^{\dagger}$ operator as summing over ways of removing a chord from the bra, with the penalty factor described above. On the LHS we don't have this extra ket chord, but instead the $\mathfrak{a}_L$ operator directly sums over ways of removing one of the bra chords, with the same penalty factor, leading to the same result as the RHS.

With the help of the *q*-deformed commutator,

$$[A, B]_q \equiv AB - qBA, \tag{21}$$

one can show that these operators together with $\bar{n}$ satisfy what we will call the *chord algebra*:

$$[\mathfrak{a}_L, \mathfrak{a}_R] = [\mathfrak{a}_L^{\dagger}, \mathfrak{a}_R^{\dagger}] = 0, \tag{22}$$

$$[\bar{n}, \mathfrak{a}_{L/R}^{\dagger}] = \mathfrak{a}_{L/R}^{\dagger}, \quad [\bar{n}, \mathfrak{a}_{L/R}] = -\mathfrak{a}_{L/R}, \tag{23}$$

$$[\mathfrak{a}_L, \mathfrak{a}_R^{\dagger}] = [\mathfrak{a}_R, \mathfrak{a}_L^{\dagger}] = q^{\bar{n}}, \tag{24}$$

$$[\mathfrak{a}_{L/R}, \mathfrak{a}_{L/R}^{\dagger}]_q = 1. \tag{25}$$

One can prove (22), (23), (24), (25) diagramatically. The first two relations (22), (23) are easy to verify. For (24), note that when we add a chord on the right $\mathfrak{a}_R^{\dagger}$ and then delete a chord $\mathfrak{a}_L$, we get a new possible Wick contraction that intersects all the $\bar{n}$ chords:

$$[\mathfrak{a}_L, \mathfrak{a}_R^{\dagger}] | HH \cdots H \rangle = \mathfrak{a}_L | \overbrace{HH \cdots H}^{\bar{n} \text{ chords}} \rangle \mathfrak{a}_R^{\dagger} = q^{\bar{n}} | HH \cdots H \rangle. \tag{26}$$

We also consider $\mathfrak{a}_L \mathfrak{a}_L^{\dagger}$ acting on an arbitrary chord state:

$$\mathfrak{a}_L \mathfrak{a}_L^{\dagger} | HH \cdots H \cdots H \rangle = q \, \mathfrak{a}_L^{\dagger} \mathfrak{a}_L | HH \cdots H \cdots H \rangle + \mathfrak{a}_L \mathfrak{a}_L^{\dagger} | HH \cdots H \rangle. \tag{27}$$

Focusing on the first term on the RHS of (27), we note that the indicated Wick contraction has one less crossing than the corresponding Wick contraction on the LHS. This explains the factor of $q$. The second term on the RHS of (27) is just the identity, which explains the RHS of (25). In (27) we have suppressed a sum on both the LHS and the RHS over which middle $H$ chord is annihilated. For notational simplicity we only wrote $H$'s in (26) and (27), but the proof goes through if we swap some of the middle $H$'s with matter chords.

Note that $H_L$ and $H_R$ look similar to the position operators of a pair of $q$-deformed Harmonic oscillators. However, due to (24), the two oscillators are not independent from each other.

It will prove useful to rewrite the chord algebra defined in (18)-(25) in terms of just $\mathfrak{a}^\dagger$ and $\mathsf{h}_{L/R} \equiv \lambda^{1/2} H_{L/R}$:

$$
\begin{aligned}
[\mathsf{h}_L, \mathsf{h}_R] &= 0\,, \\
[\mathsf{h}_{L/R}, \bar{n}] &= \mathsf{h}_{L/R} - 2\mathfrak{a}^\dagger_{L/R}\,, \\
[\mathfrak{a}^\dagger_L, \mathfrak{a}^\dagger_R] &= 0\,, \\
[\bar{n}, \mathfrak{a}^\dagger_{L/R}] &= \mathfrak{a}^\dagger_{L/R}\,, \\
[\mathsf{h}_{L/R}, \mathfrak{a}^\dagger_{L/R}]_q &= 1 + (1-q)\left(\mathfrak{a}^\dagger_{L/R}\right)^2\,, \\
[\mathsf{h}_{L/R}, \mathfrak{a}^\dagger_{R/L}] &= q^{\bar{n}}\,.
\end{aligned}
\tag{28}
$$

One benefit of writing the algebra this way is that we can immediately see how $\mathfrak{a}, \mathfrak{a}^\dagger$ can be written in terms of the microscopic fermionic operators. Using the second line of (28) together with (16),

$$
\begin{aligned}
\mathfrak{a}^\dagger_{L/R} &= \tfrac{1}{2}\left(\mathsf{h}_{L/R} + [\bar{n}, \mathsf{h}_{L/R}]\right)\,, \\
\mathfrak{a}_{L/R} &= \tfrac{1}{2}\left(\mathsf{h}_{L/R} - [\bar{n}, \mathsf{h}_{L/R}]\right)\,.
\end{aligned}
\tag{29}
$$

In particular, with respect to the microscopic inner product, $\mathfrak{a}^\dagger$ is indeed the Hermitian conjugate of $\mathfrak{a}$. Another immediate consequence of the chord algebra is the Liouville equation of motion. Setting $\ell = \lambda\bar{n}$, the second and last line of (28) gives

$$
\partial_L \partial_R \ell = [H_L, [H_R, \ell]] = -2e^{-\ell}\,.
\tag{30}
$$

This holds as an operator equation for any value $\lambda$. In Appendix H, we will discuss the Liouville approach to the large $p$ SYK model, where the same operator equation can be derived. As a sanity check let us note that in the 0-particle case $L = R$ and the last two relations of (28) imply $\lambda^{1/2}H = \mathfrak{a}^\dagger + \alpha \frac{1-q^n}{1-q}$.

## 2.3 The chord coproduct

The algebra defined by (22)-(25) is compatible with an additional structure known as a *coproduct*, which will be useful for generating representations. Let's review these concepts. First of all, an *algebra* $\mathcal{A}$ is a vector space, together with an associative bilinear map $m : \mathcal{A} \otimes \mathcal{A} \to \mathcal{A}$. In addition, there is a unit operator $\mathbf{1}$, which we can think of as a map from $\mathbb{C} \to \mathcal{A}$. An example of an algebra in quantum mechanics is the vector space of angular momentum operators together with the identity operator $\mathbf{1}$, e.g., $\mathfrak{su}(2) \oplus \mathbf{1}$. In this context, $m$ is just the Lie bracket.

A *coalgebra* also consists of a vector space but instead has a linear map $D : \mathcal{A} \to \mathcal{A} \otimes \mathcal{A}$ that is co-associative:

$$
(D \otimes \mathbf{1}) \cdot D = (\mathbf{1} \otimes D) \cdot D\,.
\tag{31}
$$

An example of a coalgebra in quantum mechanics is again provided by $\mathfrak{su}(2)\oplus\mathbf{1}$. $D$ is defined by the usual addition of angular momentum rules, e.g., $D(J_i) = J_i \otimes \mathbf{1} + \mathbf{1} \otimes J_i$ and $D(\mathbf{1}) = \mathbf{1} \otimes \mathbf{1}$.[7] Co-associativity (31) is the property that one can decompose a 3-particle state into irreducible representations by fusing $j_1, j_2$ into irreducible representations and then fusing the remaining particle $j_3$, or by fusing $j_2, j_3$, and then fusing with $j_1$.

If $\mathcal{A}$ is both an algebra and a coalgebra and the two are compatible, we say that it is a *bialgebra*. As the reader might have anticipated, $\mathfrak{su}(2)\oplus\mathbf{1}$ is in fact a bialgebra. The compatibility condition is $D([J_i, J_j]) = [D(J_i), D(J_j)]$. The chord algebra (18) is in fact a (non-counital) bialgebra, where the multiplication $m$ is just the operator product, and the coproduct is

$$D(\mathfrak{a}_L^\dagger) = \mathfrak{a}_L^\dagger \otimes 1, \quad D(\mathfrak{a}_R^\dagger) = 1 \otimes \mathfrak{a}_R^\dagger, \tag{32}$$

$$D(\mathfrak{a}_L) = \mathfrak{a}_L \otimes 1 + q^\Delta q^{\bar{n}} \otimes \mathfrak{a}_L, \tag{33}$$

$$D(\mathfrak{a}_R) = 1 \otimes \mathfrak{a}_R + q^\Delta \mathfrak{a}_R \otimes q^{\bar{n}}, \tag{34}$$

$$D(\bar{n}) = \bar{n} \otimes 1 + 1 \otimes \bar{n} + \Delta(1 \otimes 1). \tag{35}$$

One can check compatibility $D(ab) = D(a)D(b)$ by checking that the relations (22)-(25) are preserved by $D$. To interpret the coproduct, we introduce the operation $\delta$ of concatenating two 2-sided states, or joining two wormholes together, while adding a matter chord $\wr$ of dimension $\Delta$ between them, e.g.,

$$\delta : |\wr\wr\| \rangle \otimes |\wr\wr\rangle \to |\wr\wr\|\wr\wr\rangle. \tag{36}$$

Formally, let us define a map that acts on the tensor product of chord Hilbert spaces and concatenates states as in (36) $\delta : \mathcal{H} \otimes \mathcal{H} \to \mathcal{H}$. We can also define the inverse map $\delta^{-1} : \mathcal{H}|_\wr \to \mathcal{H} \otimes \mathcal{H}$. Here $\delta^{-1}$ is only defined on the subspace of $\mathcal{H}$ which contains exactly 1 matter chord of type $\wr$. Then acting on this subspace $\mathcal{H}|_\wr$

$$a = \delta \cdot D(a) \cdot \delta^{-1}, \qquad \forall\, a \in \text{chord algebra}. \tag{37}$$

The point of this equation is that the representations that appear on the RHS are simpler than the ones that appear on the LHS. For example, acting on a state with only one matter chord $\delta^{-1} |\|\|\wr\|\rangle = |\|\|\rangle \otimes |\|\rangle$. So with the help of the coproduct, we can figure out how $a$ acts on a 1-matter-chord state using just knowledge about how $a$ acts on 0-matter-chord states. For situations with more matter chords, we can define $\delta_{\wr m}^{-1}$ which is defined to act on the subspace of states with $\geq m$ matter chords of type $\wr$ and factorize them at position $m$. For example, $\delta_{\wr 2}^{-1} |\wr\|\wr\|\wr\rangle = |\wr\|\rangle \otimes |\|\wr\rangle$. With these definitions, co-associativity follows[8] from

$$(\delta_{\wr 1}^{-1} \otimes 1) \cdot \delta_{\wr 2}^{-1} = (1 \otimes \delta_{\wr 2}^{-1}) \cdot \delta_{\wr 1}^{-1}, \tag{39}$$

$$\delta \cdot (\delta \otimes 1) = \delta \cdot (1 \otimes \delta). \tag{40}$$

In Section 3 we will see that this simple property allows us to derive a crossing equation.

As an application of the coproduct, let's discuss a recursive way to build up multi-particle states. First we start with the 0 matter particle sector. These states are labeled by $|n\rangle$, where $n \in \mathbb{Z}_{\geq 0}$ is just the number of Hamiltonian chords. When we consider the 1-matter particle

---

[7]A standard coalgebra also has a special element $\epsilon$ called the *counit* which satisfies $(\epsilon \otimes \mathbf{1}) \cdot D = \mathbf{1} = (\mathbf{1} \otimes \epsilon) \cdot D$. For $\mathfrak{su}(2) \oplus \mathbf{1}$ this is $\epsilon(J_i) = 0$. However the chord algebra does not have a counit. So it is a "non-counital" bi-algebra.

[8]To see this, write $D(a) = \delta^{-1} a \delta$. Then (31) becomes

$$\delta_{\wr 1}^{-1} \delta_{\wr 2}^{-1} a \delta \cdot (\delta \otimes 1) = \delta_{\wr 2}^{-1} \delta_{\wr 1}^{-1} a \delta \cdot (1 \otimes \delta). \tag{38}$$

This is an expression that acts on 3 copies of the chord Hilbert space. Applying (39) and (40) immediately gives the desired result.

sector, we label states by $|n_0, n_1\rangle$ where $n_0$ is the number of chords to the left of the matter particle and $n_1$ is the number of particles to the right. We can think of this as a tensor product $|n_0\rangle \otimes |n_1\rangle$ of two 0-particle states. The oscillator algebra is realized as

$$
\begin{aligned}
\mathfrak{a}_L^\dagger &= \mathfrak{a}^\dagger \otimes 1\,, \quad \mathfrak{a}_R^\dagger = 1 \otimes \mathfrak{a}^\dagger\,, \\
\mathfrak{a}_L &= \mathfrak{a} \otimes 1 + r q^n \otimes \mathfrak{a}\,, \quad \mathfrak{a}_R = 1 \otimes \mathfrak{a} + r \mathfrak{a} \otimes q^n\,.
\end{aligned}
\tag{41}
$$

This procedure can be applied inductively to generate multi-particle states. In general, we imagine starting with $m$ matter particles, e.g., $|n_0\rangle \otimes \cdots \otimes |n_m\rangle$. Then we add another matter particle, say "to the right" of all the existing particles by taking a tensor product with another factor $|n_{m+1}\rangle$. This gives

$$
\begin{aligned}
\mathfrak{a}_{L,m+1}^\dagger &= \mathfrak{a}_{L,m}^\dagger \otimes 1\,, \quad \mathfrak{a}_R^\dagger = 1 \otimes \mathfrak{a}^\dagger\,, \\
\mathfrak{a}_L &= \mathfrak{a}_{L,m} \otimes 1 + r_{m+1} q^{\bar{n}_m} \otimes \mathfrak{a}\,, \quad \mathfrak{a}_R = 1 \otimes \mathfrak{a} + r_{m+1} \mathfrak{a}_{R,m} \otimes q^n\,, \\
\bar{n}_{m+1} &= \bar{n}_m \otimes 1 + 1 \otimes n + \Delta_{m+1}(1 \otimes 1)\,.
\end{aligned}
\tag{42}
$$

Using (42), one can show that (22)-(25) are satisfied for any $m$.

Microscopically, a 2-sided state $|O\rangle$ can be viewed as an operator $O$ acting on a 1-sided Hilbert space. Then given two states $O_1, O_2$ we can form a new state $O_3 = O_1 W O_2$. This defines a map from two wormhole states to a new wormhole state. However, this is not quite the map defined in the coproduct. In particular, (35) makes it clear that the map is defined in such a way that *bulk* lengths add, whereas the map $O_1, O_2 \to O_1 W O_2$ has the property that *boundary* lengths add. To fix this, we should subtract all Wick contractions going between $O_1$ and $O_2$. For example, if we take the state $O_1 = V H^2, O_2 = H$,

$$
\begin{aligned}
\delta(|VH^2\rangle \otimes |H\rangle) &= |VH^2WH\rangle - V\left(\overbrace{HHWH} + H\overbrace{HWH}\right)|0\rangle \\
&= |VH^2WH\rangle - 2q^\Delta |VHWH\rangle\,.
\end{aligned}
\tag{43}
$$

If we have sufficiently many chords to define a macroscopic geometry, $\delta$ preserves bulk lengths but not boundary lengths (e.g., the effective inverse temperature $\beta$ of the resulting state is *not* just $\beta_1 + \beta_2$.)

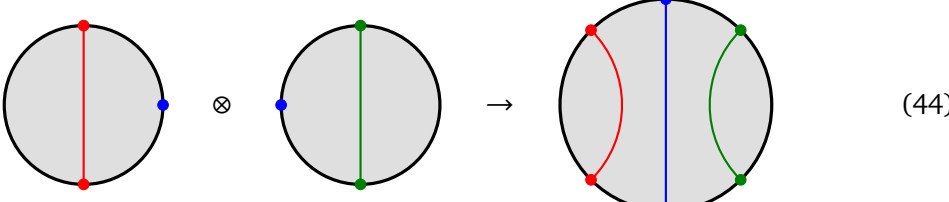

$$
\tag{44}
$$

Formally speaking the double scaled Hilbert space does not have a 1-sided Hilbert space, but we can nevertheless view the state $|O_3\rangle$ as obtained by a limit of this procedure. Thus even though the chord Hilbert space does not factorize into a tensor product of 1-sided states, there is still a factorization of sorts (obtained by reading (32)-(35) from right to left), but it is a factorization into two-sided states.

## 2.4 A subalgebra that commutes with $\bar{n}$

In JT gravity, the symmetry algebra of the bulk [10] may be identified by finding a subalgebra of the JT gravitational algebra that commutes with $\tilde{\ell}$ [28]. In the context of the chord algebra (28), one should look for elements that commute with $\bar{n}$, e.g. $\mathfrak{a}_i^\dagger \mathfrak{a}_j$ where $i, j \in \{L, R\}$. It is convenient to shift these generators so that they annihilate the thermofield double:

$$
J_{ij} = \mathfrak{a}_i^\dagger \mathfrak{a}_j - [\bar{n}]\,, \qquad i, j \in \{L, R\}\,.
\tag{45}
$$

It is convenient to define

$$c \doteq q^{\bar{n}/2} \,. \tag{46}$$

Then the chord algebra implies that the $J_{ij}$ operators satisfy the algebra

$$
\begin{aligned}
[J_{LL}, J_{RR}] &= \frac{c^2}{q}(J_{LR} - J_{RL}), \\
[J_{LR}, J_{RL}] &= J_{LL} - J_{RR}, \\
[J_{LL}, J_{LR}]_q &= c^2 J_{LR} - J_{LL}, \\
[J_{RR}, J_{RL}]_q &= c^2 J_{RL} - J_{RR}, \\
[J_{RL}, J_{LL}]_q &= c^2 J_{RL} - J_{LL}, \\
[J_{LR}, J_{RR}]_q &= c^2 J_{LR} - J_{RR}.
\end{aligned}
\tag{47}
$$

For section 4, it will be convenient to define three Hermitian combinations

$$E \doteq -\frac{1}{2c}(J_{LL} + J_{RR}), \quad B \doteq \frac{1}{2c}(J_{LL} - J_{RR}), \quad P \doteq \frac{i}{2}(J_{LR} - J_{RL}) \,. \tag{48}$$

These can also be written in terms of the Hamiltonians and the length operator as

$$
\begin{aligned}
B &= \frac{1}{4(1-q)c}[\mathsf{h}_L - \mathsf{h}_R, [\mathsf{h}_L + \mathsf{h}_R, \bar{n}]], \\
E &= -\frac{1}{4(1-q)c}[\mathsf{h}_L - \mathsf{h}_R, [\mathsf{h}_L - \mathsf{h}_R, \bar{n}]], \\
P &= iq[B, E] = \frac{i}{4}(\mathsf{h}_L[\mathsf{h}_R, \bar{n}] - \mathsf{h}_R[\mathsf{h}_L, \bar{n}]) \,.
\end{aligned}
\tag{49}
$$

## 2.5 Casimir

The algebra generated by (47) is denoted $U(J)$. In appendix C we show that one can form linear combinations of these generators so that the algebra is independent of $c$ and can be recognized as a subalgebra of $U_{\sqrt{q}}(\mathfrak{sl}_2)$. By starting with the Casimir operator $\Omega$ for $U_{\sqrt{q}}(\mathfrak{sl}_2)$ and guessing an improvement, we found that the following operator commutes with the entire chord algebra:

$$2\hat{\Omega} = q^{1-\bar{n}}\left\{1 - (1-q)\mathfrak{a}_L^\dagger \mathfrak{a}_L, 1 - (1-q)\mathfrak{a}_R^\dagger \mathfrak{a}_R\right\} + (1-q^2)\left(\mathfrak{a}_L^\dagger \mathfrak{a}_R + \mathfrak{a}_R^\dagger \mathfrak{a}_L\right) + 2q^{\bar{n}} \,. \tag{50}$$

This operator obviously commutes with $\bar{n}$. It also commutes with $\mathfrak{a}_L, \mathfrak{a}_R, \mathfrak{a}_L^\dagger, \mathfrak{a}_R^\dagger$ and therefore with the Hamiltonians $H_L, H_R$.

The Casimir may be viewed as a global symmetry of the boundary theory that emerges in the double scaling/large $p$ limit. It is unusual in that this Casimir operator is a nontrivial two-sided operator. In the case of a finite-dimensional Hilbert space, such a two-sided operator would imply the existence of ordinary one-sided symmetries. To see this, we can use the operator Schmidt decomposition to write a two-sided operator as $O = \sum_i c_i \ L_i \otimes R_i$ where $\text{tr}(L_i^\dagger L_j) = \text{tr}(R_i^\dagger R_j) = \delta_{ij}$. If $O$ is a symmetry then so are all of the $L_i$ operators, because $[H_L, L_i] = \frac{1}{c_i}\text{tr}_R(R_i^\dagger[H_L, O]) = 0$.

In Appendix C.2, we study the SYK model at finite $p$ and finite $N$ via numerical diagonalization. We study a particular microscopic realization of the operator $\hat{\Omega}$. For the finite and fairly small values of $N, p$ that we use, the would-be Casimir does not commute with the Hamiltonians but its expectation value in simple states is close to double-scaled predictions.

# 3 Representations of the chord algebra

We will discuss two types of irreducible representation of the chord algebra (22)-(25). The first is a "short" representation, obtained by starting with a state $|0\rangle$ that satisfies

$$\mathfrak{a}_L|0\rangle = \mathfrak{a}_R|0\rangle = \bar{n}|0\rangle = 0, \tag{51}$$

and then acting with raising operators

$$|n\rangle = (\mathfrak{a}_L^\dagger)^n|0\rangle = (\mathfrak{a}_R^\dagger)^n|n\rangle, \qquad n \geq 0. \tag{52}$$

The explicit formula for the representation is

$$\mathfrak{a}_L^\dagger|n\rangle = \mathfrak{a}_R^\dagger|n\rangle = |n+1\rangle, \tag{53}$$

$$\mathfrak{a}_L|n\rangle = \mathfrak{a}_R|n\rangle = \frac{1-q^n}{1-q}|n-1\rangle, \tag{54}$$

$$\bar{n}|n\rangle = n|n\rangle. \tag{55}$$

The inner product that makes $\mathfrak{a}$ the adjoint of $\mathfrak{a}^\dagger$ is [17]

$$\langle n'|n\rangle = \delta_{n,n'}[n]!. \tag{56}$$

Let's now discuss a generic lowest-weight irrep. This can be formed by starting with a state $|\Delta; 0, 0\rangle$ defined by the conditions

$$\mathfrak{a}_L|\Delta; 0, 0\rangle = \mathfrak{a}_R|\Delta; 0, 0\rangle = 0, \qquad \bar{n}|\Delta; 0, 0\rangle = \Delta, \tag{57}$$

and then using creation operators to make other (non-orthogonal!) states:

$$|\Delta; n_L, n_R\rangle = (\mathfrak{a}_L^\dagger)^{n_L}(\mathfrak{a}_R^\dagger)^{n_R}|\Delta; 0, 0\rangle. \tag{58}$$

The generators of the chord algebra act within this representation by

$$\mathfrak{a}_L^\dagger|\Delta; n_L, n_R\rangle = |\Delta; n_L+1, n_R\rangle, \tag{59}$$

$$\mathfrak{a}_R^\dagger|\Delta; n_L, n_R\rangle = |\Delta; n_L, n_R+1\rangle, \tag{60}$$

$$\mathfrak{a}_L|\Delta; n_L, n_R\rangle = \frac{1-q^{n_L}}{1-q}|\Delta; n_L-1, n_R\rangle + q^{\Delta+n_L}\frac{1-q^{n_R}}{1-q}|\Delta; n_L, n_R-1\rangle, \tag{61}$$

$$\mathfrak{a}_R|\Delta; n_L, n_R\rangle = \frac{1-q^{n_R}}{1-q}|\Delta; n_L, n_R-1\rangle + q^{\Delta+n_R}\frac{1-q^{n_L}}{1-q}|\Delta; n_L-1, n_R\rangle, \tag{62}$$

$$\bar{n}|\Delta; n_L, n_R\rangle = (\Delta + n_L + n_R)|\Delta; n_L, n_R\rangle. \tag{63}$$

To derive the formulas for $\mathfrak{a}_L$ and $\mathfrak{a}_R$, we used (24) and (25) to show that

$$\begin{aligned}
\mathfrak{a}_L(\mathfrak{a}_L^\dagger)^{n_L} &= \frac{1-q^{n_L}}{1-q}(\mathfrak{a}_L^\dagger)^{n_L-1} + q^{n_L}(\mathfrak{a}_L^\dagger)^{n_L}\mathfrak{a}_L, \\
\mathfrak{a}_L(\mathfrak{a}_R^\dagger)^{n_R} &= (\mathfrak{a}_R^\dagger)^{n_R-1}q^{\bar{n}}\frac{1-q^{n_R}}{1-q} + (\mathfrak{a}_R^\dagger)^{n_R}\mathfrak{a}_L.
\end{aligned} \tag{64}$$

The inner products of states within this representation are uniquely determined by an initial choice of normalization

$$\langle\Delta; 0, 0|\Delta; 0, 0\rangle = 1, \tag{65}$$

and by the requirement that $\mathfrak{a}^\dagger$ should be the adjoint of $\mathfrak{a}$. This is because we can write

$$\langle\Delta; n'_L, n'_R|\Delta; n_L, n_R\rangle = \langle\Delta|(\mathfrak{a}_R)^{n'_R}(\mathfrak{a}_L)^{n'_L}(\mathfrak{a}_L^\dagger)^{n_L}(\mathfrak{a}_R^\dagger)^{n_R}|\Delta\rangle, \tag{66}$$

and use (64) to move the $\mathfrak{a}$ operators to the right until they annihilate the state $|\Delta\rangle$. The resulting inner product is diagonal in $n_L + n_R$, but it connects states with different values of $n_L - n_R$. See appendix B for more on this.

For the generic representation, the value of the Casimir is

$$\hat{\Omega} = q^{\Delta} + q^{1-\Delta}. \tag{67}$$

In the limit $\Delta \to 0$ it reduces to the correct answer $1+q$ for the short representation. In fact, the entire short representation can be understood as a limit of the generic representation, because in the limit $\Delta \to 0$ the inner product has null states that impose the conditions $\mathfrak{a}_L = \mathfrak{a}_R$.

So far, we have discussed irreducible representations of the full chord algebra, consisting of a *primary* $|\Delta; 0, 0\rangle$ and *chord descendants* (58). One can also form irreps of the subalgebra U($J$) that commutes with $\bar{n}$, by restricting to a given value of $n_L + n_R$. Within this U($J$) irrep, we can define a U($J$) primary to be the eigenstate of $E$ with the smallest eigenvalue. Then the U($J$) descendants are all possible states that can be obtained by acting on such a primary with the U($J$) algebra. These U($J$) representations are finite dimensional, and the full representation (58) is a direct sum of infinitely many U($J$) representations, one for each possible value of $n_L + n_R$.

We will now discuss applications of these representations.

## 3.1 0-particle irrep

States with Hamiltonian chords but no matter chords correspond to the short representation. The shortening condition is $\mathfrak{a}_L = \mathfrak{a}_R$, which implies the Gauss law $H_L = H_R$ for an empty wormhole with no matter particles. Acting on this representation, the $J_{ij}$ generators vanish.

## 3.2 1-particle irreps

States with a single matter chord and any number of Hamiltonian chords form an irreducible representation with $\Delta$ determined by the penalty factor $r = q^{\Delta}$ for a crossing between a matter chord and a Hamiltonian chord. The state in the representation $|\Delta; n_L, n_R\rangle$ corresponds to the state $|n_L, n_R\rangle$ in the chord Hilbert space with $n_L$ Hamiltonian chords to the left of the matter particle and $n_R$ Hamiltonian chords to the right. Either from the coproduct formulas (41) or the representation (61), one finds

$$\mathfrak{a}_L = \alpha_L \frac{1-q^{n_L}}{1-q} + \alpha_R r q^{n_L} \frac{1-q^{n_R}}{1-q}, \tag{68}$$

and a similar equation with $L \leftrightarrow R$. Here $\alpha_L |n_L, n_R\rangle = |n_L - 1, n_R\rangle$ and $\alpha_R |n_L, n_R\rangle = |n_L, n_R - 1\rangle$.

The subspace with fixed $n = n_L + n_R$ gives a finite-dimensional irrep of U($J$), and substituting (68) into (45) gives explicit formulas for the $J_{ij}$ operators. The simplest case is $n = 0$, corresponding to a one-dimensional representation of U($J$). For this case, all of the $J_{ij}$ generators are equal to

$$J_{ij} = -\frac{1-r}{1-q}, \tag{69}$$

and one can check that this is consistent with (47) with $c^2 = r$. The simplest nontrivial representation is $n = 1$, corresponding to a two-dimensional representation:

$$\mathrm{R}^{\Delta}_{1\mathrm{p},1} = \mathrm{span}\{|\mathfrak{l}\rangle, |\mathfrak{l}\rangle\}. \tag{70}$$

Here $|\mathfrak{l}\rangle = |\Delta; 1, 0\rangle$ and $|\mathfrak{r}\rangle = |\Delta; 0, 1\rangle$. Acting on these non-orthogonal states, the generators are

$$
(1-q)J_{LL} = \begin{pmatrix} q(-1+r) & r(1-q) \\ 0 & -1+qr \end{pmatrix},
$$

$$
(1-q)J_{RR} = \begin{pmatrix} -1+qr & 0 \\ r(1-q) & q(-1+r) \end{pmatrix},
$$

$$
(1-q)J_{LR} = \begin{pmatrix} -1+r & -1 \\ 0 & -1+qr \end{pmatrix},
$$

$$
(1-q)J_{RL} = \begin{pmatrix} -1+qr & 0 \\ -1 & -1+r \end{pmatrix}.
$$

(71)

Later in this paper we will consider a kind of opposite limit where $n \to \infty$ (with $q^n$ is held fixed). In that limit, one linear combinations of the $J_{ij}$ operators will vanish, and the other three $(B, E, P)$ will act as a rescaled $\mathfrak{sl}_2$ algebra.

### 3.3 2-particle representations

#### 3.3.1 Decomposition into irreps

Let's now consider the case of two particles, with dimensions $\Delta_V, \Delta_W$. The generators do not change the ordering of the operators, and we will assume the V operator is to the left. The Hilbert space is spanned by states $|n_L, n_1, n_R\rangle$ with $n_L$ chords to the left of both operators, $n_1$ chords between them, and $n_R$ chords to the right of both. Acting on these states, the coproduct formula gives

$$
\mathfrak{a}_L = \alpha_L \frac{1-q^{n_L}}{1-q} + \alpha_1 r_1 q^{n_L} \frac{1-q^{n_1}}{1-q} + \alpha_R r_1 r_W q^{n_L+n_1} \frac{1-q^{n_R}}{1-q}.
$$

(72)

This is a reducible representation of the chord algebra, and it decomposes into a direct sum of irreducible representations, constructed from "double-trace" primaries with dimension $\Delta_V + \Delta_W + k$ that we will denote $[VW]_k$. To identify these primaries, we can search for states that satisfy the conditions $\mathfrak{a}_L = \mathfrak{a}_R = 0$ and then construct representations by acting with $\mathfrak{a}_L^\dagger$ and $\mathfrak{a}_R^\dagger$. The simplest case is

$$
|[VW]_0\rangle = |0, 0, 0\rangle
$$

(73)

$$
= |\mathfrak{l}\mathfrak{r}\rangle.
$$

(74)

The next simplest is a linear combination of states with one Hamiltonian chord:

$$
|[VW]_1\rangle = \gamma_{1,0,0}|\mathfrak{l}\mathfrak{l}\mathfrak{r}\rangle + \gamma_{0,1,0}|\mathfrak{l}\mathfrak{l}\mathfrak{r}\rangle + \gamma_{0,0,1}|\mathfrak{l}\mathfrak{r}\mathfrak{r}\rangle.
$$

(75)

The coefficients are determined by requiring $\mathfrak{a}_L = \mathfrak{a}_R = 0$ and requiring $\langle[VW]_1|[VW]_1\rangle = 1$:

$$
\begin{pmatrix} \gamma_{1,0,0} \\ \gamma_{0,1,0} \\ \gamma_{0,0,1} \end{pmatrix} = \frac{1}{\sqrt{\gamma_1}} \begin{pmatrix} -r_V(1-r_W^2) \\ 1-r_V^2 r_W^2 \\ -r_W(1-r_V^2) \end{pmatrix},
$$

(76)

$$
\gamma_1 = (1-r_V^2)(1-r_W^2)(1-r_V^2 r_W^2).
$$

(77)

We will also write one more explicit formula

$$
|[VW]_2\rangle = \gamma_{2,0,0}|\mathfrak{l}\mathfrak{l}\mathfrak{l}\mathfrak{r}\rangle + \gamma_{1,1,0}|\mathfrak{l}\mathfrak{l}\mathfrak{r}\rangle + \gamma_{0,2,0}|\mathfrak{l}\mathfrak{l}\mathfrak{r}\rangle + \gamma_{0,1,1}|\mathfrak{l}\mathfrak{r}\mathfrak{r}\rangle + \gamma_{0,0,2}|\mathfrak{l}\mathfrak{r}\mathfrak{r}\rangle + \gamma_{1,0,1}|\mathfrak{l}\mathfrak{l}\mathfrak{r}\mathfrak{r}\rangle, \quad (78)
$$

where

$$
\begin{pmatrix} \gamma_{2,0,0} \\ \gamma_{1,1,0} \\ \gamma_{1,0,1} \\ \gamma_{0,2,0} \\ \gamma_{0,1,1} \\ \gamma_{0,0,2} \end{pmatrix} = \frac{1}{\sqrt{\gamma_2}} \begin{pmatrix} -qr_V^2(1-r_W^2)(1-qr_W^2) \\ (1+q)r_V(1-qr_W^2)(1-qr_V^2 r_W^2) \\ -(1+q)r_V r_W(1-qr_V^2)(1-qr_W^2) \\ -(1-qr_V^2 r_W^2)(1-q^2 r_V^2 r_W^2) \\ (1+q)r_W(1-qr_V^2)(1-qr_V^2 r_W^2) \\ -qr_W^2(1-r_V^2)(1-qr_V^2) \end{pmatrix},
\tag{79}
$$

$$
\gamma_2 = (1+q)(1-r_V^2)(1-qr_V^2)(1-r_W^2)(1-qr_W^2)(1-qr_V^2 r_W^2)(1-q^2 r_V^2 r_W^2).
\tag{80}
$$

Let us mention that we may also view $[\mathsf{VW}]_1$ and $[\mathsf{VW}]_2$ as defining a "double trace" primary operator. In the $q \to 1$ limit, we checked that these $\gamma$ give rise to the usual $\mathfrak{sl}_2$ double trace primaries that one would expect in a generalized free field theory.[9] In Appendix F we derive the general decomposition of a 2-particle state into irreps, see (F.7):

$$
|n_L, n_1, n_R\rangle = \sum_{m_L + m_R + k = n_1} \psi_{k,m_L,m_R} |[\mathsf{VW}]_k; n_L + m_L, n_R + m_R\rangle.
\tag{82}
$$

Note that the Clebsch-Gordan-like coefficients $\psi$ do not depend on $n_L, n_R$, since one can act on both sides with $\mathfrak{a}_L^\dagger$ and $\mathfrak{a}_R^\dagger$ to increase $n_L, n_R$. This gives a decomposition

$$
\mathsf{R}_{2p}^{\Delta_V, \Delta_W} = \mathsf{R}_{1p}^{\Delta_V + \Delta_W} \oplus \mathsf{R}_{1p}^{\Delta_V + \Delta_W + 1} \oplus \mathsf{R}_{1p}^{\Delta_V + \Delta_W + 2} \oplus \dots
\tag{83}
$$

By restricting to the subspace with a fixed total number of Hamiltonian chords $n$, this decomposition gives a decomposition of $U(J)$ representations:

$$
\mathsf{R}_{2p,n}^{\Delta_V, \Delta_W} = \mathsf{R}_{1p,n}^{\Delta_V + \Delta_W} \oplus \mathsf{R}_{1p,n-1}^{\Delta_V + \Delta_W + 1} \oplus \dots \oplus \mathsf{R}_{1p,0}^{\Delta_V + \Delta_W + n}.
\tag{84}
$$

These are finite-dimensional representations and we can check that the dimensions match. The two-particle representation on the LHS has dimension equal to the number of ways one can decompose $n$ into the sum of 3 non-negative integers, corresponding to sprinkling the $H$ chords to the left, to the right or in between the two matter particles (this is known as a *weak composition*):

$$
\dim\left(\mathsf{R}_{2p,n}^{\Delta_V, \Delta_W}\right) = \binom{n+2}{n} = \sum_{k=0}^{n} (n+1-k).
\tag{85}
$$

The sum on the RHS corresponds precisely to the dimensions of the terms on the RHS of (84).

For a concrete example, we can consider the case with $n = 1$ one Hamiltonian chord

$$
\mathsf{R}_{2p,1}^{\Delta_V, \Delta_W} = \mathrm{span}\{|\mathsf{l}\mathsf{l}\rangle, |\mathsf{l}|\mathsf{l}\rangle, |\mathsf{l}\mathsf{l}|\rangle\}.
\tag{86}
$$

---

[9] To perform this check it was important to subtract off possible $H$ contractions in translating between a state and an operator, e.g.:

$$
|\mathsf{ll}\mathsf{l}\rangle = \lambda H^2 \mathsf{VW} - \mathsf{VW}, \qquad |\mathsf{l}|\mathsf{l}\rangle = \lambda H \mathsf{V} H \mathsf{W} - r_V \mathsf{VW}.
\tag{81}
$$

The $J_{ij}$ generators are

$$(1-q)J_{LL} = \begin{pmatrix} q(-1+r_V r_W) & r_V(1-q) & r_V r_W(1-q) \\ 0 & -1+qr_V r_W & 0 \\ 0 & 0 & -1+qr_V r_W \end{pmatrix},$$

$$(1-q)J_{RR} = \begin{pmatrix} -1+qr_V r_W & 0 & 0 \\ 0 & -1+qr_V r_W & 0 \\ r_V r_W(1-q) & r_W(1-q) & q(-1+r_V r_W) \end{pmatrix},$$

$$(1-q)J_{LR} = \begin{pmatrix} -1+r_V r_W & r_W(1-q) & 1-q \\ 0 & -1+qr_V r_W & 0 \\ 0 & 0 & -1+qr_V r_W \end{pmatrix},$$

$$(1-q)J_{RL} = \begin{pmatrix} -1+qr_V r_W & 0 & 0 \\ 0 & -1+qr_V r_W & 0 \\ 1-q & r_V(1-q) & -1+r_V r_W \end{pmatrix}. \tag{87}$$

After a change of basis so that the operators act on the states

$$\{|[\textcolor{red}{VW}]_0;1,0\rangle,|[\textcolor{red}{VW}]_0;0,1\rangle,|[\textcolor{red}{VW}]_1;0,0\rangle\},$$

this representation reduces to a block diagonal form:

$$(1-q)J_{LL} = \left(\begin{array}{cc|c} q(-1+r_V r_W) & r_V r_W(1-q) & 0 \\ 0 & -1+qr_V r_W & 0 \\ \hline 0 & 0 & -1+qr_V r_W \end{array}\right),$$

$$(1-q)J_{RR} = \left(\begin{array}{cc|c} -1+qr_V r_W & 0 & 0 \\ r_V r_W(1-q) & q(-1+r_V r_W) & 0 \\ \hline 0 & 0 & -1+qr_V r_W \end{array}\right),$$

$$(1-q)J_{LR} = \left(\begin{array}{cc|c} -1+r_V r_W & -1 & 0 \\ 0 & -1+qr_V r_W & 0 \\ \hline 0 & 0 & -1+qr_V r_W \end{array}\right),$$

$$(1-q)J_{RL} = \left(\begin{array}{cc|c} -1+qr_V r_W & 0 & 0 \\ -1 & -1+r_V r_W & 0 \\ \hline 0 & 0 & -1+qr_V r_W \end{array}\right). \tag{88}$$

In the one-dimensional block we have the representation (69) with $r_{\text{eff}} = r_V r_W q$, and in the two-dimensional block we have the two-dimensional representation (71) with $r_{\text{eff}} = r_V r_W$.

### 3.3.2 Ordering particles

We have just shown that by using the coproduct $D$ we can generate new "double trace" irreps $[VW]_k$. However, the coproduct is not commutative, so we actually get another double trace irrep $[WV_k$. These irreps are degenerate in the sense that they have the same $r_{\text{eff}} = q^{\Delta_V+\Delta_W+k}$, independent of the ordering of particles. A natural question is whether this degeneracy is a consequence of some sort of symmetry. This is indeed the case. Let us define a reflection operator R that acts on the chord Hilbert space by a reversing the order of all of the chords:

$$\mathsf{R}\,|n_0\,\mathcal{O}_1\,n_1\cdots\mathcal{O}_k\,n_k\rangle = |n_k\,\mathcal{O}_k\,n_{k-1},\cdots\mathcal{O}_1\,n_0\rangle, \tag{89}$$

$$\mathsf{R}^2 = 1. \tag{90}$$

Such an operator defines an automorphism of the chord algebra (22)-(25) that preserves the inner product:

$$R H_{L/R} R = H_{R/L}, \quad R \bar{n} R = \bar{n}, \tag{91}$$

$$R \mathfrak{a}_{L/R} R = \mathfrak{a}_{R/L}, \quad R \mathfrak{a}^\dagger_{L/R} R = \mathfrak{a}^\dagger_{R/L}, \tag{92}$$

$$R \hat{\Omega} R = \hat{\Omega}, \quad \langle v, R w \rangle = \langle R v, w \rangle. \tag{93}$$

Now since R commutes with $\hat{\Omega}$, we can simultaneously diagonalize both R and $\hat{\Omega}$. For the double traces, we can consider

$$|[\mathsf{V},\mathsf{W}]_k\rangle = \frac{|[\mathsf{VW}]_k\rangle - |[\mathsf{WV}]_k\rangle}{\mathcal{N}_k^-}, \qquad |\{\mathsf{V},\mathsf{W}\}_k\rangle = \frac{|\{\mathsf{VW}\}_k\rangle + |\{\mathsf{WV}\}_k\rangle}{\mathcal{N}_k^+}. \tag{94}$$

Such states have eigenvalues R = −1 for [·, ·] and R = +1 for {·, ·}. In Appendix F, we work out the normalization factors $\mathcal{N}_k^\pm$ so that these primaries are unit normalized.

## 3.4 Diagonalizing the chord irreps

In the above subsections, we focused on diagonalizing the algebra with respect to $\hat{\Omega}$ as well as $\bar{n}$. This had the advantage that there is a large $U(J)$ subalgebra that commutes with $\bar{n}$. However, another possibility is to diagonalize

$$\{H_L, \quad H_R, \quad \hat{\Omega}\}. \tag{95}$$

This is physically interesting because such a representation would be spanned by energy eigenstates; further, these operators form a generating set for a maximally commuting subalgebra of the chord algebra, analogous to the Cartan subalgebra of a Lie algebra.

This problem can be solved using results in [14]. First we will discuss the short representation. In this case, diagonalizing the $H_L = H_R$ corresponds to diagonalizing the transfer matrix of [14]. The eigenvectors, which we will denote $|s\rangle$, are linear combinations of $|n\rangle$ states

$$|s\rangle = \sum_{n=0}^{\infty} f_n(s)|n\rangle, \tag{96}$$

where $f_n$ must satisfy a recurrence relation that can be solved using the $q$-Hermite polynomial, as discussed in [14]. We choose to define $s$ as in [29] so that the eigenvalue of $H_L = H_R$ is

$$H_L|s\rangle = H_R|s\rangle = E(s)|s\rangle, \qquad E(s) \equiv -\frac{2\cos(\lambda s)}{\sqrt{(1-q)\lambda}}, \qquad 0 \le s \le \frac{\pi}{\lambda}. \tag{97}$$

We also normalize the states as in [29] so that

$$\langle s'|s\rangle = \frac{\delta(s'-s)}{\rho(s)}, \qquad \rho(s) \equiv \frac{1}{2\pi\Gamma_q(\pm 2is)}. \tag{98}$$

Then the matrix elements of $q^{\Delta'\bar{n}}$ are determined by [14] as

$$\langle s'|q^{\Delta'\bar{n}}|s\rangle = \quad \vcenter{\hbox{}} \quad = \frac{\lambda}{(1-q)^{1-2\Delta'}} \frac{\Gamma_q(\Delta' \pm is \pm is')}{\Gamma_q(2\Delta')}. \tag{99}$$

Next we discuss the the generic lowest-weight representation. In this case one would like to find a linear combination of states

$$|\Delta; s_L, s_R\rangle = \sum_{n_L, n_R} f_{n_L, n_R}(s_L, s_R)|\Delta; n_L, n_R\rangle, \tag{100}$$

such that

$$H_L|\Delta; s_L, s_R\rangle = E(s_L)|\Delta; s_L, s_R\rangle,$$
$$H_R|\Delta; s_L, s_R\rangle = E(s_R)|\Delta; s_L, s_R\rangle. \tag{101}$$

Without solving for $f$ explicitly, we can use results from [14] for the OTOC to determine the matrix elements of the operator $q^{\Delta'\bar{n}}$. If we normalize the states so that

$$\langle \Delta; s_L', s_R'|\Delta; s_L, s_R\rangle = \frac{\lambda}{(1-q)^{1-2\Delta}} \frac{\Gamma_q(\Delta \pm is_L \pm is_R)}{\Gamma_q(2\Delta)} \frac{\delta(s_L' - s_L)}{\rho(s_L)} \frac{\delta(s_R' - s_R)}{\rho(s_R)}. \tag{102}$$

then

$$\langle \Delta; s_L', s_R'|q^{\Delta'\bar{n}}|\Delta; s_L, s_R\rangle = \propto \sqrt{\frac{\Gamma_q(\Delta \pm is_L' \pm is_R')}{\Gamma_q(2\Delta)}} \sqrt{\frac{\Gamma_q(\Delta \pm is_L \pm is_R)}{\Gamma_q(2\Delta)}}$$
$$\times \sqrt{\frac{\Gamma_q(\Delta' \pm is_L' \pm is_L)}{\Gamma_q(2\Delta')}} \sqrt{\frac{\Gamma_q(\Delta' \pm is_R' \pm is_R)}{\Gamma_q(2\Delta')}} \left\{ \begin{matrix} \Delta' & s_L & s_L' \\ \Delta & s_R & s_R' \end{matrix} \right\}_q q^{\Delta\Delta'}. \tag{103}$$

Here the $\{\cdots\}_q$ is the 6j symbol of $U_{\sqrt{q}}(\mathfrak{sl}_2)$ [14, 29, 30].

Note that the chord algebra implies a variety of non-trivial identities involving the 6j symbol. Perhaps the simplest identity is just (30), which yields

$$(E(s_L') - E(s_L))(E(s_R') - E(s_R))\partial_\Delta' \langle \Delta; s_L', s_R'|q^{\Delta'\bar{n}}|\Delta; s_L, s_R\rangle\big|_{\Delta'=0} = 2\langle \Delta; s_L', s_R'|q^{\Delta'\bar{n}}|\Delta; s_L, s_R\rangle\big|_{\Delta'=1}. \tag{104}$$

We verified this identity numerically for $\Delta > 0$ in the triple scaling limit (using the $q = 1$ 6j symbol) and also at $q < 1$ in the short irrep.

Why does this 6j symbol appear in these formulas for representations of the chord algebra? In the JT gravity limit, one way to explain this is to back off the gauge-invariant formalism and use the boundary-particle formalism, where the Hilbert space of JT gravity consists of

$$(\mathcal{H}_L \otimes \mathcal{H}_M \otimes \mathcal{H}_R)/\mathfrak{sl}_2, \tag{105}$$

where $\mathcal{H}_L$ and $\mathcal{H}_R$ represent the left and right boundary particles, and $\mathcal{H}_M$ represents the bulk matter, see [10]. This system is acted on by a $\mathfrak{sl}_2$ gauge symmetry that corresponds to simultaneous translations or rotations of all three systems within the background hyperbolic space. In this formalism, the cubic vertex represents a state in (105) of the schematic form

$$\propto \sum_{m_L, m, m_R} \left\{ \begin{matrix} s_L & \Delta & s_R \\ m_L & m & m_R \end{matrix} \right\} |s_L, m_L\rangle \otimes |\Delta, m\rangle \otimes |s_R, m_R\rangle, \tag{106}$$

where to make an $\mathfrak{sl}_2$-invariant state, we have contracted with the "$3jm$ symbol" (essentially a Clebsch-Gordan coefficient) for the appropriate representations of $\mathfrak{sl}_2$. The proportionality constant can depend on $s_L, \Delta, s_R$ but not the $m$ variables. The four cubic vertices in (103) are contracted together in a pattern so that these $3jm$ symbols give a $6j$ symbol. So we see that the $6j$ symbol can be explained in a nice way using the $\mathfrak{sl}_2$ bulk gauge symmetry.[10] In the case of double-scaled SYK, perhaps the $q$-deformed $6j$ symbol could be nicely explained in some formulation of the system with $U_{\sqrt{q}}(\mathfrak{sl}_2)$ gauge symmetry, see [32,33].

### 3.5 Chord blocks

We have shown that a general 2-particle state in the Hilbert space may be decomposed into chord irreps. More generally, we expect that any state with an arbitrary number of particles can be decomposed into irreps. The general algorithm for performing such a decomposition is illustrated in the case with $m = 5$ matter chords (say all distinct for simplicity). We can view such a state as being obtained from applying $\delta$ on two $m = 2$ particle states as in (36). We then perform the decomposition of both $m = 2$ states to irreps. So we are left with a 3 particle state. Then we view the 3 particle state as resulting from applying $\delta$ on the short representation and a 2 particle state. We decompose the 2-particle state into irreps and we finally decompose the resulting 2 particle state into irreps. Of course, there may be choices in what ordering one performs the decomposition. The final answer will be the same, as guaranteed by co-associativity (31).

The structure of the chord Hilbert space is thus a direct sum over irreps of the chord algebra. Furthermore, a general correlation function may be computed by knowing the fusion coefficients between different chord primaries. In general, we can reduce an $m$-pt function to an $(m-1)$-pt function by inserting a resolution of the identity $1 = \sum_\Delta \Pi_\Delta$. We refer to matrix elements of $\Pi_\Delta$ as chord blocks. An explicit formula for the 4-pt blocks has been obtained in [30,34], see Appendix A of [30] for an explicit definition of the Askey-Wislon polynomials $P_n$:

$$\left\langle \Delta_V, \Delta_W, s_L, s_M', s_R \right| \Pi_{[VW]_n} \left| \Delta_V, \Delta_W, s_L, s_M, s_R \right\rangle = P_n^{\Delta_V, \Delta_W}(s_L, s_M', s_R|q) P_n^{\Delta_V, \Delta_W}(s_L, s_M, s_R|q),$$
(107)

$$\left\langle \Delta_W, \Delta_V, s_L, s_M', s_R \right| \Pi_{[VW]_n} \left| \Delta_V, \Delta_W, s_L, s_M, s_R \right\rangle = \tilde{\gamma}_n P_n^{\Delta_W, \Delta_V}(s_L, s_M', s_R|q) P_n^{\Delta_V, \Delta_W}(s_L, s_M, s_R|q),$$
(108)

where

$$\tilde{\gamma}_n = (-1)^n q^{n(\Delta_V + \Delta_W) + n(n-1)/2} q^{\Delta_V \Delta_W}.$$
(109)

Here we are simply explaining the Hilbert space interpretation of this formula. The factor $\tilde{\gamma}_n$ in (108) is explained in Appendix F (F.9) and comes from $\langle [VW]_n | [WV]_n \rangle$. By cutting the

---

[10]A somewhat different derivation using the the formulation of JT gravity as an $\mathfrak{sl}_2$ BF theory [31] shows that the $\Gamma$ function factors also have an $\mathfrak{sl}_2$ representation theory origin.

OTOC in two different ways[11] , one may decompose the 6j symbol into chord blocks [30]:

$$
\begin{array}{ccc}
\text{(diagram)} & = \sum_{n=0}^{\infty} \tilde{\gamma}_n \; \text{(diagram)} & = \sum_{n=0}^{\infty} \tilde{\gamma}_n \; \text{(diagram)}
\end{array}
\tag{111}
$$

This crossing symmetry of the chord blocks arises from the coassociativity of the coproduct (31). In particular, we can consider a slicing of the above diagrams in such a way that we view the correlator as an overlap between a 3-particle state and a 1-particle state $\langle VWVW \rangle = \langle V_1 | W_1 V_2 W_2 \rangle$. Then we can either decompose $V_2 W_2$ first or $W_1 V_2$ first.

Stated more formally, we can apply the map $\delta_{W,1}^{-1} |W_1 V_2 W_2\rangle = |\psi\rangle \otimes |V_2 W_2\rangle$ where $|\psi\rangle$ is a state with no matter particles (in the short irrep). Then we may decompose the state $|V_2 W_2\rangle = \sum_n \gamma_n |[VW]_n\rangle$ into 1-particle irreps.

$$
\begin{aligned}
|W_1 V_2 W_2\rangle &= \delta_W \cdot \delta_{W,1}^{-1} |W_1 V_2 W_2\rangle = \sum_n \gamma_n \delta(|\psi\rangle \otimes |[VW]_n\rangle) = \sum_n \gamma_n |W[VW]_n\rangle \\
&= \delta_W \cdot \delta_{W,2}^{-1} |W_1 V_2 W_2\rangle = \sum_n \gamma_n' \delta(|[WV]_n\rangle \otimes |\psi'\rangle) = \sum_n \gamma_n' |[WV]_n V\rangle \, . \quad (112)
\end{aligned}
$$

For the TOC, we also have an analogous decomposition:

$$
\sum_{n=0}^{\infty} \; \text{(diagram)} = \sum_{n=0}^{\infty} \; \text{(diagram)}
\tag{113}
$$

Here the "chord identity block" **1** is the sum over states that appear in the short irrep, e.g., states with no matter chords. These chord blocks are similar to the Virasoro block in 2D CFT in that they sum insertions of the "stress tensor" $H_L$ and $H_R$ (these are included in the "chord descendants"). Note also that by taking the triple scaling limit, we can have also explained the Hilbert space meaning of the "JT blocks" that appeared in [30]. This applies in any situation where JT gravity is relevant, beyond just the SYK model.

## 4 The semiclassical limit and the fake disk

In this section we will consider the limit $\lambda \to 0$ (or $q \to 1$) holding fixed the inverse temperature:

$$
\lambda \to 0 \, , \qquad \text{holding fixed} \qquad \beta \, .
\tag{114}
$$

---

[11]As a check, [30] Appendix C.1 considered the semiclassical limit of these JT blocks in the special case $\Delta_V = \Delta_W$. They obtained OPE coefficients (see their equations C.1, C.2, and C.25)

$$
c_n = \frac{\Gamma(2\Delta + n)^2 \Gamma(4\Delta + n - 1)}{\Gamma(n+1)\Gamma(4\Delta + 2n - 1)} \, .
\tag{110}
$$

This agrees precisely with our equation (F.14) setting $\Delta_W = \Delta_V = \Delta$.

This is a semiclassical limit of the model, in which fluctuations are small. For example, if we expand the thermofield double state in chord states with $n$ chords, then the mean value of $n$ diverges, but $\ell \equiv \lambda n$ remains finite and becomes sharply peaked, with fluctuations of order $\lambda^{1/2}$. So we can also think about the limit directly in terms of the chord Hilbert space as

$$\{\lambda \to 0, n \to \infty\}, \qquad \text{holding fixed} \qquad \ell = \lambda \bar{n}. \tag{115}$$

For some purposes (like computing $\langle H \rangle$) one would need to keep track of the small fluctuations in $\ell$ when comparing the limits (114) and (115). However, for studying the U($J$) algebra and some of its consequences, we will be able to ignore this.

We would like to determine the location of the peak in the distribution for $\ell$, as a function of $\beta$ (or equivalently, the operator size of the square root of the density matrix [27] see also [35]). One can use the fact that the limit $\lambda \to 0$ of double-scaled SYK is equivalent to the ordinary large $N$ limit of the SYK model, with $p$ taken large after $N$.[12] It is convenient to parametrize $\beta$ using a parameter $\nu$ between zero and one

$$\frac{\pi \nu}{\beta} = \cos \frac{\pi \nu}{2}. \tag{116}$$

Here $\nu = 0$ corresponds to high temperature (small $\beta$) and $\nu = 1$ corresponds to low temperature ($\beta = \infty$). This looks like a strange way to parametrize $\beta$, but it simplifies some formulas. In particular the two point function of SYK fermions is [7]

$$\langle \psi(\tau)\psi(0) \rangle = e^{g(\tau)/p}, \qquad g(\tau) = 2 \log \frac{\cos \frac{\pi \nu}{2}}{\cos[\frac{\pi \nu}{2}(1 - \frac{2\tau}{\beta})]}. \tag{117}$$

The fundamental fermion $\psi$ fields are matter operators with dimension $\Delta = 1/p$, and the two point function (117) determines the typical number of chords intersected by a matter chord passing between two boundary points $\langle \psi\psi \rangle = r^n = e^{-\Delta \ell} = e^{-\ell/p}$. This implies that $\ell = -g$. The parameter $\bar{n}$ in (115) should be compared to the number of chords intersected on the $t = 0$ slice, $-\lambda \bar{n} = g(\beta/2) = 2 \log \cos \frac{\pi \nu}{2}$, so

$$\ell \equiv \lambda \bar{n} = -2 \log \cos \frac{\pi \nu}{2}, \qquad \text{or equivalently} \qquad c \equiv e^{-\ell/2} = \cos \frac{\pi \nu}{2}. \tag{118}$$

This is the desired relationship between the length $\ell$ and the temperature (parametrized by $\nu$). As an application of this, one can use thermodynamic formulas for the large $N$ large $p$ SYK model to find

$$\langle H \rangle = -\frac{2}{\lambda} \sin \frac{\pi \nu}{2} = -\frac{2}{\lambda} \sqrt{1 - c^2}. \tag{119}$$

So, in particular, the low-energy low-temperature limit of SYK corresponds to $c \to 0$, and the infinite temperature limit where the energy vanishes corresponds to $c \to 1$.

Let's now consider a thermofield double state with an operator of dimension $\Delta$ inserted at some location in the Euclidean preparation. This state can be expanded in the chord Hilbert space in terms of the states $|\Delta; n_L, n_R\rangle$:

$$\vcenter{\hbox{$\beta_R/2$ \\ $\beta_L/2$}} = e^{-\frac{\beta_L}{2}H_L - \frac{\beta_R}{2}H_R}|\Delta; 0, 0\rangle = \sum_{n_L, n_R} \Psi(\beta_L, \beta_R \to n_L, n_R)|\Delta; n_L, n_R\rangle. \tag{120}$$

---

[12]A more general statement is that the expansion in powers of $\lambda$ coincides with the ordinary $1/N$ expansion of the SYK model, with $p$ taken large in each term. Concretely, the leading power of $p$ in each term of the $1/N$ expansion is such that it becomes an expansion in $p^2/N \propto \lambda$. One way to see this is to use the collective field description of the SYK model in the large $p$ limit (Appendix H).

The wave function $\Psi$ on the RHS can be obtained by writing $H_L$ and $H_R$ in terms of oscillators and then using the explicit representation (61) to act on the state. Each time we act with $H_L$ or $H_R$ we have some amplitude to change $n_L, n_R$ by one unit. So preparing $\Psi$ can be regarded as a biased random walk problem, and in the semiclassical limit the number of steps will be large, of order $1/\lambda$. This means that the variables $\lambda n_L$ and $\lambda n_R$ will be sharply peaked around mean values that depend on $\beta_L, \beta_R$, with fluctuations of order $\lambda^{1/2}$.

It is convenient to parametrize $n_L$ and $n_R$ by

$$\ell = \lambda(n_L + n_R + \Delta), \qquad x = \lambda \frac{n_L - n_R}{2}. \tag{121}$$

We will leave $\ell$ implicit and write a single-particle state as

$$|\Delta; n_L, n_R\rangle \to |x\rangle. \tag{122}$$

The distribution for the length $\ell$ will be sharply peaked around the value (118), and based on the above argument the distribution for $x$ will also be sharply peaked, around a value that we will determine later.

## 4.1 Semiclassical limit of the U($J$) algebra

The chord algebra simplifies in this limit, in particular the U($J$) algebra becomes a rescaled version of the ordinary $\mathfrak{sl}_2$ algebra. To derive this, it is important to keep in mind how different operators scale in the semiclassical limit. The number of chords, the Hamiltonian, and the oscillators are all large:

$$\bar{n} \sim \lambda^{-1}, \qquad H \sim \lambda^{-1}, \qquad \mathfrak{a} \sim \lambda^{-1/2}. \tag{123}$$

Here the notation $\mathcal{O} \sim \lambda^p$ means that acting on a normalized state, $\|\mathcal{O}|\psi\rangle\|$ scales as $\lambda^p$. We will be interested in states with only an $O(1)$ number of matter chords, each with $O(1)$ dimension $\Delta$. Acting on such states, combinations of operators that would vanish without matter remain small:

$$H_L - H_R \sim 1, \qquad J_{ij} \sim 1, \qquad (\mathfrak{a}_L - \mathfrak{a}_R) \sim \lambda^{1/2}. \tag{124}$$

The fact that the $J_{ij}$ operators are of order one allows us to simplify their algebra (47) by replacing $q$-commutators $[J, J]_q$ with ordinary commutators. The error in this approximation is schematically $(1-q)J^2 \sim \lambda$. One can also remove the one explicit factor of $1/q$ in the first line of (47). With these simplifications, the $J_{ij}$ algebra becomes an ordinary Lie algebra. It simplifies even further once we realize that there is a central element that can be approximated as zero:

$$J_{LR} + J_{RL} - J_{LL} - J_{RR} = (\mathfrak{a}_L - \mathfrak{a}_R)^\dagger (\mathfrak{a}_L - \mathfrak{a}_R)$$
$$\sim \lambda. \tag{125}$$

There are three remaining linear combinations of $J_{ij}$ operators, and they can be parametrized by the combinations $E, B, P$ in (48). In terms of these generators, the simplified $J_{ij}$ algebra becomes a rescaled $\mathfrak{sl}_2$ algebra. In particular, after defining the generators $\mathsf{B}, \mathsf{E}, \mathsf{P}$ by

$$B = \sqrt{1-c^2}\mathsf{B}, \qquad E = \mathsf{E}, \qquad P = \sqrt{1-c^2}\mathsf{P}, \tag{126}$$

the algebra is the standard $\mathfrak{sl}_2$ algebra

$$[\mathsf{B}, \mathsf{P}] = i\mathsf{E}, \qquad [\mathsf{E}, \mathsf{P}] = i\mathsf{B}, \qquad [\mathsf{B}, \mathsf{E}] = i\mathsf{P}. \tag{127}$$

We emphasize that this is true even for $c > 0$, i.e. nonzero temperature.

## 4.2 Semiclassical limit of single particle representations of the U($J$) algebra

We would like to work out how these generators act on single-particle states. To do so, we can start with the exact formulas

$$
\begin{aligned}
(1-q)J_{LL} &= c^2 - q^{n_L} + \mathfrak{a}_L^\dagger \alpha_R \left( rq^{n_L} - c^2 \right), \\
(1-q)J_{LR} &= rq^{n_R} - 1 + \mathfrak{a}_L^\dagger \alpha_R \left( 1 - q^{n_R} \right),
\end{aligned}
\tag{128}
$$

together with their images under $L \leftrightarrow R$. Let's consider these operators acting on a single particle state $|n_L, n_R\rangle \to |x\rangle$ in the semiclassical limit. The $\mathfrak{a}_L^\dagger \alpha_R$ operator shifts the value of $x$ by a small amount $\lambda$, and if this operator acts on superposition of states that is smooth in $x$, we expect to be able to approximate this using a formal derivative operator $\hat{\partial}_x$:

$$
\mathfrak{a}_L^\dagger \alpha_R |x\rangle = |x + \lambda\rangle \to |x\rangle + \lambda \hat{\partial}_x |x\rangle, \tag{129}
$$

$$
\mathfrak{a}_R^\dagger \alpha_L |x\rangle = |x - \lambda\rangle \to |x\rangle - \lambda \hat{\partial}_x |x\rangle. \tag{130}
$$

We also write

$$
q^{n_L} = ce^{(\lambda\Delta - x)/2}, \qquad q^{n_R} = ce^{(\lambda\Delta + x)/2}. \tag{131}
$$

Substituting these into (128) and expanding to leading order in $\lambda$, we find

$$
\begin{aligned}
P &= -ic\Delta \sinh x + i(1 - c\cosh x)\hat{\partial}_x, \\
B &= \Delta \sinh x - (c - \cosh x)\hat{\partial}_x, \\
E &= \Delta \cosh x + \sinh x\, \hat{\partial}_x.
\end{aligned}
\tag{132}
$$

Using $[\hat{\partial}_x, x] = 1$, one can check that these satisfy the algebra (127) after the rescaling (126).

The fact that $\mathfrak{a}$ and $\mathfrak{a}^\dagger$ are Hermitian conjugates with respect to the chord inner product implies that $E, B, P$ should be Hermitian. This gives three differential equations that the inner product must satisfy, for example for the $B$ generator the condition is

$$
\left( \Delta \cosh x' + \sinh x' \partial_x' \right) \langle x'|x\rangle = \left( \Delta \cosh x + \sinh x \partial_x \right) \langle x'|x\rangle. \tag{133}
$$

One can check that the following is a solution to all three equations (and in appendix B we verify that it is the correct solution)

$$
\langle x|x'\rangle = [n]! \left[ \frac{(1-c^2)/2}{\cosh \frac{x-x'}{2} - c\cosh \frac{x+x'}{2}} \right]^{2\Delta}. \tag{134}
$$

The overall normalization was obtained from considering the limit $x = x' = \log c^2$ which corresponds to the particle "all the way to the right". Then we may delete the matter chord and match to the 0-particle inner product to find $[n]!$. As a sanity check, we may also consider the case $x = -x' = \log c^2$. This gives $\langle x|x'\rangle = [n]!c^{2\Delta}$. The factor $c^{2\Delta} = e^{-\Delta\ell}$ is expected since the configuration is equivalent to inserting two operators on opposite sides of the wormhole.

To understand the $B, E, P$ generators better, we would like to convert (132) to a formula for matrix elements in states created by operator insertions at definite locations within the thermal cirlce, e.g.

$$
\langle \mathcal{O}(\theta_2) E\, \mathcal{O}(\theta_1)\rangle = \quad \tag{135}
$$

Here the coordinate $\theta = 2\pi\tau/\beta$ has been defined so that the thermal circle of size $\beta$ has a total angle $2\pi$. In the configuration sketched above, $\theta_1$ is negative and $\theta_2$ is positive. The $E$ operator insertion can be viewed as acting on the ket, moving the location of the insertion at $\theta_1$ slightly.

To write the answers for the matrix elements of $B, E, P$, it is useful to map the thermal circle to an extended "fake circle" as shown here

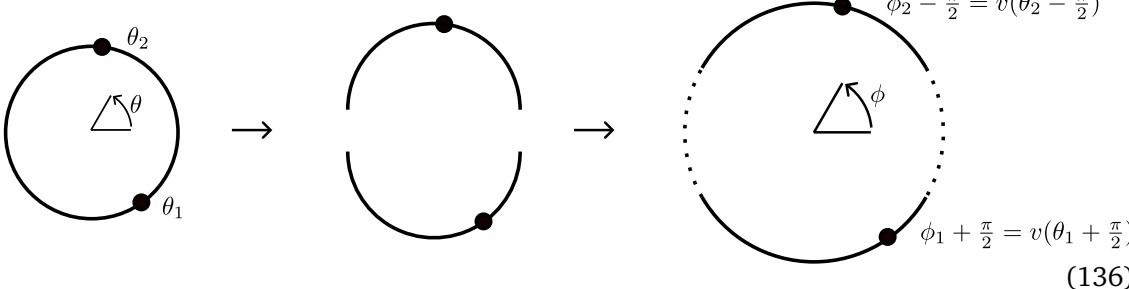

$$(136)$$

The fake circle is parametrized by a coordinate $\phi$, and there are two maps that connect the fake circle to the physical circle – one for operators in the bra and one for operators in the ket:

$$\text{ket:} \qquad \phi_1 + \tfrac{\pi}{2} = v(\theta_1 + \tfrac{\pi}{2}), \tag{137}$$

$$\text{bra:} \qquad \phi_2 - \tfrac{\pi}{2} = v(\theta_2 - \tfrac{\pi}{2}). \tag{138}$$

The images of the top and bottom halves of the physical circle under this map are shown as solid lines in the rightmost figure of (136). Note that because of the factor of $v$ these two solid portions do not cover the full fake circle. The "physical" regions are $\phi_2 \in \tfrac{\pi}{2}[1-v, 1+v], \phi_1 \in -\tfrac{\pi}{2}[1+v, 1-v]$.

In terms of this coordinate, it turns out that the rescaled $B, E, P$ generators simply act as the standard $\mathfrak{sl}_2$ generators on the fake circle:

$$\langle \mathcal{O}(\theta_2) B \mathcal{O}(\theta_1) \rangle = \partial_{\phi_1} \langle \mathcal{O}(\theta_2) \mathcal{O}(\theta_1) \rangle,$$
$$\langle \mathcal{O}(\theta_2) E \mathcal{O}(\theta_1) \rangle = (\cos(\phi_1)\partial_{\phi_1} - \Delta \sin \phi_1) \langle \mathcal{O}(\theta_2) \mathcal{O}(\theta_1) \rangle, \tag{139}$$
$$\langle \mathcal{O}(\theta_2) P \mathcal{O}(\theta_1) \rangle = i(\sin(\phi_1)\partial_{\phi_1} + \Delta \cos \phi_1) \langle \mathcal{O}(\theta_2) \mathcal{O}(\theta_1) \rangle.$$

To derive this, one uses (127) together with a relationship between the $|x\rangle$ states and states with a particle inserted on the boundary at location $\phi$. We will refer to such states as

$$|\phi\rangle = \mathcal{O}(\theta(\phi))|\text{TFD}\rangle. \tag{140}$$

As discussed above, the wave function for $|\phi\rangle$ is sharply peaked as a function of $\ell$ and $x$, and one can figure out the the location of the peak by matching the two point function (134) to the two point function of the large $p$ SYK model (117). These two inner products are

$$\langle x'|x\rangle = [n!] \left[ \frac{(1-c^2)/2}{\cosh \frac{x-x'}{2} - c \cosh \frac{x+x'}{2}} \right]^{2\Delta}, \qquad \langle \phi'|\phi\rangle = \left[ \frac{c}{\sin\left(\frac{\phi'-\phi}{2}\right)} \right]^{2\Delta}, \tag{141}$$

and their compatibility determines

$$|\phi\rangle \sim \frac{1}{\sqrt{[n!]}} \left( \frac{2c}{c + \sin(-\phi)} \right)^{\Delta} |x_\phi\rangle, \tag{142}$$

where

$$\cosh(x_\phi) = \frac{1 + c \sin(-\phi)}{c + \sin(-\phi)}. \tag{143}$$

One can then directly show (139) from (127).

## 4.3 Conformally covariant correlators

In this section we discuss a simple application of (139). The Hermiticity of the $E, B, P$ generators

$$\langle G\chi, \psi \rangle = \langle \chi, G\psi \rangle, \qquad G \in \{B, E, P\}, \tag{144}$$

implies $\mathfrak{sl}_2$ covariance of correlation functions at leading order for small $\lambda$. For example, in the context of the two point function $\langle \chi, \psi \rangle = \langle \mathcal{O}(\theta_2)\mathcal{O}(\theta_1) \rangle$, the RHS of (144) was written in (139), and the LHS is

$$\langle \mathcal{O}(\theta_2)B\mathcal{O}(\theta_1) \rangle = -\partial_{\phi_2}\langle \mathcal{O}(\theta_2)\mathcal{O}(\theta_1) \rangle,$$
$$\langle \mathcal{O}(\theta_2)E\mathcal{O}(\theta_1) \rangle = -(\cos(\phi_2)\partial_{\phi_2} - \Delta \sin \phi_2)\langle \mathcal{O}(\theta_2)\mathcal{O}(\theta_1) \rangle, \tag{145}$$
$$\langle \mathcal{O}(\theta_2)P\mathcal{O}(\theta_1) \rangle = -i(\sin(\phi_2)\partial_{\phi_2} + \Delta \cos \phi_2)\langle \mathcal{O}(\theta_2)\mathcal{O}(\theta_1) \rangle.$$

Equating (139) and (145) gives the usual constraints of $\mathfrak{sl}_2$-covariance of the two point function, with the unique solution

$$\langle \mathcal{O}(\theta_2)\mathcal{O}(\theta_1) \rangle = \frac{\text{const.}}{\sin^{2\Delta}(\frac{\phi_2 - \phi_1}{2})}. \tag{146}$$

This matches the two point function in the large $p$ SYK model [7] written in the $\phi$ coordinate. This may not be very impressive, since we used this form of the two point function in deriving the relationship between $|\phi\rangle$ and $|x\rangle$ that led to (139), but we can now go further and try to apply similar logic to the four point function.

Choosing $\langle \chi, \psi \rangle = \langle W_4 V_3 V_2 W_1 \rangle$ or $\langle \chi, \psi \rangle = \langle V_3 W_4 V_2 W_1 \rangle$ does not lead to much, because for small $\lambda$ these four point functions factorize into a product of two point functions. However, there are related quantities $\langle [W_4, V_3][W_2, V_1] \rangle$ and $\langle \{W_4, V_3\}[W_2, V_1] \rangle$ for which the factorized contribution vanishes.[13] The leading order answer then appears at order $\lambda$ and is a nontrivial function of the positions that can be computed using the Streicher formula [8, 9]. Since we are studying correlators at order $\lambda$, we need to be a bit careful with the constraint (144). We can write it as

$$\langle G_c\chi, \psi \rangle + \langle (G - G_c)\chi, \psi \rangle = \langle \chi, G_c\psi \rangle + \langle \chi, (G - G_c)\psi \rangle, \tag{147}$$

where $G_c$ is the leading small $\lambda$ expression, which is a sum of the operators in (139) or (145) acting on each of the operator insertions independently. $G - G_c$ is the difference between this expression and the exact generator, and it will involve an explicit factor of $\lambda$ in the small $\lambda$ limit. In the case where both $\chi$ and $\psi$ states are commutators, with norms of order $\lambda^{1/2}$, then the terms involving $G_c$ will be of order $\lambda$, but the terms involving $(G - G_c)$ will be smaller becuase

$$|\langle (G - G_c)\chi, \psi \rangle| \leq \|(G - G_c)\chi\| \|\psi\|, \tag{148}$$

and if both $\chi$ and $\psi$ are commutator states then $\|\psi\|$ is of order $\lambda^{1/2}$ and $\|(G - G_c)\chi\|$ will be at most of order $\lambda$ because of the explicit factor of $\lambda$ in $G - G_c$. So (147) implies conformal covariance of the $O(\lambda)$ term in $\langle [W_4, V_3][W_2, V_1] \rangle$. However, if $\chi$ is a commutator state and $\psi$ is an anticommutator (with norm of order one) we cannot obviously use (148) to show the LHS is smaller than order $\lambda$, and we cannot easily show conformal covariance of the leading order $\langle \{W_4, V_3\}[W_2, V_1] \rangle$.

---

[13]Here by the commutator or anticommutator, we mean to change the operator ordering of the operators but leave them at the same positions; most of the orderings will involve timefolds and for purely Euclidean positions these will be Euclidean timefolds.

These expectations are borne out by the Streicher formula for the leading connected contribution to the four point function in large $p$ SYK [8,9]. This formula leads to the conformally covariant double commutator

$$\frac{\langle [W_2, V_4][V_3, W_1] \rangle}{\langle W_2 W_1 \rangle \langle V_4 V_3 \rangle} = 2\lambda \Delta_W \Delta_V \frac{1+\chi}{1-\chi}, \qquad \chi = \frac{\sin \frac{\phi_{13}}{2} \sin \frac{\phi_{42}}{2}}{\sin \frac{\phi_{14}}{2} \sin \frac{\phi_{32}}{2}}, \tag{149}$$

and the non-conformally covariant commutator-anticommutator:

$$\frac{\langle \{W_2, V_4\}[V_3, W_1] \rangle}{\langle W_2 W_1 \rangle \langle V_4 V_3 \rangle} = 2\lambda \Delta_W \Delta_V \tan\left(\frac{\pi v}{2}\right) \frac{2\sin \frac{\phi_{13}}{2} \cos \frac{\phi_{24}}{2} - \phi_{13} \cos \frac{\phi_{12}}{2} \cos \frac{\phi_{34}}{2}}{\sin \frac{\phi_{12}}{2} \sin \frac{\phi_{34}}{2}}. \tag{150}$$

It is interesting that in addition to being $\mathfrak{sl}_2$ invariant, the RHS of the double commutator expression (149) is completely independent of temperature once we write it in terms of the $\phi$ coordinate. Presumably this is because the full chord algebra dictates the temperature dependence.

## 4.4 Exploring the fake region

In this section we discuss the interpretation of the "fake circle." On the one hand, this is just a mapping of coordinates that simplifies the action of the $B, E, P$ generators. But it has an interesting property that the image of the thermal circle covers only a subset of the $\phi$ coordinate (the solid lines in (136)). What is the interpretation of the extra "fake" regions (dotted lines)?

One can clarify this by using the $B, E, P$ generators to mover an operator insertion into the fake region. For example, in the $\lambda \to 0$ limit, the boost operator $B = \sin\left(\frac{\pi v}{2}\right) \mathsf{B}$ acts as

$$e^{-a\mathsf{B}}|\phi_1\rangle = |\phi_1 - a\rangle. \tag{151}$$

If $\phi_1$ starts out in the physical region, meaning the image of $-\pi < \theta_1 < 0$ under the map (137), then by acting with a sufficiently large negative $a$, we can leave the physical region:

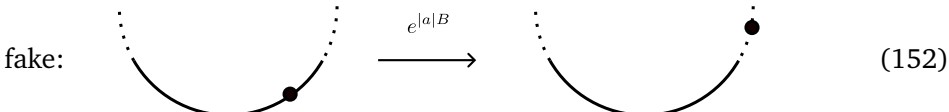

$$\text{fake:} \qquad\qquad \xrightarrow{e^{|a|B}} \qquad\qquad \tag{152}$$

In terms of the original thermal boundary, this corresponds to a state with a timefold of Euclidean time:

$$\text{physical:} \qquad\qquad \xrightarrow{e^{|a|B}} \qquad\qquad \tag{153}$$

Euclidean timefolds include factors of $e^{+\tau H}$, and for systems with unbounded Hamiltonians (like double-scaled SYK in the limit $\lambda \to 0$) the resulting state may not be normalizable. In the present case, the norm of the state is a two point function of $\mathcal{O}$ operators and in terms of the $\phi$ coordinate of the fake circle, the two operators will be located at $\pm\phi_1$. The formula (146) implies that this norm will remain finite as long as $\phi_1$ remains negative – even if $\theta_1$ is positive. In other words, the state remains normalizable as long as we don't have to introduce a Euclidean timefold to represent the correlator in the *fake* circle.

So the fake region represents the fact that the two point function on the physical circle does not diverge when operators approach each other. Operators can be smoothly continued past this point into a region described by timefolds of the physical circle or the dotted portion of the fake circle.

It is interesting to ask what happens to the wave function of the state, expressed in the chord basis, when we translate the matter particle into the fake region. To analyze this, it is convenient to act with the generator $B + E$, because it can be diagonalized explicitly. Then we consider a state where the operator insertion starts out at the extreme end of the physical region, to the right of all of the Hamiltonian chords. In this chord Hilbert space $|n_L, n_R\rangle$, this corresponds to the state $|n, 0\rangle$. We can act on this and produce a new state

$$e^{-a(B+E)}|n,0\rangle = \sum_{n_L=0}^{n} \psi_{n_L}(a)|n_L, n-n_L\rangle. \tag{154}$$

The explicit formula for $\psi$ is (see Appendix E)

$$\psi_{n_L}(a) = \sum_{m=n_L}^{n} \frac{(-1)^m q^{\frac{m(m-1)}{2}+n\Delta}(q^{1+m};q)_\infty}{(q^{1+n};q)_\infty (q;q)_{n-m}} \times e^{-a\frac{q^{n-m}-c^2}{(1-q)c}} \times (-1)^{n_L} q^{\frac{n_L(n_L-1)}{2}-n_L(m-1)-n_L\Delta} \binom{m}{n_L}_q. \tag{155}$$

For $a = 0$, this wave function is supported entirely on $n_L = n$. For $a > 0$ the operator translates the particle back into the physical region, and for this case we get a smooth wave function peaked at a location that depends on the value of $a$. For $a < 0$, the operator ought to move the particle into the fake region. In this case, what happens concretely is that the wave function becomes large and rapidly oscillating, with a sign that depends on the parity of $n_L$. Below we plot the answer for the case $n = 100$, $c = 3/10$, $\Delta = 1$, $a = \pm 2/10$:

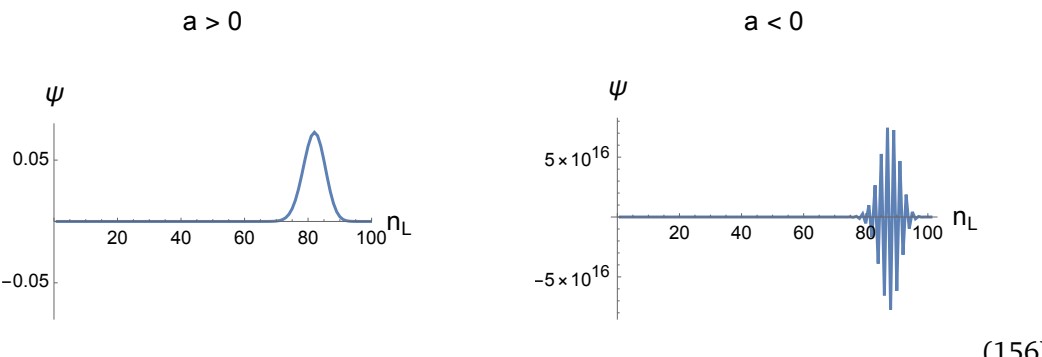

$$\tag{156}$$

The wave function in the $a < 0$ region is a bit impractical to work with, but it does reproduce the predictions of the fake circle. For example, one can use this oscillating wave function and the inner product $\langle n, 0|n_L, n-n_L\rangle$ to compute $\langle n, 0|e^{-a(E+B)}|n, 0\rangle$, and in Appendix E we show explicitly that

$$\lim_{\substack{n\to\infty \\ c \text{ fixed}}} \frac{\langle n, 0|e^{-a(E+B)}|n, 0\rangle}{\langle n, 0|n, 0\rangle} = \left(\frac{c^2 e^{-ca/2}}{1-(1-c^2)e^{-ca}}\right)^{2\Delta}. \tag{157}$$

The formula on the RHS is precisely the answer predicted by acting with the $\mathfrak{sl}_2$ generators on the fake circle (see Appendix D).

A mathematical interpretation of the fake region is that it represents a subtlety in the $\lambda \to 0$ limit of the microscopic chord Hilbert space: acting on smooth states with exponentiated approximate $\mathfrak{sl}_2$ generators, we can produce rapidly oscillating states that one might have been tempted to ignore in the $\lambda \to 0$ limit. The fake circle offers a smooth way to parametrize these states and to represent their inner products and $\mathfrak{sl}_2$ representation. As an analogy, we can consider the famously tricky problem of putting chiral fermions on a lattice. Naive attempts to make only right-moving fermions on the lattice will typically result in making left-moving fermions, where the left movers involve wavefunctions that are oscillating on the lattice scale. This is because the dispersion relation $E \sim \sin(\lambda p)$ looks linear near $p \sim 0$ but also near

$p \sim \pi/(2\lambda)$ The infrared limit of the system involves not just smooth wavefunctions, but also involves states that are oscillating on the lattice scale. The left-movers are analogous to the states in the fake region.

## 4.5 Lyapunov exponent and Pomeron/scramblon operator

In this section we use the $\mathfrak{sl}_2$ algebra (127) to derive the finite-temperature Lyapunov exponent of double-scaled SYK. Using $\lambda^{1/2} H_L = \mathfrak{a}_L + \mathfrak{a}_L^\dagger$ and $\lambda^{-1/2}[H_L, \ell] = \mathfrak{a}_L - \mathfrak{a}_L^\dagger$, one can write the following exact formula for the $B$ generator (48):

$$e^{-\ell/2} B = \lambda \frac{H_L^2 - H_R^2}{4(1+q)} - \frac{1}{\lambda} \frac{[H_L, \ell]^2 - [H_R, \ell]^2}{4(1+q)}. \tag{158}$$

We would like to write an approximate version of this equation for small perturbations to the thermofield double state. To do so, we write

$$H_{L/R} = -\frac{2\sqrt{1-c^2}}{\sqrt{(1-q)\lambda}} + \delta H_{L/R},$$
$$\ell(\tau_L, \tau_R) = 2\log\frac{\cos(c(\tau_L + \tau_R))}{c} + \delta\ell(\tau_L, \tau_R). \tag{159}$$

Here, the first terms on the RHS are the expectation values in the thermofield double state, evolved forwards by time $\tau_L$ on the left and $\tau_R$ on the right. In principle one can use this to approximate the $B$ operator acting at general times $\tau_L, \tau_R$, but the answer is simplest if we restrict to the $t = 0$ slice at $\tau_L = \tau_R = 0$, because the $[H_{L/R}, \ell]^2$ terms then do not contribute because $\partial_{L/R}\ell$ vanishes at zeroth order in the configuration $\tau_L = \tau_R = 0$. In terms of the rescaled generator $\sqrt{1-c^2}B = B$, one finds that to first order in $\delta H$ and $\delta\ell$

$$2c B = H_R - H_L. \tag{160}$$

We derived this as a formula for B, but it is actually useful in reverse, as a formula for $H_R - H_L$. This is because the $\mathfrak{sl}_2$ algebra for the B, E, P operators implies that adjoint action of the B operator has the following eigenvalues and eigenvectors

$$i[B, P^\pm] = \mp P^\pm, \qquad P^\pm = E \pm P. \tag{161}$$

So (160) implies that the operator $P^-$ grows exponentially under time evolution

$$e^{i(H_R - H_L)t} P^- e^{-i(H_R - H_L)t} = e^{2ct} P^-. \tag{162}$$

This suggests that the Lyapunov exponent is $\lambda_L = 2c = 2\pi v/\beta$, which is indeed correct [7].

The leading growing operator in the Regge limit is sometimes referred to as the "Pomeron" (or in the thermal case the "scramblon"), and the above computation identifies it in large $p$ SYK as the $P^-$ operator. (If we reversed the sign of $t$, the growing operator would be $P^+$.) The proof of the chaos bound implies that these $P^\pm$ operaturs must be positive operators that annihilate the thermofield double state. A special feature in the present context is that $P^\pm$ act as $\mathfrak{sl}_2$ symmetry generators on the fake circle. This is a familiar in the JT limit, but we have seen here that it is also true at finite temperature in large $p$ SYK.

## 4.6 Two-sided OPE and the Streicher formula

In this section we will use the $B, E, P$ generators (plus energy fluctuations) to reproduce a formula for the connected four-point function of fermions in the large $N$ large $p$ SYK model

[8,9]. To begin, we use $\lambda^{1/2}H_L = \mathfrak{a}_L + \mathfrak{a}_L^\dagger$ and $\lambda^{-1/2}[H_L, \ell] = \mathfrak{a}_L - \mathfrak{a}_L^\dagger$, to write the following exact formula for the $E$ generator (48):

$$e^{-\ell/2}E = -\lambda \frac{H_L^2 + H_R^2}{4(1+q)} + \frac{1}{\lambda}\frac{[H_L,\ell]^2 + [H_R,\ell]^2}{4(1+q)} + \frac{1 - e^{-\ell}}{1 - q}\,. \tag{163}$$

One can approximate this equation by substituting in (159) and expanding to first order. When acting on the $t = 0$ slice with $\tau_L = \tau_R = 0$, one gets the simple equation

$$E = \frac{\sqrt{1-c^2}}{2c}(\delta H_L + \delta H_R) + \frac{c}{\lambda}\delta\ell\,. \tag{164}$$

This is closely related to the Maldacena-Qi Hamiltonian [25]. We would like to derive a similar equation for general times $\tau_L, \tau_R$. To start, it is helpful to write the formula (163) more explicitly for general times:

$$e^{-\frac{1}{2}\ell(\tau_L,\tau_R)}E(\tau_L,\tau_R) = -\lambda\frac{H_L^2 + H_R^2}{4(1+q)} + \frac{1}{\lambda}\frac{(\partial_L\ell(\tau_L,\tau_R))^2 + (\partial_R\ell(\tau_L,\tau_R))^2}{4(1+q)} + \frac{1 - e^{-\ell(\tau_L,\tau_R)}}{1 - q}\,. \tag{165}$$

Here $\tau_L$ and $\tau_R$ are measured from the horizontal $t = 0$ slice that crosses the thermal circle. It will be convenient to change coordinates to

$$\tau_+ = \tau_R + \tau_L\,, \qquad \tau_- = \tau_R - \tau_L\,. \tag{166}$$

In particular, $\tau_+$ is conjugate to $(H_R + H_L)/2$ and $\tau_-$ is conjugate to $(H_R - H_L)/2$. Let's start by working out the time dependence of the $E(\tau_L, \tau_R)$ operator. Eq. (164) implies that $E = E(0,0)$ commutes with $H_L + H_R$, so $E(\tau_L, \tau_R)$ is independent of $\tau_+$. Its remaining dependence on $\tau_-$ can be worked out using (160):

$$E(\tau_L,\tau_R) \approx e^{\frac{\tau_-}{2}(H_R-H_L)}Ee^{-\frac{\tau_-}{2}(H_R-H_L)} \tag{167}$$

$$\approx e^{c\tau_-\mathsf{B}}\frac{\mathsf{P}^+ + \mathsf{P}^-}{2}e^{c\tau_-\mathsf{B}} \tag{168}$$

$$\approx \frac{\mathsf{P}^+e^{ic\tau_-} + \mathsf{P}^-e^{-ic\tau_-}}{2}\,. \tag{169}$$

Here and elsewhere, operators $E, \mathsf{P}^\pm, \mathsf{B}$ without time arguments are assumed to be at $\tau_L = \tau_R = 0$.

Substituting this together with the first-order expansions (159) into the equation for $E(\tau_L, \tau_R)$ (165) gives a differential equation for $\delta\ell$:

$$\frac{1}{\lambda}\left[\frac{c}{\cos^2(c\tau_+)} - \tan(c\tau_+)\partial_{\tau_+}\right]\delta\ell(\tau_L,\tau_R) = \frac{\mathsf{P}^+e^{ic\tau_-} + \mathsf{P}^-e^{-ic\tau_-}}{2\cos(c\tau_+)} - \frac{\sqrt{1-c^2}}{2c}(\delta H_L + \delta H_R)\,. \tag{170}$$

The solution to this equation is

$$\delta\ell(\tau_L,\tau_R) = \lambda\left[\frac{\mathsf{P}^+e^{ic\tau_-} + \mathsf{P}^-e^{-ic\tau_-}}{2c\cos(c\tau_+)} - \frac{\sqrt{1-c^2}}{2c^2}(1 + c\tau_+\tan(c\tau_+))(\delta H_L + \delta H_R)\right]$$
$$+ \frac{\tan(c\tau_+)}{c}\partial_+\delta\ell(-\tfrac{\tau_-}{2}, \tfrac{\tau_-}{2})\,. \tag{171}$$

The final term can be evaluated as follows:

$$\partial_+\delta\ell(-\tfrac{\tau_-}{2}, \tfrac{\tau_-}{2}) = \tfrac{1}{2}[H_L + H_R, e^{\frac{\tau_-}{2}(H_R-H_L)}\delta\ell(0,0)e^{-\frac{\tau_-}{2}(H_R-H_L)}] \tag{172}$$

$$= \partial_{\tau_+}\delta\ell(0,0) + c\lambda\tau_-\mathsf{B}\,. \tag{173}$$

In the final step we used (49). Higher powers of $\tau_-$ will multiply terms proportional to $[H_L + H_R, E]$ which vanishes due to (164).

So we have the following formula for the fluctuation in the $\ell$ operator as a function of time:

$$
\delta\ell(\tau_L, \tau_R) = \lambda\left\{ \frac{\mathsf{P}^+ e^{ic\tau_-} + \mathsf{P}^- e^{-ic\tau_-}}{2c\cos(c\tau_+)} + \sqrt{1-c^2}\tan(c\tau_+)\tau_-\mathsf{B} \right.
$$
$$
\left. - \frac{\sqrt{1-c^2}}{2c^2}(1 + c\tau_+\tan(c\tau_+))(\delta H_L + \delta H_R) \right\} + \partial_+\delta\ell(0,0)\frac{\tan(c\tau_+)}{c}.
\tag{174}
$$

This can be regarded as a type of OPE that expresses two-sided correlators (which are functions of $\ell$) in terms of operators with simple time dependence under boost evolution by $H_R - H_L$. Note that this is different from a more conventional one-sided OPE. In some respects this is similar to the light-ray OPE in conformal field theory [36, 37].

In particular, this OPE allows us to compute out of time order correlators. (In Appendix G, we will use similar ideas to compute the $O(\lambda)$ corrections to the time-ordered correlators.) We will illustrate this by computing the OTOC in the following configuration:

$$
\begin{aligned}
&= \langle \mathsf{W}(\tfrac{\beta}{2} - \tau_L)\mathsf{V}(\tfrac{\beta}{4} - \tfrac{\tau'_+}{2})\mathsf{W}(\tau_R)\mathsf{V}(-\tfrac{\beta}{4} + \tfrac{\tau'_+}{2})\rangle \\
&= \langle \mathsf{W}_L \mathsf{V}_T \mathsf{W}_R \mathsf{V}_B\rangle \\
&= \langle \mathsf{V}_T|e^{-\Delta_W\ell(\tau_L, \tau_R)}|\mathsf{V}_B\rangle.
\end{aligned}
\tag{175}
$$

The $O(\lambda)$ part of the connected correlator is

$$
\frac{\langle \mathsf{V}_T|e^{-\Delta_W\ell(\tau_L, \tau_R)}|\mathsf{V}_B\rangle_c}{\langle \mathsf{V}_T\mathsf{V}_B\rangle\langle \mathsf{W}_L\mathsf{W}_R\rangle} = -\Delta_W\frac{\langle \mathsf{V}_T|\delta\ell(\tau_L, \tau_R)|\mathsf{V}_B\rangle_c}{\langle \mathsf{V}_T\mathsf{V}_B\rangle}.
\tag{176}
$$

This can be evaluated using (174). The terms on the first line reduce to expectation values of the $\mathsf{E}, \mathsf{B}, \mathsf{P}$ generators in a single particle state (139):

$$
\frac{\langle \mathsf{V}_T|\mathsf{P}^\pm|\mathsf{V}_B\rangle}{\langle \mathsf{V}_T\mathsf{V}_B\rangle} = \frac{\Delta_V}{\cos(c\tau'_+)}, \qquad \frac{\langle \mathsf{V}_T|\mathsf{B}|\mathsf{V}_B\rangle}{\langle \mathsf{V}_T\mathsf{V}_B\rangle} = \Delta_V\tan(c\tau'_+).
\tag{177}
$$

The final term involving $\langle\partial_+\delta\ell(0,0)\rangle$ vanishes because this operator is odd under a reflection of the vertical direction on the page, while the configuration of $V$ operators is even under this reflection.

It remains to calculate a term involving $\langle V_T|(\delta H_L + \delta H_R)|V_B\rangle$. To do this, we can use (174) a second time, this time applied to the $V_T, V_B$ operators. The terms involving the $\mathsf{P}^\pm$ and $\mathsf{B}$ generators vanish because the the only other operator insertion is the Hamiltonian, and the generators vanish on states with no matter insertions. The term involving $\partial_+\delta\ell(0,0)$ will contribute in an equal an opposite way to the $\delta H_L$ and $\delta H_R$ terms, so it will also vanish. All that remains are the terms involving $\delta H$, which combine to

$$
\frac{\langle V_T|(\delta H_L + \delta H_R)|V_B\rangle_c}{\langle V_T V_B\rangle} = 2\lambda\Delta_V\frac{\sqrt{1-c^2}}{c^2}(1 + c\tau'_+\tan(c\tau'_+))\langle(\delta H)^2\rangle.
\tag{178}
$$

To evaluate this energy fluctuation, we can use the thermodynamics of the large $p$ SYK model. As a function of the energy $H$, the action for the thermal partition function is

$$I(H) = \beta H + S(H) \tag{179}$$

$$= \beta H + \frac{2}{\lambda}\arcsin^2\left(\frac{\lambda H}{2}\right), \tag{180}$$

where we used (H.19). We choose a value of $\beta$ so that the saddle point value of $H$ is equal to $-\frac{2}{\lambda}\sqrt{1-c^2}$. Then one can check that the action to quadratic order is

$$I = \frac{2\arccos(c)}{\lambda c}\left[-2\sqrt{1-c^2} + c\arccos(c)\right] + \frac{\lambda}{2c^3}\left[c + \sqrt{1-c^2}\arccos(c)\right](\delta H)^2 + \dots \tag{181}$$

This implies that the variance of the energy fluctuations is

$$\langle(\delta H)^2\rangle = \frac{c^3}{\lambda(c + \sqrt{1-c^2}\arccos(c))}. \tag{182}$$

Putting the pieces together, we find

$$\frac{\langle W_L V_T W_R V_B\rangle_c}{\langle V_T V_B\rangle\langle W_L W_R\rangle} = \lambda\Delta_W\Delta_V\left[-\frac{\cos(c\tau_-)}{c\cos(c\tau_+)\cos(c\tau'_+)} - \sqrt{1-c^2}\tan(c\tau_+)\tan(c\tau'_+)\tau_- \right.$$
$$\left. + \frac{1-c^2}{c^2 + c\sqrt{1-c^2}\arccos(c)}(1 + c\tau_+\tan(c\tau_+))(1 + c\tau'_+\tan(c\tau'_+))\right]. \tag{183}$$

This agrees with [8, 9] once we use $c = \cos\frac{\pi\nu}{2}$. In Appendix G we use a similar method to evaluate the TOC, also finding agreement.

### 4.7 Traversable wormhole protocol

The appearance of the generators $P^\pm$ in the OPE expansion of (174) is related to the traversable wormhole protocol of Gao, Jafferis, and Wall [26]. In brief, one considers inserting a matter perturbation at early Lorentzian times $-t$ on the right and then acting with a 2-sided interaction at $t = 0$. In the SYK context [38], a popular choice of interaction is

$$\frac{g}{N}\sum_{\alpha=1}^{N}(1 + i\psi^L_\alpha\psi^R_\alpha) = \frac{gp}{N}\bar{n} = \frac{g\Delta}{2}\ell, \tag{184}$$

where $\Delta = 1/p$ is the dimension of a single fermion operator.

To analyze this problem, it is convenient to make a two-sided time translation to a frame where the particle is inserted at $t = 0$ on the right, and the 2-sided interaction occurs at time $\tau_R = it$ and $\tau_L = -it$. Then we have $\tau_- = 2it$ and $\tau_+ = 0$ so (174) gives

$$\delta\ell(-it, +it) = \lambda\left[\frac{P^+ e^{-2ct} + P^- e^{2ct}}{2c} - \frac{1-c^2}{2c^2}(\delta H_L + \delta H_R)\right] \approx \frac{\lambda}{2c}P^- e^{2ct}. \tag{185}$$

Here we have dropped terms that are subleading at large $t$; more precisely, we are working in a limit where $\lambda \to 0$ and $t \to \infty$ holding fixed $\lambda\Delta_V\Delta e^{2ct}$. Using the fact that the $P^-$ operator is an $\mathfrak{sl}_2$ generator, and that operator insertions $V(\theta_1)$ transform as primaries under this $\mathfrak{sl}_2$, we can compute (see Appendix D)

$$\langle V(\theta_2)|e^{-iaP^-}|V(\theta_1)\rangle = \left[\frac{c}{\sin\left(\frac{\phi_2-\phi_1}{2}\right) + i\frac{a}{2}e^{\frac{i}{2}(\phi_1+\phi_2)}}\right]^{2\Delta_V}, \qquad a = \frac{\lambda}{4c}g\Delta e^{2ct}. \tag{186}$$

This is a limit of the twisted correlator computed in [27], see also [39]. (The states $|V(0)\rangle$ and $|V(\pi)\rangle$ have unit norm, so the LHS can be interpreted as an overlap.)

We are interested in the case where the $V$ signal is launched from the right boundary at time zero, so $\theta_1 = 0$. We put the other $V$ operator at time zero on the left boundary, so $\theta_2 = \pi$. These correspond to $\phi_1 = -(1-v)\frac{\pi}{2}$ and $\phi_2 = \pi - (1-v)\frac{\pi}{2}$ and one finds

$$\langle V(\pi)|e^{-iaP^-}|V(0)\rangle = \left[\frac{\cos\frac{\pi v}{2}}{1 + i\frac{a}{2}e^{i\frac{\pi}{2}v}}\right]^{2\Delta_V}.\tag{187}$$

We will make two comments about this formula.

First, we discusss the phase of the term multiplying $a$. If we expand to linear order in $a$, the term that appears should be i times an OTOC with one $V$ and one $W$ operator on each side of the thermofield double. The phase of this correlator is determined by the Lyapunov exponent and is related to the magnitude of inelastic effects in the $2 \to 2$ scattering problem, see [40]. In the present case we can interpret the phase as due to the fact that on the fake circle where the $P^-$ operator acts simply, the $V$ operators are not inserted on the time-symmetric slice (right), even though they are on the $t = 0$ slice of the original thermal circle (left):

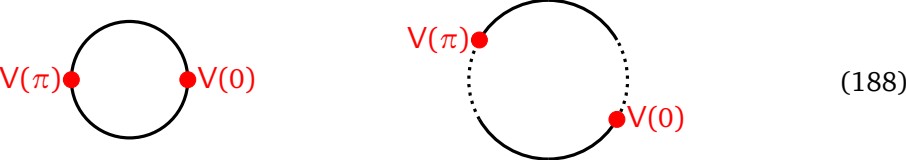

$$(188)$$

Second, we discuss the magnitude of the correlator (187). As a function of $a$, the absolute value of this inner product has a maximum value of one, achieved at $a_* = 2\sin\frac{\pi v}{2}$. At this point the RHS is exactly equal to $e^{-i\pi v\Delta_V}$, so the traversable wormhole protocol maps

$$e^{-ia_*P^-}V_R|\text{TFD}\rangle = e^{-i\pi v\Delta_V}V_L|\text{TFD}\rangle.\tag{189}$$

There is some sense[14] in which the protocol works as well as possible, moving the bulk excitation from the right side to the left. However, thermal fluctuations in the $L$ system mean that the signal cannot be recovered perfectly – for example in the case of infinite temperature (189) holds with $a = 0$ and clearly no transmission takes place in that case. For more discussion on this point, see [39].

## 4.8 The Hilbert space of large $p$ SYK

In the $q < 1$ chord Hilbert space, we have seen that the entire chord Hilbert space (spanned by multiparticle states) can be decomposed into irreps of the chord algebra. Each irrep is associated to a special chord primary state $|\Delta, 0, 0\rangle$, see (57).

This two-sided state can equivalently be viewed as an operator; furthermore, in the $q \to 1$ limit such an operator will become an $\mathfrak{sl}_2$ primary. To check this we may compute:

$$E|\Delta, 0, 0\rangle = \frac{1}{c}[\bar{n}]|\Delta, 0, 0\rangle = \frac{q^{-\Delta/2} - q^{\Delta/2}}{1-q}|\Delta, 0, 0\rangle \approx \Delta|\Delta, 0, 0\rangle.\tag{190}$$

Thus the Hilbert space in the large $p$ SYK limit is a tensor product of wavefunctions of $\ell \geq 0$ and irreps of $\mathfrak{sl}_2$. Clearly all operators which create single matter chords are primaries, but in addition there are multi-particle, or "multi-trace" primaries. We discuss double-trace primaries in Appendix F; here let us just outline the key points. First, by considering $V(\epsilon)W(-\epsilon)$, we

---

[14]The perfect size winding condition in [41] implies an equation similar to (189).

produce the "obvious" double trace primaries that one would expect if we had free fields propagating on rigid AdS$_2$. Indeed, to leading order in $\lambda$, the two pt function of $V(\epsilon)W(-\epsilon)$ just factorizes into the product of $VV$ and $WW$ correlators.

What is more interesting is that these are not all the double trace primaries. Indeed, one can also consider $[V(\epsilon), W(-\epsilon)] = V(\epsilon)W(-\epsilon) - W(-\epsilon)V(\epsilon)$. This is some state which we depict as

$$[\mathsf{V}(\epsilon), \mathsf{W}(-\epsilon)]|0\rangle = e^{\epsilon H}\mathsf{V}e^{-2\epsilon H}\mathsf{W}e^{\epsilon H}|0\rangle - e^{-\epsilon H}\mathsf{W}e^{+2\epsilon H}\mathsf{V}e^{-\epsilon H}|0\rangle \tag{191}$$

$$= \Big| \; \underset{\longrightarrow}{\overline{\phantom{xxxxxxxxx}}} \; \Big\rangle - \Big| \; \underset{\longrightarrow}{\overline{\phantom{xxxxxxxxx}}} \; \Big\rangle . \tag{192}$$

Since the commutator removes the disconnected contribution, the leading 2-pt function of the operator $[V(\epsilon), W(-\epsilon)]$ comes from the Streicher formula (183), which we already processed in (149). This can be expanded in a sum of conformal blocks:

$$\frac{\langle [\mathsf{W}_2, \mathsf{V}_4][\mathsf{V}_3, \mathsf{W}_1]\rangle}{\langle \mathsf{W}_2\mathsf{W}_1\rangle\langle \mathsf{V}_4\mathsf{V}_3\rangle} = \sum_k c_k^2 \mathcal{F}_{\Delta_V, \Delta_W, k}(\chi). \tag{193}$$

See (F.25) in Appendix F for details. The main point for us here is that one gets a sum over *new* primaries that are distinct from the naive double trace primaries. As we explain in Appendix F, such primaries are of the form (94) with $[V, W]_k$ for even $k$ and $\{V, W\}_k$ for odd $k$. We refer to this as $[V, W]_k$. In the $\lambda \to 0$ limit, these primaries decouple from the naive $VW$ primaries. However, at order $\lambda$ they are important: in section 5 of [40], the commutator operator $[V, W]$ was associated to the inelastic part of the final state that is produced in the scattering of $V$ and $W$ particles. So an interpretation of these new primaries is that they are the inelastic states that can be produced in scattering at order $\lambda$.

# 5 Discussion

## 5.1 Chords vs. bulk geometry

To what extent can one associate a (discrete) spacetime geometry to large $p$ SYK and the chord diagrams? One naive try would be to make a graph out of the intersecting chords by, for example, drawing the chords as straight lines connecting equally spaced boundary points. A typical resulting configuration is shown here for 400 chords with $q = 0.98$:

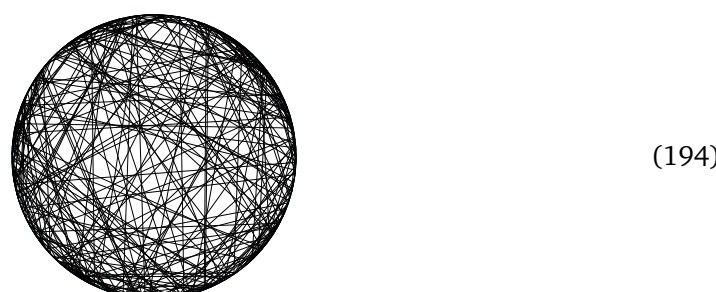

$$\tag{194}$$

However, the decision to draw the chords as straight lines in flat space was arbitrary – a chord diagram does not actually define a graph. It specifies which chords intersect, but it does not specify in what order (along a given chord) these intersections take place. For example, there is

a unique chord diagram for tr($O_1 O_2 O_3 O_1 O_2 O_3$), but we may draw it as two different graphs:[15]

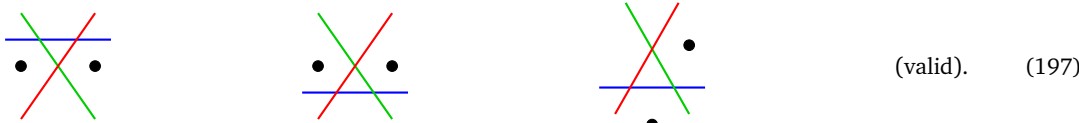

$$\equiv \qquad\qquad\qquad\qquad\qquad (195)$$

More formally, we can say that

$$\text{chord diagrams} = \text{chord graphs/rearrangements,} \qquad (196)$$

where the rearrangements are the Yang-Baxter moves (195).[16]

If we view the chord rearrangements as a kind of gauge symmetry, then one can try to study aspects of a bulk geometry that are manifestly gauge invariant. An example of this is the definition of the length between two boundary points, $\ell = \lambda \bar{n}$. This is well-defined because it does not depend on the order in which chords intersect. As shown in [17], the resulting concept of $\ell$ is closely analogous to the length between boundary points in JT gravity.

One can study further aspects of chord "geometry" by picking a gauge. For example, consider the case of a single matter particle. Then as we just said, the total length $\ell = \lambda(n_L + \Delta + n_R)$ on a slice between two boundary points is gauge-invariant. However, the distance of the matter particle from the midpoint of this slice, $x = \lambda(n_L - n_R)/2$ is not gauge invariant, because it depends on the ordering of the chords relative to the matter particle. In this paper we defined the $x$ operator using a gauge-fixing procedure where the chords are ordered along this slice according to their boundary ordering within the "ket" boundary (which means the part of the boundary below the slice). The hermitian conjugate $x^\dagger$ would be defined using a gauge-fixing where the chords are ordered as in the "bra" or top portion of the boundary.

This is just one possible definition of the $x$ operator, corresponding to one particular gauge-fixing. It has the advantage that this gauge-fixing can be used to define a chord Hilbert space where the operators $\mathfrak{a}_L^\dagger$ and $H_L$ act in a simple way. Its main disadvantage is that the resulting $x$ is not Hermitian, and states with different values of $x$ are not orthogonal. This is because states with the "ket" ordering have a nontrivial inner product with states with the "bra" ordering, defined by the sum over ways these chords can pair together to complete the chord diagram, see [17] and B.

---

[15]Here we define the graph by declaring all boundary operators and all intersections between chords to be vertices. The edges are self-explanatory. On the LHS of (195), the green-red intersection has a graph distance of 2 from the upper red dot, whereas it has graph distance 1 on the RHS.

[16]Before we specify which "slice" we want to associate a Hilbert space, either of (195) are valid and we should only count one. However, once we "slice" the chord diagrams by picking two points on the boundary, we are forced to choose one or the other:

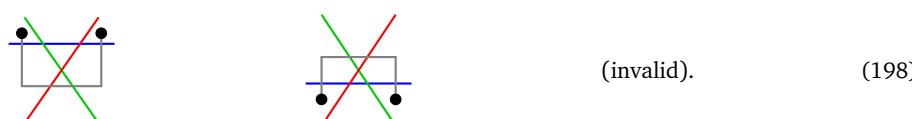

$$\text{(valid).} \qquad (197)$$

The following diagrams are examples of invalid chord arrangements:

$$\text{(invalid).} \qquad (198)$$

We have shown in gray the problem with these chord arrangements. On the left, the ket slice must be to the past of the intersection between the green and red chord. But this forces it to intersect the blue chord twice. Similarly, on the right of (198), the bra slice must intersect to the future of the green-red intersection, which forces a double intersection with the blue chord. We see that adding the black dots (e.g. picking out a slice of the Hilbert space) forces a rearrangement of the chord geometry.

## 5.2 Properties of the $x$ operator and its eigenstates

One consequence of the gauge-fixing we just described is that in the semiclassical limit, the value of $x$ is determined by the position of the matter operator in the ket portion of the boundary. So although the $x$ coordinate resembles the location of the matter particle along a bulk slice, our gauge-fixing procedure relates the ordering of chords on this "bulk" slice to their order on the boundary, so $x$ actually behaves more like a boundary coordinate.

Another consequence of this gauge fixing (and in particular the fact that it makes $x$ a non-Hermitian operator) is that the expectation value of $x$ can be surprisingly large. From the definition $x = \lambda(n_L - n_R)/2$, it is clear that in the subspace with $n_L + n_R = n$ held fixed, the eigenvalues of $x$ are bounded by $\lambda n/2 = \ell/2 = \log(1/c)$, but the expectation value of $x$ can be larger than this. We can see this from (143): values of $\phi$ with $-\pi < \phi < 0$ correspond to normalizable states, and at the extreme ends of this range we have $|x| = \text{arccosh}(1/c) > \log(1/c)$. This is not a contradiction with linear algebra, because the expectation value of a non-Hermitian operator $\mathcal{O}$ is not bounded by its eigenvalues but instead by the square root of the eigenvalues of $\mathcal{O}^\dagger \mathcal{O}$, see (B.19).

An interesting feature of states where $|x| > \ell/2$ is that the distance between the particle and one of the boundaries should become negative:

$$|x| < \ell/2, \tag{199}$$

$$|x| > \ell/2. \tag{200}$$

One can compute the distance from the particle to the boundary by computing $\langle n_R \rangle$ in the chord Hilbert space. Because $n_R$ is not Hermitian, the fact that its eigenvalues are positive does not necessarily mean its expectation value must be. As a simple example of a 1-particle state where $\langle n_R \rangle$ is negative, consider

$$\langle n_R \rangle = \frac{\langle \Omega | V (H_R + \bar{\alpha} H_L) n_R (H_R + \alpha H_L) V | \Omega \rangle}{\langle \Omega | V (H_R + \bar{\alpha} H_L)(H_R + \alpha H_L) V | \Omega \rangle} = \frac{1 + r\bar{\alpha}}{1 + \alpha\bar{\alpha} + r(\alpha + \bar{\alpha})}. \tag{201}$$

For $r\bar{\alpha} < -1$ this expectation value is clearly negative. As a more direct check, we numerically computed the expectation value of various chord number operators $n_L, n_R$ in the chord Hilbert space with fixed but large total chord number $n$ and found good approximate agreement with (143).

There is a simple physical scenario where negative lengths arise, even in the regime where JT gravity is expected to hold. We consider the thermofield double at early times $t_L = t_R = -|t|$. Then we insert some operators on the left and right sides $W_L V_R e^{-iH_L t_L} e^{-iH_R t_R} |\text{TFD}\rangle$. We then evolve this state forwards in time. This produces two particles in the wormhole that are directed towards each other:

$$\tag{202}$$

As time progresses forwards, the two particles approach each other, the middle distance $\ell_M$ shrinks:

$$\tag{203}$$

and the two (naively) pass through each other:

$$\tag{204}$$

However the last picture is not quite right. In particular, we are evolving with $H_L + H_R$ or perhaps the $E$ generator, but acting on the chord Hilbert space, neither of these operators can

change the ordering of the particles. The more accurate picture is that the wormhole gets "folded":

$$\tag{205}$$

The meaning of this picture[17] is that the state of the wormhole is always a superposition of chord states where the red matter particle is to the left of the blue matter particle. Furthermore, we expect that $\langle \ell_M \rangle < 0$ in this late time state, so that the total length $\langle \ell_L \rangle + \langle \ell_M \rangle + \langle \ell_R \rangle$ is dictated by hyperbolic geometry.[18]

### 5.3 Inelastic states

In the scattering experiment that we just described, we can think of (202) as an in state, and (204) as an out state. The scattered component of the final state will be the difference between the out state and the time-evolved in state

$$|\text{out}\rangle - S\,|\text{in}\rangle = \left| \underset{\longrightarrow}{\overset{\longleftarrow}{\rule{6cm}{0pt}}} \right\rangle - \left| \underset{\longrightarrow}{\overset{\longrightarrow}{\rule{6cm}{0pt}}} \right\rangle \tag{206}$$

$$= \left| \underset{\longrightarrow}{\overset{\longleftarrow}{\sim\!\sim\!\sim\!\sim\!\sim\!\sim\!\sim}} \right\rangle . \tag{207}$$

This contains both the elastic and inelastic component of the scattering, but the inelastic component dominates in norm for small $\lambda$ [40].[19] At small $\lambda$, we expect this state can be expanded purely in terms of $[V, W\}_k$ irreps. It would be nice to understand these states better and to see if there is a direct relationship to the phase in the traversable wormhole protocol discussed in section 4.7. In string theory, the commutator state analogous to (207) would be a long string [3]; the notation (207) is perhaps evocative of this.

### 5.4 Open questions

- Is there a formulation of double-scaled SYK in which the appearance of $6j$ and $3j$ symbols of $U_{\sqrt{q}}(\mathfrak{sl}_2)$ in the representation (103) is obvious? Based on the example of JT gravity, perhaps this could be a formulation with some version of $U_{\sqrt{q}}(\mathfrak{sl}_2)$ gauge symmetry. This might also shed new light on the chord algebra, and in particular the subalgebra $U(J)$ which is isomorphic to a subalgebra of $U_{\sqrt{q}}(\mathfrak{sl}_2)$.

- What is the lesson for more general systems with sub-maximal chaos? One possible scenario in quantum mechanical systems is that there are both "fake disk" and "stringy" effects that contribute, see Appendix J.

- The quantum algebra $U_q(\mathfrak{sl}_2)$ makes an appearance in 2D Liouville theory. The double scaled theory is also related to an unconventional Liouville theory, see appendix H. It would be interesting to explore this connection further.

- It would be nice to understand the fake geometry better. A concrete question is whether there is a version of the fake circle/disk that holds in the semiclassical limit, but where two or more operator insertions are heavy, e.g. holding $r$ fixed in the $\lambda \to 0$ limit [8].

---

[17]This picture is reminiscent of zig-zag strings in [42]. Thanks to Juan Maldacena for pointing out this reference.

[18]In Appendix (B.4), we discuss a somewhat similar class of two-particle states which manifestly have negative middle lengths. These states have the unusual property that although their middle lengths are *negative*, they have nearly maximal overlap in the $\lambda \to 0$ limit with states with positive middle lengths. We explain there why this is not in contradiction with linear algebra.

[19]In the Schwarzian approximation, the LHS of (149) vanishes to order $O(1/C)$, see Section 4.2 of [43], where $1/C = 2\lambda$.

## Acknowledgments

We especially thank Zhenbin Yang for initial collaboration and many useful discussions. We thank Ahmed Almheiri, Daniel Bump, Akash Goel, David Kolchmeyer, Juan Maldacena, Henry Maxfield, Jinzhao Wang, and Ying Zhao for helpful discussions.

**Funding information** HL is supported by a Bloch Fellowship. DS is supported in part by DOE grant DE-SC0021085, by the Sloan Foundation, and by a grant from the Simons foundation (926198, DS).

## A  Two other limits of the chord algebra

In the main text, we considered the limit

$$\text{semiclassical:} \qquad q \to 1, \ \ell \text{ fixed}. \tag{A.1}$$

In this appendix we will discuss two other limits. First, we have a close relative of the semiclassical limit, but where we also take the temperature low so that certain fluctuations (the Schwarzian mode) remains quantum:

$$\text{triple-scaling (JT):} \qquad q \to 1, \ \tilde{\ell} \text{ fixed} \qquad (\tilde{\ell} \equiv \ell + 2\log\lambda). \tag{A.2}$$

Second, we have a sort of opposite limit to the semiclassical limit. We refer to this as the long wormhole limit:

$$\text{long wormhole:} \qquad q \text{ fixed, } \ell \to \infty. \tag{A.3}$$

In both cases the algebra will simplify. In the JT limit the entire chord algebra simplifies to the gravitational algebra of [17, 44] with the $\mathfrak{sl}_2$ algebra of [10] as a subalgebra. In the long wormhole limit, the U($J$) algebra will simplify to $U_{\sqrt{q}}(\mathfrak{sl}_2)$.

As an aside, we also mention that in the $q \to 0$ limit, the algebra simplifies as any crossing of chords is completely forbidden. This corresponds to summing only planar Wick contractions and is the subject of "free probability," see [15, 16]

### A.1  The triple scaling/JT limit

The JT gravitational algebra $\mathcal{A}_{\text{JT}}$ was introduced in [44] in the context of semi-classical JT gravity. In [17] it was shown that the algebra exists exactly in the quantum Schwarzian theory (coupled to arbitrary matter). The algebra is given by

$$[\tilde{H}_L, \tilde{H}_R] = 0,$$

$$-\mathrm{i}[\tilde{\ell}, \tilde{H}_{L/R}] = \frac{\tilde{k}_{L/R}}{C},$$

$$[\tilde{\ell}, \tilde{k}_{L/R}] = \mathrm{i},$$

$$[\tilde{k}_L, \tilde{k}_R] = 0, \tag{A.4}$$

$$-\mathrm{i}[\tilde{k}_{L/R}, \tilde{H}_{L/R}] = \tilde{H}_{L/R} - \frac{\tilde{k}_{L/R}^2}{2C},$$

$$-\mathrm{i}[\tilde{k}_{L/R}, \tilde{H}_{R/L}] = \frac{e^{-\tilde{\ell}}}{2C}.$$

More formally, we consider the arbitrary products of $\tilde{H}_L, \tilde{H}_R$ and $\tilde{\ell}$, subject to the above relations.

To derive this from the chord algebra, we set

$$\tilde{H} \equiv H + \frac{2}{\sqrt{(1-q)\lambda}}, \qquad \tilde{\ell} \equiv \ell + 2\log\lambda, \qquad C \equiv \frac{1}{2\lambda}, \qquad \tilde{k}_L \equiv -\mathrm{i}C[\tilde{\ell},\tilde{H}_L]. \tag{A.5}$$

Here, $\tilde{H}$ is the energy above the ground state, and $\tilde{\ell}$ is the renormalized length. Then the exact chord algebra (28) becomes

$$[\tilde{H}_L, \tilde{H}_R] = 0, \tag{A.6}$$

$$-\mathrm{i}[\tilde{\ell}, \tilde{H}_{L/R}] = \frac{\tilde{k}_{L/R}}{C}, \tag{A.7}$$

$$[\tilde{\ell}, \tilde{k}_{L/R}] = \mathrm{i}\left(\sqrt{\frac{\lambda}{1-q}} - \frac{1}{4C}\tilde{H}_{L/R}\right), \tag{A.8}$$

$$[\tilde{k}_L, \tilde{k}_R] = 0, \tag{A.9}$$

$$-\mathrm{i}[\tilde{k}_{L/R}, \tilde{H}_{L/R}] = \sqrt{\frac{1-q}{\lambda}}\frac{2}{1+q}\tilde{H}_{L/R} - \frac{1-q}{\lambda}\frac{2}{1+q}\frac{\tilde{k}_{L/R}^2}{2C} - \frac{1-q}{2(1+q)}\tilde{H}_{L/R}^2, \tag{A.10}$$

$$-\mathrm{i}[\tilde{k}_{L/R}, \tilde{H}_{R/L}] = \frac{e^{-\tilde{\ell}}}{2C}. \tag{A.11}$$

To get the JT algebra (A.4) from this, we drop the final terms in (A.8) and (A.10) and further approximate $1-q \approx \lambda$ and $1+q \approx 2$.

An interesting sub-algebra is the symmetry group of AdS$_2$, as discussed in [10]. Here we show the three symmetry generators of $\mathfrak{sl}_2$,

$$B = \quad P = \quad E = \tag{A.12}$$

In terms of the algebra, these can be written as

$$L_0 = P = \tilde{k}_R - \tilde{k}_L, \tag{A.13}$$

$$L_+ = E + B = e^{\tilde{\ell}/2}(2C\tilde{H}_R - \tilde{k}_R^2 - e^{-\tilde{\ell}}), \tag{A.14}$$

$$L_- = E - B = e^{\tilde{\ell}/2}(2C\tilde{H}_L - \tilde{k}_L^2 - e^{-\tilde{\ell}}), \tag{A.15}$$

where $\mathrm{i}[L_m, L_n] = (m-n)L_{m+n}$. One can show that these symmetry generators commute with $\tilde{\ell}$ and the total momentum $\tilde{k} = \frac{1}{2}(\tilde{k}_L + \tilde{k}_R)$. This allows us to identify an important subset

$$\text{Heisenberg} \times \mathfrak{sl}_2 \subset \mathcal{A}_{\text{JT}}. \tag{A.16}$$

The Heisenberg Lie algebra is spanned by $\{\tilde{\ell}, \tilde{k}, 1\}$. In fact, one can argue that the JT gravitational algebra is in fact the universal enveloping algebra of this Lie algebra (3). This means that starting with the elements in the Lie algebra, we consider arbitrary words made out of these elements. Since $\mathcal{A}_{\text{JT}}$ is generated by $\{\tilde{\ell}, \tilde{H}_{L/R}\}$ and $\tilde{\ell} \in$ Heisenberg, to show (3) we only need to express $H_{L/R}$ in terms of $\tilde{\ell}, \tilde{k}$ and the $\mathfrak{sl}_2$ generators. This can be done by using (A.13)-(A.15), see also [23]. In our conventions:

$$\tilde{H}_R = \frac{1}{2C}\left[\left(\tilde{k} + \frac{1}{2}L_0\right)^2 + L_+ e^{-\tilde{\ell}/2} + e^{-\tilde{\ell}}\right], \tag{A.17}$$

$$\tilde{H}_L = \frac{1}{2C}\left[\left(\tilde{k} - \frac{1}{2}L_0\right)^2 + L_- e^{-\tilde{\ell}/2} + e^{-\tilde{\ell}}\right]. \tag{A.18}$$

To verify this claim, it suffices to reproduce (A.4) using just the commutation relations $[\tilde{\ell}, \tilde{k}] = 1$ and $i[L_m, L_n] = (m-n)L_{m+n}$.

This algebra has explicit multi-particle representations that were worked out in [17].

## A.2 The long wormhole limit

Here we consider the limit $\bar{n} \to \infty$ holding $q$ fixed. In this limit $c \to 0$, and it is convenient to rescale two of the generators $\tilde{J}_{LL} \equiv \frac{1}{c} J_{LL}$ and $\tilde{J}_{RR} \equiv \frac{1}{c} J_{RR}$ so that they remain nonzero. The algebra (47) becomes

$$
\begin{aligned}
q[\tilde{J}_{LL}, \tilde{J}_{RR}] &= J_{LR} - J_{RL}, \\
[J_{LR}, J_{RL}] &= 0, \\
[J_{LL}, J_{LR}]_q &= -J_{LL}, \\
[J_{RR}, J_{RL}]_q &= -J_{RR}, \\
[J_{RL}, J_{LL}]_q &= -J_{LL}, \\
[J_{LR}, J_{RR}]_q &= -J_{RR}.
\end{aligned}
\tag{A.19}
$$

This turns out to be exactly equivalent to the algebra $U_{\sqrt{q}}(\mathfrak{sl}_2)$ (I.1), after defining

$$
\begin{aligned}
K &= 1 - (q-1)J_{LR}, \quad K^{-1} = 1 - (q-1)J_{RL}, \\
E_+ &= q^{3/4}\tilde{J}_{RR}, \quad E_- = q^{3/4}\tilde{J}_{LL}.
\end{aligned}
\tag{A.20}
$$

**Representations:** One can write more explicit formulas for the $K, E_\pm$ generators as follows. First, consider a single-particle state $|n_L, n_R\rangle$ that the U($J$) algebra acts on. The sum $n_L + n_R$ commutes with U($J$), so we will suppress this variable and label the single-particle state by $|y\rangle$, where $y \equiv (n_L - n_R)/2$ is the signed distance of the particle from the "midpoint." This makes the limit $n_L + n_R \to \infty$ effortless. More generally, we describe a state of $m$ particles $|n_0, n_1, \ldots, n_m\rangle$ by suppressing $\bar{n}$ and using

$$
|y_1, \ldots, y_m\rangle,
\tag{A.21}
$$

where the $y_i$ are the signed distances of the various particles from the midpoint

$$
y_1 = n_0 + \frac{\Delta_1}{2} - \frac{\bar{n}}{2}, \qquad y_i = y_{i-1} + n_{i-i} + \frac{\Delta_i + \Delta_{i-1}}{2}.
\tag{A.22}
$$

We can define an operator that shifts the first $i$ particles to the right as

$$
\mathfrak{a}_0^\dagger \alpha_i |y_1, \ldots, y_i, \ldots, y_m\rangle = |y_1 + 1, \ldots, y_i + 1, \ldots, y_m\rangle.
\tag{A.23}
$$

Then by writing out the explicit formulas for (A.20) in terms of oscillators and taking the large $\bar{n}$ limit, one finds

$$
K = \mathfrak{a}_0^\dagger \alpha_m = \text{shift all particles to the right},
\tag{A.24}
$$

$$
K^{-1} = \mathfrak{a}_m^\dagger \alpha_0 = \text{shift all particles to the left},
\tag{A.25}
$$

$$
E_- = \frac{q^{3/4}}{1-q}\left(-q^{y_1 - \Delta_1/2} + \sum_{i=1}^{m-1} \mathfrak{a}_0^\dagger \alpha_i (q^{y_i + \Delta_i/2} - q^{y_{i+1} - \Delta_{i+1}/2}) + K q^{y_m + \Delta_m/2}\right),
\tag{A.26}
$$

$$
E_+ = \frac{q^{3/4}}{1-q}\left(K^{-1}q^{-y_1 + \Delta_1/2} + \sum_{i=2}^{m} \mathfrak{a}_m^\dagger \alpha_{i-1}(q^{-y_m + \Delta_m/2} - q^{-y_{m-1} - \Delta_{m-1}/2}) - q^{-y_m - \Delta_m/2}\right).
\tag{A.27}
$$

These formulas give an infinite dimensional representations of $U_{\sqrt{q}}(\mathfrak{sl}_2)$ with the property that $K$ is both Hermitian and unitary, and $E_\pm$ are Hermitian. We expect the one-particle representation is irreducible and the multiparticle representations are reducible to one-particle representations.

**Coproduct:** One can consider a tensor product

$$|y_1, \ldots, y_i\rangle \otimes |y_{i+1}, \ldots, y_m\rangle. \tag{A.28}$$

Note that this tensor product in the $y$ variables is slightly different from the tensor product in the $n$ variables that was used in the discussion of the chord algebra. The generators acting on the above tensor product can be obtained by a coproduct

$$D'(K) = K \otimes K, \qquad D'(E_+) = 1 \otimes E_+ + E_+ \otimes K^{-1}, \qquad D'(E_-) = K \otimes E_- + E_- \otimes 1. \tag{A.29}$$

This is a legal coproduct for $U_{\sqrt{q}}(\mathfrak{sl}_2)$ that differs from the textbook one (I.4) by

$$D' = P(\tau \otimes \tau) \circ D_{\text{textbook}} \circ \tau^{-1}, \tag{A.30}$$

where $P(v \otimes w) = w \otimes v$ swaps the two tensor factors [45] and $\tau$ is the anti-automorphism[20] defined by $\tau(K^\pm) = K^\mp$ and $\tau(E_\pm) = E_\pm$.

# B  Inner product

In this section, we discuss the inner product of states in the chord Hilbert space, see [16,17]. The ket vectors in this Hilbert space can be obtained by slicing open chord diagrams. Given two points on the boundary (indicated by the gray points below), there is a unique "ket slice" (dashed gray) which is defined so that all chords which cross the ket slice have not intersected in the Euclidean *past*, for example:[21]

$$\Rightarrow \quad ||\mathcal{U}|\rangle = |1,1\rangle. \tag{B.1}$$

Similarly, bra vectors are defined by "bra slices" which are defined so that all chords which cross this slice have not intersected in the Euclidean *future*, for example:

$$\Rightarrow \quad \langle|\mathcal{U}|| = \langle 1,1|. \tag{B.2}$$

The inner product between two chord states corresponds to the "middle region" in between the bra slice and the ket slice. It is the region where chords which cross the ket or bra slice may intersect each other. The numerical value of the inner product $\langle v|w\rangle$ is determined by summing over all ways that chords may intersect, weighted by the appropriate factors of $q^{\#\text{intersections}}$ for each intersection of $H$ chords, (or factors of $q^{\Delta_i}, q^{\Delta_i \Delta_j}$ for $H$-matter intersections or matter-matter intersections, respectively.)

---

[20]An anti-automorphism means $\tau$ satisfies the same algebra but with multiplication defined in the reverse order.

[21]As explained in footnote 16, rearrangement of the chord picture may be needed before such a slice exists.

## B.1 Single-particle states

For states with a single particle, the inner product is determined by the chord algebra, up to an overall choice of normalization that can be fixed by setting $\langle 0, 0 | 0, 0 \rangle = 1$. To see this, one can use (61) to write a recursion relation that lowers $n'_L$ [17]:

$$\langle n'_L, n'_R | n_L, n_R \rangle = \langle n'_L - 1, n'_R | \mathfrak{a}_L | n_L, n_R \rangle \tag{B.3}$$
$$= [n_L] \langle n'_L - 1, n'_R | n_L - 1, n_R \rangle + q^{\Delta + n_L} [n_R] \langle n'_L - 1, n'_R | n_L, n_R - 1 \rangle, \tag{B.4}$$

and a similar one that lowers $n'_R$:

$$\langle n'_L, n'_R | n_L, n_R \rangle = \langle n'_L, n'_R - 1 | \mathfrak{a}_L | n_L, n_R \rangle \tag{B.5}$$
$$= [n_R] \langle n'_L, n'_R - 1 | n_L, n_R - 1 \rangle + q^{\Delta + n_R} [n_L] \langle n'_L, n'_R - 1 | n_L - 1, n_R \rangle. \tag{B.6}$$

The inner product is then a sum over lattice paths that begin at $(n_L, n_R)$ and end at $(0, 0)$ after a sequence of moves of the form $n_L \to n_L - 1$ or $n_R \to n_R - 1$. Specifically, we make $n'_L$ moves with weighting factors (B.4) and $n'_R$ moves with weighting factors (B.6):

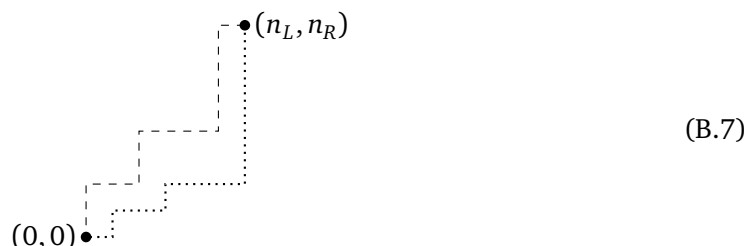

$$\tag{B.7}$$

The result will have a simple factor of $[n_L]![n_R]!$ that is independent of the path of the walker, and the remaining sum over paths can be evaluated using the $q$-Binomial coefficient

$$\sum_{1 \le x_1 < \cdots < x_k \le m} q^{x_1 + x_2 + \cdots + x_k} = q^{k(k+1)/2} \frac{[m]!}{[k]![m-k]!}. \tag{B.8}$$

We will not show the details of this, only the answer. Setting

$$n_L = \tfrac{n}{2} + y, \qquad n_R = \tfrac{n}{2} + y, \qquad n'_L = \tfrac{n}{2} + y', \qquad n'_R = \tfrac{n}{2} - y', \tag{B.9}$$

one can use the symmetry of the inner product to assume $|y| > y'$ and $y > 0$. Then

$$\langle n'_L, n'_R | n_L, n_R \rangle = \sum_{0 \le k \le \frac{n}{2} - y} q^{k^2 + (2\Delta + y - y')k + (y - y')\Delta} \frac{[\tfrac{n}{2} + y]![\tfrac{n}{2} - y]![\tfrac{n}{2} + y']![\tfrac{n}{2} - y']!}{[k]![y - y' + k]![\tfrac{n}{2} + y' - k]![\tfrac{n}{2} - y - k]!}. \tag{B.10}$$

More generally, it is the same sum, restricted to the range for $k$ such that the arguments of the $q$-factorials in the denominator are positive.

In the semiclassical limit, we claimed in the main text that the inner product becomes

$$\langle n'_L, n'_R | n_L, n_R \rangle = [n]! \left( \frac{(1 - c^2)/2}{\cosh \frac{x - x'}{2} - c \cosh \frac{x + x'}{2}} \right)^{2\Delta}, \tag{B.11}$$

where $x = \lambda y$ and the $y$ coordinate is defined in (B.9). For the purposes of this formula, we can set either $c^2 = q^n$ or $c^2 = q^{n+\Delta}$ – the difference is higher order in $\lambda$. To verify (B.11), we need to check the normalization and the recursion relations. For the normalization, in the case $n'_L = n_L = n$ and $n'_R = n_R = 0$ we get the correct zero-particle inner product $[n]!$. Due to the

symmetry of (B.11) under $x \leftrightarrow x'$ and $(x, x') \leftrightarrow (-x, -x')$ we can just check one recursion relation (B.4), which amounts to

$$(1-c^2)\left(\frac{(1-c^2)/2}{\cosh\frac{x-x'}{2} - c\cosh\frac{x+x'}{2}}\right)^{2\Delta} \overset{?}{=} (1-ce^{-x})\left(\frac{(1-c^2e^\lambda)/2}{\cosh\frac{x-x'}{2} - ce^{\lambda/2}\cosh\frac{x+x'-\lambda}{2}}\right)^{2\Delta} \quad (B.12)$$

$$+ e^{-\lambda\Delta-x}c(1-ce^x)\left(\frac{(1-c^2e^\lambda)/2}{\cosh\frac{x-x'+\lambda}{2} - ce^{\lambda/2}\cosh\frac{x+x'}{2}}\right)^{2\Delta}.$$

This equation is trivially satisfied at order $\lambda^0$, and nontrivially satisfied at order $\lambda$.

## B.2 Two-particle states

For states with two particles $\textcolor{red}{V}, \textcolor{blue}{W}$, we can use (72) to write down a recursion relation that lowers $n_L'$. If the particles are in the order $\textcolor{red}{V}\textcolor{blue}{W}$ in both the ket, the recursion is

$$\langle n_L', n_1', n_R' | n_L, n_1, n_R\rangle = \langle n_L'-1, n_1', n_R' | \mathfrak{a}_L | n_L, n_1, n_R\rangle, \quad (B.13)$$

$$= [n_L]\langle n_L'-1, n_1', n_R' | n_L-1, n_1, n_R\rangle$$

$$+ q^{n_L}r_V[n_1]\langle n_L'-1, n_1', n_R' | n_L, n_1-1, n_R\rangle$$

$$+ q^{n_L+n_1}r_V r_W[n_R]\langle n_L'-1, n_1', n_R' | n_L, n_1, n_R-1\rangle. \quad (B.14)$$

One can similarly write down three further equations that lower $n_R', n_L, n_R$. These can be used to reduce to the case where $n_L' = n_R' = n_L = n_R = 0$, for which the inner product is determined by the 0-particle inner product

$$\langle 0, n_m, 0 | 0, n_m, 0\rangle = [n_m]!, \qquad\qquad \text{no crossing}, \quad (B.15)$$

$$\langle 0, n_m, 0 | 0, n_m, 0\rangle = [n_m]!(r_1 r_2)^{n_m} q^{\Delta_1\Delta_2}, \qquad \text{crossing}. \quad (B.16)$$

To analyze the inner product in the limit $\lambda \to 0$, it is convenient to define $x_V, x_W$ coordinates that measure the locations of the $V$ and $W$ particles:

$$n_L = \tfrac{1}{2}n + y_V,$$
$$n_1 = y_W - y_V,$$
$$n_R = \tfrac{1}{2}n - y_W, \quad (B.17)$$
$$x_{V/W} = \lambda y_{V/W},$$
$$\lambda n = -2\log c.$$

One can check that (B.14) is satisfied to order $\lambda$ by

$$\langle n_L', n_1', n_R' | n_L, n_1, n_R\rangle = [n]!\left[\frac{(1-c^2)/2}{\cosh\frac{x_V-x_V'}{2} - c\cosh\frac{x_V+x_V'}{2}}\right]^{2\Delta_V}\left[\frac{(1-c^2)/2}{\cosh\frac{x_W+x_W'}{2} - c\cosh\frac{x_W-x_W'}{2}}\right]^{2\Delta_W}.$$
$$(B.18)$$

In principle there could be a different proportionality constant in the crossed vs. uncrossed cases. This can be fixed by considering the case where $x_V \to x_W$ and $x_V' \to x_W'$. Then the ratio between the crossed and uncrossed inner products is just $q^{\Delta_V\Delta_W} \approx 1$.

## B.3 Non-hermiticity of position operators

The position operator $x$ of a single particle (relative to the center of the wormhole) is a non-Hermitian operator with respect to the inner product that we defined above. Similarly, if we

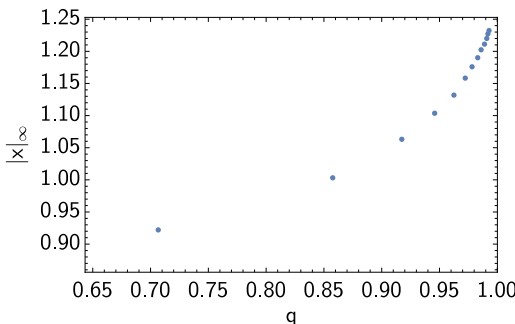

Figure 1: We plot $|x|_\infty$ for different values of $q$ with $q^{n/2} = 1/2$ and $\Delta = 1$. The maximum eigenvalue of $x$ is constant and equal to $\log 2 \approx 0.693$. As $q \to 1$, we expect $|x|_\infty \geq \operatorname{arccosh}(2) \approx 1.317$.

have multiple particles, the various positions $x_i$ or the lengths $\ell_i$ between the particles are non-Hermitian.

The non-Hermiticity of these operators means that their expectation values can have counter-intuitive properties. For example, we are used to the idea that the expectation value of a Hermitian operator is bounded by its maximum eigenvalues. However, this is not necessarily true for a non-Hermitian operator. Instead,

$$\frac{|\langle \mathbf{v}, \mathcal{O}\mathbf{v}\rangle|}{\langle \mathbf{v}, \mathbf{v}\rangle} \leq \sqrt{\frac{\langle \mathcal{O}\mathbf{v}, \mathcal{O}\mathbf{v}\rangle}{\langle \mathbf{v}, \mathbf{v}\rangle}} \leq |\mathcal{O}|_\infty = \sqrt{\text{max eigenvalue}(\mathcal{O}^\dagger \mathcal{O})}. \tag{B.19}$$

For a general inner product matrix $\langle \mathbf{v}, \mathbf{w}\rangle = \mathbf{v}^\mathsf{T}\mathbf{G}\mathbf{w}$ where $\mathsf{T}$ is matrix Hermitian conjugation (e.g. with respect to the trivial inner product), $|\mathcal{O}|_\infty$ is equal to the maximum singular value of $\mathbf{G}^{1/2}\mathbf{O}\mathbf{G}^{-1/2}$, which is equivalent to the square root of the maximum eigenvalue of the matrix $\mathbf{G}^{1/2}\mathbf{O}\mathbf{G}^{-1}\mathbf{O}^\mathsf{T}\mathbf{G}^{1/2}$. If $\mathbf{G}$ has very small eigenvalues, it is possible that the singular values of $\mathbf{G}^{1/2}\mathbf{O}\mathbf{G}^{-1/2}$ could be quite large even if the eigenvalues of $\mathbf{G}^{1/2}\mathbf{O}\mathbf{G}^{-1/2}$ are not.

A related counterintuitive property is that even if two states have a large overlap, their expectation values of a non-Hermitian operator can be surprisingly different. Let $\mathbf{v}, \mathbf{w}$ be two states such that $\langle \mathbf{v}, \mathbf{v}\rangle = \langle \mathbf{w}, \mathbf{w}\rangle \approx \langle \mathbf{v}, \mathbf{w}\rangle$. The $\approx$ sign means that the error is controlled by a small parameter $\lambda$. Then on general grounds,

$$\langle \mathbf{v}, \mathcal{O}\mathbf{v}\rangle - \langle \mathbf{w}, \mathcal{O}\mathbf{w}\rangle = \langle \mathbf{v}, \mathcal{O}(\mathbf{v}-\mathbf{w})\rangle - \langle \mathbf{w}-\mathbf{v}, \mathcal{O}\mathbf{w}\rangle \tag{B.20}$$

$$\leq |\langle \mathbf{v}, \mathcal{O}(\mathbf{v}-\mathbf{w})\rangle + |\langle \mathbf{w}-\mathbf{v}, \mathcal{O}\mathbf{w}\rangle| \tag{B.21}$$

$$\leq \sqrt{\langle \mathbf{v}, \mathbf{v}\rangle \langle \mathcal{O}(\mathbf{v}-\mathbf{w}), \mathcal{O}(\mathbf{v}-\mathbf{w})\rangle} + \sqrt{\langle \mathbf{w}-\mathbf{v}, \mathbf{w}-\mathbf{v}\rangle \langle \mathcal{O}\mathbf{w}, \mathcal{O}\mathbf{w}\rangle} \tag{B.22}$$

$$\leq 2|\mathcal{O}|_\infty \sqrt{\langle \mathbf{v}-\mathbf{w}, \mathbf{v}-\mathbf{w}\rangle \langle \mathbf{v}, \mathbf{v}\rangle}. \tag{B.23}$$

## B.4 Lengths of 2-particle states

One can obtain states that have some qualitative similarity with the late time wormhole states with two matter particles that we discussed in Section 5.3. We obtain such states by starting with a state where the particle is in the fake region, e.g., a state like in (200) and then acting with $W_R$:

$$W_R \left| \begin{array}{c} \overset{\leftarrow\ a_R\ \rightarrow}{\longrightarrow} \end{array} \right\rangle = \left| \begin{array}{c} \longrightarrow \end{array} \right\rangle . \tag{B.24}$$

Then one can act with the translation operator $e^{iPa}$ to move both particles out of the fake region:

$$e^{-iPa}W_R e^{iPa_R}V_R|\text{TFD}\rangle = \left|\begin{array}{c}\xleftarrow{\quad a_R \quad}\\ \xleftarrow{\quad a \quad}\end{array}\right\rangle. \tag{B.25}$$

This state is prepared by evolving with an "out-of-time-order" Schwinger-Keldysh contour. However since the "Hamiltonian" associated with this contour is $P$, we call this an "out-of-space-order" contour. One may wonder how the state specified above compares to the state where the red particle is to the left of the blue particle (but the positions relative to the center are the same), e.g.,

$$\left\langle\begin{array}{c}\bullet\qquad\bullet\end{array}\right|\left|\begin{array}{c}\bullet\\ \bullet\end{array}\right\rangle$$

$$= \langle\text{TFD}|(e^{-iP(a-a_R)}V_R e^{iP(a-a_R)}e^{-iPa}W_R e^{iPa})^\dagger e^{-iPa}W_R e^{iPa_R}V_R|\text{TFD}\rangle \tag{B.26}$$

$$= \langle\text{TFD}|W_R e^{iPa_R}V_R e^{-iPa_R}W_R e^{iPa_R}V_R|\text{TFD}\rangle. \tag{B.27}$$

The bra and ket states here have an inner product that tends to one in the limit $q \to 1$, but $\langle\ell_M\rangle$ remains fixed and of opposite sign for the two states. This appears to require $|\ell_M|_\infty \to \infty$.

## C  Characterizing the U($J$) algebra

The algebra (47) involves $c$ explicitly. However, one can define four new combinations of the generators that eliminate this dependence. These combinations are:

$$\begin{aligned}
\mathcal{L}_+ &\doteq b\frac{c^2+(q-1)J_{RR}}{(1-q)c^{4/3}},\\
\mathcal{L}_- &\doteq b\frac{c^2+(q-1)J_{LL}}{(1-q)c^{4/3}},\\
\kappa_+ &\doteq -\frac{1+(1-q)J_{LR}}{bc^{2/3}},\\
\kappa_- &\doteq -\frac{1+(1-q)J_{RL}}{bc^{2/3}},\\
b &= q^{1/6}(1-q)^{1/2}
\end{aligned} \tag{C.1}$$

Since we have shifted the generators by the identity, $\mathcal{L}_\pm, \kappa_\pm$ do not annihilate the thermofield double. Their algebra is

$$\begin{aligned}
[\kappa_\pm, \mathcal{L}_\pm]_q &= -1,\\
[\mathcal{L}_\pm, \kappa_\mp]_q &= -1,\\
[\mathcal{L}_-, \mathcal{L}_+] &= -\left(\frac{1-q}{q}\right)^{1/2}(\kappa_+ - \kappa_-),\\
[\kappa_-, \kappa_+] &= -\left(\frac{1-q}{q}\right)^{1/2}(\mathcal{L}_+ - \mathcal{L}_-).
\end{aligned} \tag{C.2}$$

We chose the factor of $b$ in (C.1) so that the same factor appears in the last two lines of (C.2).

Notice that the algebra (C.2) has an automorphism $A$ that preserves the algebra (C.2):

$$A(\kappa_\pm) = -\mathcal{L}_\pm, \quad A(\mathcal{L}_\pm) = -\kappa_\mp. \tag{C.3}$$

We can use this indirectly to show that the algebra generated by $\mathcal{L}_\pm, \kappa_\pm$ is a proper subalgebra of $U_{\sqrt{q}}(\mathfrak{sl}_2)$. We will define the $U_{\sqrt{q}}(\mathfrak{sl}_2)$ algebra following the conventions[22] of [46]:

$$KK^{-1} = K^{-1}K = 1, \quad KE_\pm K^{-1} = q^{\pm 1}E_\pm,$$
$$[E_+, E_-] = \frac{K - K^{-1}}{q^{1/2} - q^{-1/2}}. \tag{C.4}$$

The notation $U(\cdots)$ means that we can multiply/take linear combinations of the generators. For example, $U(\mathfrak{sl}_2)$ contains not only the usual Lie algebra $\mathfrak{sl}_2$ but also elements such as $L_0 L_1 L_{-1}^2$, which cannot be written as a Lie bracket of two generators. (The Lie bracket conditions are imposed as relations on this universal enveloping algebra.) Then, there exists an algebra homomorphism $\rho : U(J) \to U_{\sqrt{q}}(\mathfrak{sl}_2)$:

$$\rho(\kappa_\pm) = -b^{-1}\left(q^{1/2} + (1-q)E_\pm\right)K^\pm,$$
$$\rho(\mathcal{L}_\pm) = bq^{-1/2}\left(\frac{1}{1-q}K^\mp - E_\pm\right). \tag{C.5}$$

One can check explicitly that the operators $\rho(\kappa_\pm), \rho(\mathcal{L}_\pm)$ satisfy the algebra (C.2). Using the automorphism $A$ we can also define an alternative homorphism $\rho' = \rho \circ A$:

$$\rho'(\kappa_\pm) = -\rho(\mathcal{L}_\pm) = -bq^{-1/2}\left(\frac{1}{1-q}K^\mp - E_\pm\right),$$
$$\rho'(\mathcal{L}_\pm) = -\rho(\kappa_\mp) = b^{-1}\left(q^{1/2} + (1-q)E_\pm\right)K^\pm. \tag{C.6}$$

Given that we found an algebra homomorphism $\rho: U(J) \to U_{\sqrt{q}}(\mathfrak{sl}_2)$, it is natural to wonder whether whether $\rho$ could be an isomorphism. We will now prove that $\rho$ is not an isomorphism by considering the action of the automorphism $A$. If the algebras were isomorphic, there must exist an automorphism $\theta$ of $U_{\sqrt{q}}(\mathfrak{sl}_2)$ such that $\rho \circ A = \theta \circ \rho$. So we must be able to find a $\theta$ that satisfies, e.g.,

$$\rho'(\mathcal{L}_\pm) = b^{-1}q^{1/2}K^\mp + b^{-1}(1-q)E_\mp K^\mp$$
$$= bq^{-1/2}(1-q)^{-1}\theta(K^\mp) - bq^{-1/2}\theta(E_\pm). \tag{C.7}$$

On the other hand, if $q$ isn't a root of unity, it has been proved [47] (see also Proposition 3 of [46]) that the automorphism group of $U_{\sqrt{q}}(\mathfrak{sl}_2)$ is generated by $\tau$ and $\vartheta_{\alpha,n,\nu}$, where

$$\tau(E_\pm) = E_\mp, \quad \tau(K^\pm) = K^\mp,$$
$$\vartheta_{\alpha,n,\nu}(E_+) = \alpha K^n E_+, \quad \vartheta_{\alpha,n,\nu}(E_-) = \nu\alpha^{-1}q^{-n}K^{-n}E_-, \quad \vartheta_{\alpha,n,\nu}(K) = \nu K, \tag{C.8}$$

where $\alpha$ is any non-zero complex number, $n \in \mathbb{Z}$ and $\nu \in \{-1, +1\}$. For $A$ to be compatible with this theorem, the first term in (C.7) implies that $\theta(K^\pm) \propto K^\pm$ but the second term implies that $\theta$ sends $E_\pm$ to something proportional to $E_\mp$. However this implies $\theta(K^\pm) \propto K^\mp$ which is a contradiction. Hence we conclude that $\rho$ is not an isomorphism.

## C.1 Casimir

The $U_{\sqrt{q}}(\mathfrak{sl}_2)$ algebra has a Casimir element which is quadratic in the generators and commutes with all the elements [46]. A simple question is whether the $U(J) \subset U_{\sqrt{q}}(\mathfrak{sl}_2)$ subalgebra that

---

[22]This convention is common in the math literature on quantum groups. It differs slightly from [14].

we found also has a Casimir. In fact, it is possible to write the standard $U_{\sqrt{q}}(\mathfrak{sl}_2)$ Casimir in terms of the $U(J)$ generators:

$$
\begin{aligned}
\Omega &= \left(q^{1/2} - q^{-1/2}\right)^2 E_+ E_- + K q^{-1/2} + K^{-1} q^{1/2} \\
&= b\left[\left(\frac{1-q}{q}\right)^{1/2} \mathcal{L}_- \mathcal{L}_+ - \kappa_+ - q^{-1}\kappa_-\right] - \frac{1}{q}.
\end{aligned}
\tag{C.9}
$$

Using some experimentation, one can upgrade $\Omega$ to a Casimir $\hat{\Omega}$ of the full chord algebra by defining

$$
\hat{\Omega} = q^{\bar{n}/3}(\Omega + q^{-1}) - q^{\bar{n}}.
\tag{C.10}
$$

We can write the full Casimir in terms of the $\mathfrak{a}$ operators:

$$
\begin{aligned}
2\hat{\Omega} &= q^{1-\bar{n}}\{q^{m_L}, q^{m_R}\} + (1-q^2)\left(\mathfrak{a}_L^\dagger \mathfrak{a}_R + \mathfrak{a}_R^\dagger \mathfrak{a}_L\right) + 2q^{\bar{n}}, \\
\mathfrak{a}_{L/R}^\dagger \mathfrak{a}_{L/R} &= [m_{L/R}] = \frac{1-q^{m_{L/R}}}{1-q}.
\end{aligned}
\tag{C.11}
$$

One can check that $\hat{\Omega}$ commutes with $H_L, H_R, \bar{n}$, and therefore commutes with the entire algebra.

## C.2 Finite $N, p$ numerics for the Casimir

We simulated two copies of $N = 14$ SYK with $p = 4, 6, 8$. For each instance of the random draw, we normalized the Hamiltonian $H$ such that $2^{-N} \operatorname{tr} H^2 = \lambda^{-1}$. Then we used the expressions (16) and (29) to write finite $N, p$ versions of the chord oscillators and finally computed $\hat{\Omega}$ from (C.11).

We started with the maximally entangled state $|\Omega\rangle$ which satisfies $(\psi_i^L + i\psi_i^R)|0\rangle = 0$. We then considered states obtained by acting with a small number of fermions, e.g., $\psi_i|0\rangle, \psi_i\psi_j|0\rangle, \psi_i\psi_j\psi_k|0\rangle$. We computed the expectation value of the Casimir $\hat{\Omega}$ in these states. To get statistics, we averaged over the choice of operator that we inserted. For instance for $\Delta = 2/p$ case, we considered $\psi_i\psi_{i+1}$ and averaged over the choice of $i$. Note that since we have the maximally entangled state, it does not matter whether we insert left or right fermions. We then compared the expectation value to the theoretical prediction in the double scaled limit (67):

$$
\hat{\Omega} = q^{1-\Delta} + q^\Delta, \quad \Delta = \{1/p, 2/p, 3/p\}.
\tag{C.12}
$$

For a direct test of whether $\hat{\Omega}$ is a symmetry, one can compute

$$
-\frac{\langle 0|[H_L, \hat{\Omega}]^2|0\rangle}{\langle 0|H_L^2|0\rangle \langle 0|\hat{\Omega}^2|0\rangle} \approx 0.5, \quad N = 14, \quad p = 4.
\tag{C.13}
$$

Here the agreement with the double scaling limit is not very impressive, although the RHS would be four times larger if $H_L$ and $\hat{\Omega}$ were random operators.

# D Single-particle representations of $\mathfrak{sl}_2$

In this appendix we will discuss the representation of $\mathfrak{sl}_2$ associated to single-particle chord Hilbert space states (139) and (132).

First, let's review the representation of $\mathfrak{sl}_2$ as isometries of hyperbolic space. The hyperbolic disk can be viewed as the locus $X \cdot X = -1$, embedded in Minkowski space with signature

Table 1: Results of numerics for the Casimir $\hat{\Omega}$, for $N = 14$ SYK and different values of of $p$.

| Operator | $p$ | Double scaled prediction | $N$=13 SYK | Error |
|---|---|---|---|---|
| $\psi_i$ | 4 | 0.7448 | $0.722 \pm 0.002$ | $-3\%$ |
| | 6 | 0.4381 | $0.4498 \pm 0.0005$ | $+3\%$ |
| | 8 | 0.3192 | $0.3103 \pm 0.0002$ | $-3\%$ |
| $\psi_i \psi_j$ | 4 | 0.638 | $0.605 \pm 0.003$ | $-5\%$ |
| | 6 | 0.213 | $0.207 \pm 0.001$ | $-3\%$ |
| | 8 | 0.103 | $0.114 \pm 0.001$ | $+11\%$ |
| $\psi_i \psi_j \psi_k$ | 4 | 0.745 | $0.673 \pm 0.008$ | $-10\%$ |
| | 6 | 0.153 | $0.133 \pm 0.002$ | $-13\%$ |
| | 8 | 0.0357 | $0.037 \pm 0.002$ | $+3\%$ |

$(-, +, +)$. Two explicit parametrizations will be useful:

$$
\begin{aligned}
X^0 &= \cosh\rho &&= \cosh\sigma \cosh x\,, \\
X^1 &= \sinh\rho \cos\phi &&= \cosh\sigma \sinh x\,, \\
X^2 &= \sinh\rho \sin\phi &&= \sinh\sigma\,.
\end{aligned}
\tag{D.1}
$$

The isometries of the hyperbolic disk act on the $X$ coordinate as

$$
\mathsf{b} = \begin{pmatrix} 0 & 0 & 0 \\ 0 & 0 & -1 \\ 0 & 1 & 0 \end{pmatrix}, \qquad \mathsf{e} = \begin{pmatrix} 0 & 0 & 1 \\ 0 & 0 & 0 \\ 1 & 0 & 0 \end{pmatrix}, \qquad \mathsf{p} = \begin{pmatrix} 0 & i & 0 \\ i & 0 & 0 \\ 0 & 0 & 0 \end{pmatrix}.
\tag{D.2}
$$

So in particular the $\mathsf{b}$ generator increases the $\phi$ coordinate and the $\mathsf{p}$ generator increases the $x$ coordinate. These matrices satisfy the $\mathfrak{sl}_2$ algebra

$$
[\mathsf{e}, \mathsf{p}] = i\mathsf{b}, \qquad [\mathsf{b}, \mathsf{e}] = i\mathsf{p}, \qquad [\mathsf{b}, \mathsf{p}] = i\mathsf{e}.
\tag{D.3}
$$

Now let's relate this representation to chord states with one matter particle inserted. Consider a state $|\phi\rangle$ in the chord Hilbert space obtained by acting with a matter operator at a position $\phi$ on the fake circle, $|\phi\rangle = |O(\theta)\rangle$ with $\theta$ and $\phi$ related by (137). Then we have

$$
|\phi\rangle \sim \lim_{\rho \to \infty} (ce^\rho)^\Delta \mathbf{O}(\rho, \phi)|\mathsf{TFD}\rangle.
\tag{D.4}
$$

In this expression, the $\sim$ means "transforms under $\mathfrak{sl}_2$ the same way as," where the transformation of the LHS under $\mathfrak{sl}_2$ is (139) and the transformation of the RHS under $\mathfrak{sl}_2$ is obtained by acting with the corresponding matrices (D.2) on the $X$ coordinate parametrized by $\rho, \phi$. The $\mathfrak{sl}_2$ representation determines the inner product up to normalization

$$
\langle \mathsf{TFD}|\mathbf{O}(X')\mathbf{O}(X)|\mathsf{TFD}\rangle = e^{-\Delta \mathrm{dist}(X \to X')} \approx \left( \frac{-1}{2X' \cdot X} \right)^\Delta.
\tag{D.5}
$$

The normalization in (D.4) was defined so that this inner product matches the inner product of the states on the LHS. Concretely, after taking the $\rho \to \infty$ limit one finds

$$
\langle \phi_2 | \phi_1 \rangle = \left( \frac{-2c^2}{Y(\phi_2) \cdot Y(\phi_1)} \right)^\Delta = \frac{c^{2\Delta}}{(\sin\frac{\phi_2 - \phi_1}{2})^{2\Delta}}\,,
\tag{D.6}
$$

where

$$Y(\phi) = \{1, \cos(\phi), \sin(\phi)\}. \tag{D.7}$$

More usefully, the $\mathfrak{sl}_2$ representation also determines the inner product with $\mathfrak{sl}_2$ generators inserted:

$$\langle\phi_2|e^{a\mathsf{B}+a'\mathsf{E}+a''\mathsf{P}}|\phi_1\rangle = \left(\frac{-2c^2}{Y(\phi_2)\cdot e^{ab+a'e+a''p}Y(\phi_1)}\right)^\Delta. \tag{D.8}$$

The $\mathsf{B}, \mathsf{E}, \mathsf{P}$ operators on the LHS are the generators acting on the chord Hilbert space, or the double-scaled SYK Hilbert space. The $b, e, p$ operators on the RHS are the explicit three-by-three matrices above, acting on the $Y$ vector. As an application of this formula, we can compute the two point function of matter operators inserted at $\theta = 0$ on the physical circle, with an insertion of the generator $E + B$:

$$\langle\mathcal{O}(0)e^{-a(E+B)}\mathcal{O}(0)\rangle = \langle(1-\nu)\tfrac{\pi}{2}|e^{-a(\mathsf{E}+\sqrt{1-c^2}\mathsf{B})}|-(1-\nu)\tfrac{\pi}{2}\rangle \tag{D.9}$$

$$= \left(\frac{c^2 e^{-ca/2}}{1-(1-c^2)e^{-ca}}\right)^{2\Delta}. \tag{D.10}$$

In the first line, we used (137) to translate $\theta_2 = 0$ and $\theta_1 = 0$ to the fake circle as $\phi_2 = (1-\nu)\tfrac{\pi}{2}$ and $\phi_1 = -(1-\nu)\tfrac{\pi}{2}$. These lead to $Y(\phi_1) = \{1, \sqrt{1-c^2}, -c\}$ and $Y(\phi_2) = \{1, \sqrt{1-c^2}, c\}$ and to get the final expression we just did the matrix exponentiation and dot products in mathematica.

So far we have discussed the representation for states $|\phi\rangle$ defined by inserting a matter operator in the boundary. For the states $|x\rangle$ corresponding to inserting the matter particle at a fixed position $x = \lambda(n_L - n_R)$ among the chords, the representation (132) can also be matched to an operator inserted at the boundary of hyperbolic space, but in a different coordinate system:

$$|x\rangle \sim \sqrt{[n]!}\lim_{\sigma\to-\infty}\left(\frac{e^{|\sigma|}}{2\cosh\eta}\right)^\Delta e^{\eta\mathsf{E}}\mathcal{O}(\sigma,x)|\mathsf{TFD}\rangle, \tag{D.11}$$

$$\langle x| \sim \sqrt{[n]!}\lim_{\sigma\to\infty}\left(\frac{e^{|\sigma|}}{2\cosh\eta}\right)^\Delta \langle\mathsf{TFD}|\mathcal{O}(\sigma,x)e^{\eta\mathsf{E}}. \tag{D.12}$$

The normalization here is chosen to reproduce the inner product (134).

As an aside, let us note that using (D.1) to find the relationship between the $(\rho,\phi)$ and $(\sigma,x)$ coordinate systems, one can confirm (142). This implies that

$$\langle\phi+\delta\phi|x|\phi-\delta\phi\rangle = \langle\phi|e^{-\mathsf{B}\delta\phi}xe^{\mathsf{B}\delta\phi}|\phi\rangle \tag{D.13}$$

$$= x_{\phi-\delta\phi}\langle\phi|\phi\rangle. \tag{D.14}$$

One can check this using to find the Heisenberg equation of motion:

$$\sqrt{1-c^2}[x,\mathsf{B}] = c - \cosh x, \tag{D.15}$$

$$\Rightarrow \left\langle e^{-\mathsf{B}\delta\phi}xe^{\mathsf{B}\delta\phi}\right\rangle = -2\tanh^{-1}\left[\sqrt{\frac{1-c}{1+c}}\tan\left(\tfrac{1}{2}(\phi-\delta\phi)\right)\right]. \tag{D.16}$$

Here the expectation value means the normalized quantity $\langle\phi|e^{-\mathsf{B}\delta\phi}xe^{\mathsf{B}\delta\phi}|\phi\rangle/\langle\phi|\phi\rangle$. With the help of (143), one can show that this agrees with (D.14).

# E  Diagonalizing $E + B$

## E.1  Finite $q$ omputation

Acting on the Hilbert space with a single matter particle, the combination of generators $E + B$ is given by

$$E + B = \frac{1}{(1-q)c}\left(q^{n-n_L} - c^2 + c^2\mathfrak{a}_R^\dagger\alpha_L(1-q^{-n_L})\right). \tag{E.1}$$

Acting on a state

$$|f\rangle = \sum_{n_L=0}^{n} f_{n_L}|n_L, n-n_L\rangle, \tag{E.2}$$

the eigenvector condition is

$$(E+B)|f\rangle = \chi|f\rangle \quad\Longrightarrow\quad \chi f_{n_L} = \frac{1}{(1-q)c}\left((q^{n-n_L}-c^2)f_{n_L} + c^2(1-q^{-n_L-1})f_{n_L+1}\right), \tag{E.3}$$

which can be rewritten as

$$f_{n_L+1} = \frac{(1-q)c\chi - q^{n-n_L} + c^2}{c^2(1-q^{-n_L-1})}f_{n_L}. \tag{E.4}$$

Because $f_{n_L}$ should be supported on $n_L \leq n$, we need the numerator in this expression to vanish for some value $n_L = m \in \{0, 1, 2, 3, \ldots, n\}$. This determines the possible eigenvalues

$$\chi^{(m)} = \frac{q^{n-m} - c^2}{(1-q)c}, \qquad m = 0, 1, 2, 3, \ldots, n. \tag{E.5}$$

In the limit $q \to 1$ holding fixed $c^2 = q^{n+\Delta}$ and $m$, the eigenvalues are approximately

$$\chi^{(m)} \to (m + \Delta)c. \tag{E.6}$$

In the main text we found that in this limit, $B = \sqrt{1-c^2}\tilde{B}$ and $E = \tilde{E}$, where $\tilde{B}$ and $\tilde{E}$ are standard SL(2) generators. This is consistent with (E.6), because $\sqrt{1-c^2}\tilde{B} + \tilde{E}$ is conjugate to $c\tilde{E}$, and the eigenvalues of $\tilde{E}$ in a representation with primary of dimension $\Delta$ are $m + \Delta$.

The corresponding eigenvectors of $E + B$ (normalized so that $f_0^{(m)} = 1$) are

$$|f^{(m)}\rangle = \sum_{0 \leq n_L \leq m}(-1)^{n_L}q^{\frac{n_L(n_L-1)}{2}-n_L(m+\Delta-1)}\binom{m}{n_L}_q|n_L, n-n_L\rangle, \qquad \binom{m}{n_L}_q \equiv \prod_{k=0}^{n_L-1}\frac{1-q^{m-k}}{1-q^{k+1}}. \tag{E.7}$$

The state corresponding to a particle all the way on the left side of the chord Hilbert space is very simple to express in this eigenbasis:

$$|0, n\rangle = |f^{(0)}\rangle. \tag{E.8}$$

The state with a particle all way on the right side is more complicated, but because the matrix of eigenvectors is triangular, one can still solve for the correct linear combination explicitly, and the answer can be written as

$$|n, 0\rangle = \sum_{m=0}^{n}(-1)^m q^{\frac{m(m-1)}{2}+n\Delta}\frac{(q^{1+m};q)_\infty}{(q^{1+n};q)_\infty(q;q)_{n-m}}|f^{(m)}\rangle. \tag{E.9}$$

Now let's compute

$$\langle n, 0|e^{-a(E+B)}|n, 0\rangle = \sum_{m=0}^{n}(-1)^m q^{\frac{m(m-1)}{2}+n\Delta}\frac{(q^{1+m};q)_\infty}{(q^{1+n};q)_\infty(q;q)_{n-m}}\langle n, 0|f^{(m)}\rangle e^{-a\chi^{(m)}}. \tag{E.10}$$

After inserting the expression for $|f^{(m)}\rangle$ in (E.7), we will get terms that involve the inner product $\langle n, 0|n_L, n - n_L\rangle$. This was computed in [17]:

$$\frac{\langle n, 0|n_L, n - n_L\rangle}{\langle n, 0|n, 0\rangle} = q^{\Delta(n-n_L)}. \tag{E.11}$$

Substituting this in gives the expression

$$\frac{\langle n, 0|e^{-a(E+B)}|n, 0\rangle}{\langle n, 0|n, 0\rangle} = \sum_{m=0}^{n} (-1)^m \frac{q^{\frac{m(m-1)}{2}+n\Delta}(q^{1+m}; q)_\infty}{(q^{1+n}; q)_\infty (q; q)_{n-m}}$$

$$\times \sum_{n_L=0}^{m} (-1)^{n_L} \frac{q^{\frac{n_L(n_L-1)}{2}}}{q^{n_L(m-1)+n_L\Delta}} \binom{m}{n_L}_q q^{\Delta(n-n_L)} e^{-a\chi^{(m)}}$$

$$= \sum_{m=0}^{n} (-1)^m \frac{q^{\frac{m(m-1)}{2}+2n\Delta}(q^{1+m}; q)_\infty}{(q^{1+n}; q)_\infty (q; q)_{n-m}} (q^{1-m-2\Delta}; q)_m e^{-a\chi^{(m)}}. \tag{E.12}$$

To get to the last line we used the "$q$-Binomial theorem"

$$\sum_{n_L=0}^{m} q^{\frac{n_L(n_L-1)}{2}}(-q^{-2\Delta-(m-1)})^{n_L} \binom{m}{n_L}_q = \prod_{p=0}^{m-1}(1 - q^{p+1-m-2\Delta}) = (q^{1-m-2\Delta}; q)_m. \tag{E.13}$$

A nice feature of the expression (E.12) is that it is a sum of positive terms, because the $(q^{1-m-2\Delta}, q)_m$ term contains a factor of $(-1)^m$ that cancels the explicit one in (E.12). This is nontrivial because for $a < 0$, the wave function of the evolved state is very large and oscillatory, see (156), and dangerous to try to approximate. The sum of positive terms is convenient because it can be safely approximated in the $q \to 1$ limit, using

$$(q^{1+m}, q)_\infty \to \frac{(1-q)^{-m}(q, q)_\infty}{\Gamma(1+m)}, \tag{E.14}$$

$$(q^{1-m-2\Delta}; q)_m \to \frac{\Gamma(m+2\Delta)}{\Gamma(2\Delta)}(-1)^m(1-q)^m, \tag{E.15}$$

$$(q; q)_{n-m} \to \frac{(q; q)_n}{(1-c^2)^m}, \tag{E.16}$$

$$e^{-a\chi^{(m)}} \to e^{-ac(m+\Delta)}. \tag{E.17}$$

In each case the leading error is a multiplicative correction of order $(1-q)$ times a quadratic polynomial in $m$. Using also $(q, q)_\infty \approx (q; q)_n (q^{1+n}; q)_\infty$, we get

$$\frac{\langle n, 0|e^{-a(E+B)}|n, 0\rangle}{\langle n, 0|n, 0\rangle} \to q^{2\Delta n} \sum_{m=0}^{\infty} \frac{\Gamma(m+2\Delta)}{\Gamma(2\Delta)\Gamma(1+m)}(1-c^2)^m e^{-ac(m+\Delta)} \tag{E.18}$$

$$= \left(\frac{c^2 e^{-ca/2}}{1-(1-c^2)e^{-ca}}\right)^{2\Delta}, \tag{E.19}$$

as quoted in the main text.

## E.2 Semiclassical limit

The wormhole state with the matter chord all the way to the left $|0, n\rangle = |\rangle\text{||}\cdots\text{|}\rangle$ is an eigenvector of $E + B$ with an eigenvalue that approaches $c\Delta$ in the $\lambda \to 0$ limit. We can see this directly from the expression (139). Acting on the two point function, $E + B = (\sin(\pi v/2) + \cos\phi_1)\partial_{\phi_1} - \Delta\sin\phi_1$ so at $\phi = -\frac{1}{2}\pi(1+v)$ the derivative term vanishes

and the last term gives $\Delta \cos(\pi\nu/2) = \Delta c$. Similarly, the state $|n, 0\rangle = |\!|\!| \cdots |\!\langle\rangle$ is an eigenvector of $E - B$. This gives another interpretation of the boundary of the fake region as the fixed point of the symmetry generators $E \pm B$:

$$E + B = \mathsf{E} + \sqrt{1-c^2}\mathsf{B} \quad = \qquad \qquad \tag{E.20}$$

One can also consider the expressions for $E + B$ (132) acting on wave functions $\psi(x)$ for a state $\int dx\, \psi(x)|x\rangle$:

$$(E + B)\psi = \Delta e^x \psi + \partial_x[(c - e^x)\psi].\tag{E.21}$$

Setting $x_* = \log(c) = -\ell/2$, one finds that the wave function $\psi = \delta(x - x_*)$ is an eigenvector with eigenvalue $c\Delta$ as expected. It is curious that a particle localized at some position $x = x_*$ is an eigenstate of the $E + B$ generator given that there is no bulk fixed point of the symmetry $E + B$ isometry. More generally, we can diagonalize the $\lambda \to 0$ expression (132) for $E + B$. We arrive at

$$(\Delta - 1)e^x \psi(x) + (c - e^x)\psi'(x) = \chi\psi(x).\tag{E.22}$$

Writing $\chi = c(m + \Delta)$ we seem to find smooth eigenfunctions $(c - e^x)^{-m-1} e^{x(\Delta+m)}$. These are not normalizable at $x = x_*$, but the closely related distributions

$$f^{(m)}(x) = e^{x(\Delta+m)}(e^{-x}\partial_x)^m \delta(c - e^x),\tag{E.23}$$

are normalizable with respect to the chord inner product, and one can check that these are also eigenvectors in the distributional sense, with eigenvalue $\chi = c(\Delta + m)$. We can then compare these to (E.7). For example, the $m = 1$ and $m = 2$ cases are

$$f^{(1)}(x) \approx \Delta\delta(x - x_\star) - \delta'(x - x_\star),\tag{E.24}$$

$$f^{(2)}(x) \approx \Delta(\Delta - 1)\delta(x - x_\star) - (1 + 2\Delta)\delta'(x - x_\star) + \delta''(x - x_\star),\tag{E.25}$$

and the states $\int dx\, f(x)|x\rangle$ agree with the answers from (E.7) for small $\lambda$:

$$|f^{(1)}\rangle = |0, n\rangle - q^{-\Delta}|1, n - 1\rangle\tag{E.26}$$

$$\approx \lambda\Delta(|x_\star + \lambda\rangle) + (|x_\star\rangle - |x_\star + \lambda\rangle),\tag{E.27}$$

$$|f^{(2)}\rangle = |0, n\rangle - q^{(\Delta+1)}|1, n - 1\rangle + \frac{q^{-1-2\Delta}}{1 + q}|2, n - 2\rangle\tag{E.28}$$

$$\approx |x_\star\rangle - (1 - \lambda(\Delta + 1))|x_\star + \lambda\rangle + \tfrac{1}{2}(1 - (1 + 2\Delta)\lambda)|x_\star + 2\lambda\rangle.\tag{E.29}$$

Note that if we use $x$ to label the 1-particle states, we have $|m, n - m\rangle = |x_* + \lambda m\rangle$.

# F    Expanding two-particle states in double-trace primaries

In this appendix we will discuss the decomposition of a two particle VW state into a sum of "double-trace" primary states:

$$|n_L, n_1, n_R\rangle = \sum_{k+m_L+m_R=n_1} \psi_{k,m_L,m_R}|[\mathsf{VW}]_k; n_L + m_L, n_R + m_R\rangle.\tag{F.1}$$

To determine the coefficients, one can start by acting with $\mathfrak{a}_L$ on both sides of this equation in the special case $n_L = n_R = 0$. One finds

$$r_V \frac{1-q^{n_1}}{1-q}|0, n_1 - 1, 0\rangle \tag{F.2}$$
$$= \sum_{k+m_L+m_R=n_1} \psi_{km_L m_R} \left( \frac{1-q^{m_L}}{1-q}|[\mathsf{VW}]_k; m_L - 1, m_R\rangle + r_V r_W q^{m_L+k} \frac{1-q^{m_R}}{1-q}|[\mathsf{VW}]_k; m_L, m_R - 1\rangle \right).$$

The LHS can be written in terms of the double-trace states using (F.1). The descendants in these double-trace families are not orthogonal but they are linearly independent, so we get a recursion relation for $\psi$ by collecting the coefficients that multiply a given state. Explicitly, from this equation and a similar one obtained by acting with $\mathfrak{a}_R$, we get the recursions

$$r_V \frac{1-q^{k+m_L+m_R+1}}{1-q}\psi_{k,m_L,m_R} = \frac{1-q^{m_L+1}}{1-q}\psi_{k,m_L+1,m_R} + \frac{1-q^{m_R+1}}{1-q} r_V r_W q^{k+m_L}\psi_{k,m_L,m_R+1}, \tag{F.3}$$

$$r_W \frac{1-q^{k+m_L+m_R+1}}{1-q}\psi_{k,m_L,m_R} = \frac{1-q^{m_L+1}}{1-q} r_V r_W q^{k+m_R}\psi_{k,m_L+1,m_R} + \frac{1-q^{m_R+1}}{1-q}\psi_{k,m_L,m_R+1}. \tag{F.4}$$

These two equations can be rearranged to give decoupled recursions

$$\psi_{k,m_L,m_R} = r_V \frac{(1-r_W^2 q^{k+m_L-1})}{1-r_V^2 r_W^2 q^{2k+m_L+m_R-1}} \frac{1-q^{k+m_L+m_R}}{1-q^{m_L}}\psi_{k,m_L-1,m_R}, \tag{F.5}$$

$$\psi_{k,m_L,m_R} = r_W \frac{1-r_V^2 q^{k+m_R-1}}{1-r_V^2 r_W^2 q^{2k+m_L+m_R-1}} \frac{1-q^{k+m_L+m_R}}{1-q^{m_L}}\psi_{k,m_L,m_R-1}, \tag{F.6}$$

with the solution

$$\psi_{k,m_L,m_R} = r_V^{m_L} r_W^{m_R} \frac{(q^{k+1}r_W^2;q)_{m_L}(q^{k+1}r_V^2;q)_{m_R}}{(q^{2k+1}r_W^2 r_V^2;q)_{m_L+m_R}} \frac{(q^{k+1};q)_{m_L+m_R}}{(q;q)_{m_L}(q;q)_{m_R}}\psi_{k,0,0}. \tag{F.7}$$

These equations are linear and homogeneous so they do not determine $\psi_{k,0,0}$. But one can impose that the LHS and RHS of (F.1) should have the same norm in the case $n_L = n_R = 0$. The norm of the LHS is simply $\langle 0, n_1, 0|0, n_1, 0\rangle = [n_1]!$, and the norm of the RHS can be computed using (B.10). Based on the explicit answers for small values of $k$, we guessed the following formula

$$\psi_{k,0,0}^2 = [k]! \frac{(r_V^2;q)_k(r_W^2;q)_k}{(r_V^2 r_W^2 q^{k-1};q)_k}, \tag{F.8}$$

which we then checked up to $k = 25$, giving us confidence that it is exactly correct. The ambiguity in the sign in the square root reflects an ambiguity in the overall sign of the operator $[\mathsf{VW}]_k$, and to be definite we will choose $\psi_{k,0,0}$ to be the positive square root.

Another interesting property that we guessed based on small $k$ is

$$\langle[\mathsf{VW}]_k|[\mathsf{WV}]_k\rangle = (-1)^k q^{\frac{k(k-1)}{2}+k(\Delta_V+\Delta_W)+\Delta_V\Delta_W} \langle[\mathsf{VW}]_k|[\mathsf{VW}]_k\rangle. \tag{F.9}$$

Note that on the LHS we have $[\mathsf{VW}]$ and $[\mathsf{WV}]$ and on the RHS we have $[\mathsf{VW}]$ in both terms. We checked this up to $k = 25$ using our formula for the expansion (F.1) in the case $n_L = n_R = 0$, where $\langle \wr n_1 \wr | \wr n_1 \wr \rangle = q^{n_1(\Delta_V+\Delta_W)+\Delta_V\Delta_W}[n_1]!$.

As an application, let's see how to use (F.7) to expand the operator

$$\mathsf{V}(\tfrac{\epsilon}{2})\mathsf{W}(-\tfrac{\epsilon}{2}) = e^{\epsilon H/2}\mathsf{V}e^{-\epsilon H}\mathsf{W}e^{\epsilon H/2}, \tag{F.10}$$

into double-trace primaries in the semiclassical limit. More explicitly, what we mean is that we act with these operators on the infinite temperature state with zero chords and exand the result in terms of double-trace primaries and their descendants:

$$V(\tfrac{\epsilon}{2})W(-\tfrac{\epsilon}{2})|\Omega\rangle = |\wr\wr\rangle + \frac{\epsilon}{\sqrt{\lambda}}\left(\tfrac{1}{2}|\wr\wr\rangle - |\wr\wr\rangle + \tfrac{1}{2}|\wr\wr\wr\rangle\right) + \ldots \tag{F.11}$$

$$= |0,0,0\rangle + \frac{\epsilon}{\sqrt{\lambda}}\left(|1,0,0\rangle - 2|0,1,0\rangle + |0,0,1\rangle\right) + \ldots \tag{F.12}$$

One can then use (F.1) to write the RHS in terms of double-trace primaries and descendants. At higher orders in $\epsilon$ the expansion becomes tricky because one needs to account for the fact that $H$ insertions can contract with each other. However, the leading order (in $\epsilon$) term that multiplies $|[VW]_k; 0, 0\rangle$ is easy to write down: this comes from the term

$$\begin{aligned}
V(\tfrac{\epsilon}{2})W(-\tfrac{\epsilon}{2})|\Omega\rangle &\supset \frac{1}{k!}\left(\frac{-\epsilon}{\sqrt{\lambda}}\right)^k |0,k,0\rangle \\
&\supset \frac{1}{k!}\left(\frac{-\epsilon}{\sqrt{\lambda}}\right)^k \psi_{k,0,0}|[VW]_k; 0, 0\rangle \\
&\approx (-\epsilon)^k \sqrt{\frac{(2\Delta_V)_k(2\Delta_W)_k}{(2\Delta_V + \Delta_W + k - 1)_k k!}}|[VW]_k; 0, 0\rangle.
\end{aligned} \tag{F.13}$$

Here $(x)_k = \Gamma(x+k)/\Gamma(x)$ is the ordinary Pochhammer symbol. In the semiclassical limit we expect an $\mathfrak{sl}_2$-invariant OPE, and this indicates that the OPE coefficients for the expansion of $VW$ in terms of primaries $[VW]_k$ are

$$f^2_{VW[VW]_k} = \frac{(2\Delta_V)_k(2\Delta_W)_k}{(2\Delta_V + \Delta_W + k - 1)_k k!}. \tag{F.14}$$

The other terms in the expansion should be $\mathfrak{sl}_2$ descendants. This is consistent with the leading order results for the $\langle WVVW\rangle$ correlation function. At leading order for small $\lambda$, this simply factorizes into the product of two point functions

$$\frac{\langle W_2 V_4 V_3 W_1\rangle}{\langle W_2 W_1\rangle\langle V_4 V_3\rangle} = 1. \tag{F.15}$$

This "t channel" identity is reproduced by the sum over "s channel" conformal blocks with the coefficients (F.14):

$$1 = \sum_{n=0}^{\infty} f^2_{VW[VW]_n} k_{\Delta_W + \Delta_V + n}(\chi), \tag{F.16}$$

$$k_h(\chi) \equiv (1-\chi)^{2\Delta_W} \chi^{h-\Delta_W-\Delta_V} {}_2F_1(h + \Delta_W - \Delta_V, h + \Delta_W - \Delta_V, 2h, \chi), \tag{F.17}$$

$$\chi \equiv \frac{\sin\frac{\phi_{13}}{2}\sin\frac{\phi_{42}}{2}}{\sin\frac{\phi_{14}}{2}\sin\frac{\phi_{32}}{2}}. \tag{F.18}$$

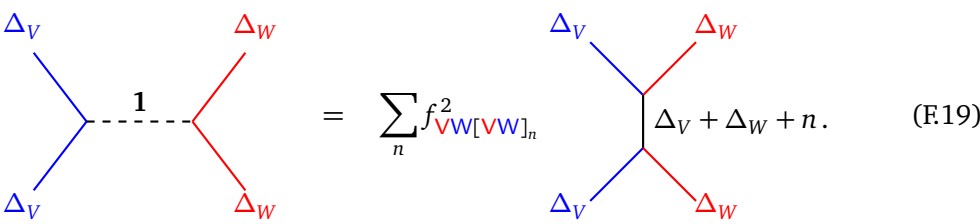

$$\tag{F.19}$$

### F.1 Decomposition of a commutator state

Let's now apply this same idea to decompose a state created by the commutator $[V(\frac{\epsilon}{2}), W(-\frac{\epsilon}{2})]$. One can just re-use (F.13) and subtract off an additional term with $\epsilon \leftrightarrow -\epsilon$ and $V \leftrightarrow W$:

$$[V(\tfrac{\epsilon}{2}), W(-\tfrac{\epsilon}{2})]|\Omega\rangle \supset \frac{\psi_{k,0,0}}{k!}\left(\frac{-\epsilon}{\sqrt{\lambda}}\right)^k |[VW]_k; 0,0\rangle - \frac{\psi_{k,0,0}}{k!}\left(\frac{\epsilon}{\sqrt{\lambda}}\right)^k |[WV]_k; 0,0\rangle \tag{F.20}$$

$$= \frac{\psi_{k,0,0}}{k!}\left(-\frac{\epsilon}{\sqrt{\lambda}}\right)^k \left(|[VW]_k; 0,0\rangle - (-1)^k |[WV]_k; 0,0\rangle\right). \tag{F.21}$$

The state on the RHS is a primary of the chord algebra, but it is not normalized. We will use $|[VW]_k\rangle$ and $|\{V,W\}_k\rangle$ to mean normalized commutator and anticommutator primaries. Using (F.9) one finds

$$|[VW]_k\rangle - (-1)^k |[WV]_k\rangle = \sqrt{2(1 - q^{\frac{k(k-1)}{2} + k(\Delta_W + \Delta_V) + \Delta_W \Delta_V})}\begin{cases} |[VW]_k\rangle, & k \text{ even,} \\ |\{V,W\}_k\rangle, & k \text{ odd.} \end{cases} \tag{F.22}$$

In the semiclassical limit the factor inside the square root is small, proportional to $\lambda$:

$$2\left(1 - q^{\frac{k(k-1)}{2} + k(\Delta_W + \Delta_V) + \Delta_W \Delta_V}\right) \approx \lambda\left(n(n-1) + 2n(\Delta_V + \Delta_W) + 2\Delta_V \Delta_W\right). \tag{F.23}$$

This means that our state $[V,W]|\Omega\rangle$ has small norm in the semiclassical limit, where the operator almost commute. We expect that the double-commutator four point function $\langle[V,W][V,W]\rangle$ should have a conformal block decomposition identical to $\langle WVVW\rangle$, except with an extra factor of (F.23). This is in fact correct because

$$\frac{\langle[W_2, V_4][V_3, W_1]\rangle}{\langle W_2 W_1\rangle\langle V_4 V_3\rangle} = 2\lambda\Delta_V \Delta_W \frac{1 + \chi}{1 - \chi} \tag{F.24}$$

$$= \lambda\sum_{n=0}^{\infty}\left(n(n-1) + 2n(\Delta_V + \Delta_W) + 2\Delta_V \Delta_W\right)f_{VW[VW]_k}^2 k_{\Delta_W + \Delta_V + n}(\chi), \tag{F.25}$$

where the coefficients $f_{VW[VW]_k}$ are given in (F.14).

Notice that the primaries that appear in (F.21) are $[V,W]_k$ for even $k$ and $\{V,W\}_k$ for odd $k$. To explain this, consider the operator

$$\tilde{R} = (-1)^n R, \tag{F.26}$$

where $n$ counts the number of Hamiltonian chords. Then the commutator satisfies

$$\tilde{R}[V(\tfrac{\epsilon}{2}), W(-\tfrac{\epsilon}{2})]\tilde{R} = -1, \tag{F.27}$$

and the primaries that appear are precisely the ones with the same property.

# G  Time-ordered 4-pt function

## G.1  Computation using the 2-sided OPE

We consider the time-ordered correlator in the following configuration:

$$\left\langle W(\tau_L)W(\tau_R)V(\tau_L')V(\tau_R')\right\rangle = \tag{G.1}$$



To compute the $O(\lambda)$ connected contribution to the TOC, we will adopt the following strategy. First we write $WW = e^{-\Delta_W \ell}$ and $VV = e^{-\Delta_V \ell}$. This is a valid substitution as long as there are no other $V$ and $W$ operators around. We then expand $\ell = \langle \ell \rangle_0 + \delta \ell$ where $\langle \ell \rangle_0$ is the classical answer, e.g., the answer obtained by solving Liouville's equation (30). Then the $O(\lambda)$ contribution to the connected correlator will be just $\Delta_V \Delta_W \langle \delta \ell(\tau_+) \delta \ell(\tau_+') \rangle$ where $\tau_+ = \tau_L + \tau_R$ and $\tau_+' = \tau_L' + \tau_R'$.

We can write an operator expression for $\delta \ell$ using (174), specialized to the case with no matter:

$$\delta \ell(\tau_+) = -\lambda \frac{\sqrt{1-c^2}}{c^2}(1 + c\tau \tan c\tau_+)(\delta H) + c^{-1} \tan(c\tau_+)\dot\ell. \tag{G.2}$$

Here $\dot\ell$ is evaluated at the time-symmetric slice: $\dot\ell = \partial_{\tau_+} \ell(\tau_+)|_{\tau_+=0}$. Then the 2-pt function of this operator is given by

$$\begin{aligned}
\langle \delta \ell(\tau_+) \delta \ell(\tau_+') \rangle =& \lambda^2 \frac{1-c^2}{c^4}(1 + c\tau_+ \tan c\tau_+)(1 + c\tau_+' \tan c\tau_+') \langle \delta H^2 \rangle \\
&- \lambda \frac{\sqrt{1-c^2}}{c^3} \Big[ (1 + c\tau_+ \tan c\tau_+) \tan c\tau_+' \langle H\dot\ell \rangle + (1 + c\tau_+' \tan c\tau_+') \tan c\tau_+ \langle \dot\ell H \rangle \Big] \\
&+ c^{-2} \tan c\tau_+ \tan c\tau_+' \langle \dot\ell^2 \rangle.
\end{aligned} \tag{G.3}$$

The $\langle (\delta H)^2 \rangle$ term can be evaluated using (182). The $\langle H\dot\ell \rangle = \langle \dot\ell H \rangle$ terms can be evaluated by writing $\tau_+ = \frac{1}{2}(\tau_{\text{bottom}} - \tau_{\text{top}})$ and representing $H = -\partial/\partial\tau_{\text{bottom}} = -\frac{1}{2}\partial/\partial\tau_+$:

$$\dot\ell(\tau_+) = -2c \tan(c\tau_+), \tag{G.4}$$

$$\langle \dot\ell H \rangle = -\frac{1}{2}(\partial_{\tau_+} \dot\ell)|_{\tau_+=0} = c^2. \tag{G.5}$$

In principle, we still need to evaluate $\langle \dot\ell^2 \rangle$. However, we can avoid a direct computation by demanding that the TOC agree with the OTOC in the coincident limit. More precisely, note that the 4-pt function in the coincident limit must satisfy

$$\langle WVWV \rangle \to q^{\Delta_V \Delta_W} \langle WWVV \rangle. \tag{G.6}$$

This relation is derived from the chord formalism; in the limit where the operators are nearly coincident the only difference between the crossed and uncrossed configuration is a single intersection between red and blue chords:

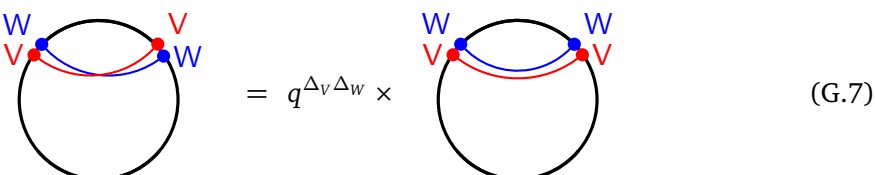

$$= q^{\Delta_V \Delta_W} \times \tag{G.7}$$

In the small $\lambda$ limit, this becomes the requirement

$$\frac{\langle WVWV \rangle_c}{\langle WW \rangle \langle VV \rangle} + \lambda \Delta_V \Delta_W \approx \frac{\langle VVWW \rangle_c}{\langle VV \rangle \langle WW \rangle}. \tag{G.8}$$

The OTOC on the LHS is (183) with $\tau_+ \to \tau_+', \tau_- \to -\beta/2$, and the correlator on the RHS is $\Delta_V \Delta_W \langle \delta \ell(\tau_+)^2 \rangle$. This fixes the final term in (G.3) to be $\langle \dot\ell^2 \rangle / c^2 = 1 + c^{-1}\sqrt{1-c^2} \arccos c$, so

$$\langle \delta \ell(\tau_+) \delta \ell(\tau_+') \rangle = A[1 + (b + c\tau_+) \tan c\tau_+][1 - (b - c\tau_+') \tan c\tau_+'], \tag{G.9}$$

$$A = \lambda \frac{1-c^2}{c^2 + c\sqrt{1-c^2} \arccos(c)} = \lambda \frac{\tan \frac{\pi\nu}{2}}{\frac{\pi\nu}{2} + \cot \frac{\pi\nu}{2}}, \tag{G.10}$$

$$b = \frac{c}{\sqrt{1-c^2}} + \arccos c = \cot\left(\frac{\pi\nu}{2}\right) + \frac{\pi\nu}{2}. \tag{G.11}$$

To write this in a way that transparently matches [8], we can define

$$\phi_{WW} = \frac{2\pi v}{\beta}(\tau_+ + \tfrac{1}{2}\beta), \quad \phi_{VV} = \frac{2\pi v}{\beta}(\tfrac{1}{2}\beta - \tau'_+), \tag{G.12}$$

$$f(\phi) = 1 - \left[\frac{\phi}{2} + \cot\frac{\pi v}{2}\right]\tan\left(\tfrac{1}{2}(\pi v - \phi)\right). \tag{G.13}$$

Here, $\phi$ are some fake angles going the "long way around". Then

$$\frac{\langle WWVV\rangle_c}{\langle WW\rangle\,\langle VV\rangle} = \lambda\Delta_V\Delta_W\frac{\tan\frac{\pi v}{2}}{\frac{\pi v}{2} + \cot\frac{\pi v}{2}}f(\phi_{VV})f(\phi_{WW}). \tag{G.14}$$

### G.2 Chord block argument

As noticed by Streicher [8], the formula (G.14) may be written

$$\langle VVWW\rangle = \langle VV\rangle\langle WW\rangle + \langle(\delta H)VV\rangle\langle(\delta H)WW\rangle/\big\langle(\delta H)^2\big\rangle_\beta. \tag{G.15}$$

Here $\delta H = (H_L + H_R)/2 - \langle H\rangle$, where $\langle H\rangle$ is the average energy at temperature $1/\beta$. We can argue for this as follows. We can insert a resolution of the identity in the 2-sided Hilbert space. If we cut (G.1) on the horizontal $\tau_L = \tau_R$ slice, the state has zero matter particles. So we just need a resolution of the identity in the 0-matter-particles portion of the chord Hilbert space. A convenient basis for this is

$$|\text{TFD}\rangle, H|\text{TFD}\rangle, H^2|\text{TFD}\rangle, \dots \tag{G.16}$$

Then performing the Gram-Schmidt procedure,

$$|\text{TFD}\rangle, \quad \delta H\frac{|\text{TFD}\rangle}{\sqrt{\langle\delta H^2\rangle_\beta}}, \quad \frac{H^2 - \langle H^2\rangle}{\sqrt{(H^2 - \langle H^2\rangle)_\beta^2}}|\text{TFD}\rangle, \dots \tag{G.17}$$

Now so far this argument is at general $\lambda$. However as $\lambda \to 0$, we expect that the states (G.16) to correspond to the power series expansion in $\lambda$. For example, the $O(\lambda^2)$ terms gives

$$\frac{1}{\langle(H^2 - \langle H^2\rangle)^2\rangle_\beta}\langle VV|\big(H^2 - \langle H^2\rangle\big)|\text{TFD}\rangle\langle\text{TFD}|\big(H^2 - \langle H^2\rangle\big)|WW\rangle$$

$$= \frac{\big(\langle VVH^2\rangle - \langle VV\rangle\langle H^2\rangle\big)\big(\langle H^2WW\rangle - \langle H^2\rangle\langle WW\rangle\big)}{\langle H^4\rangle_\beta - \langle H^2\rangle_\beta\langle H^2\rangle_\beta}. \tag{G.18}$$

In (G.18), the denominator is of order $1/\lambda^2$, whereas both factors in the numerator are $O(1)$. Hence at $O(\lambda)$ we only need to insert the first two terms in the Gram-Schmidt basis (G.17) into the LHS of (G.15), which gives the desired result.

## H The collective field approach

In this appendix, we describe how some of the results using the chord algebra can be derived from the collective-field description of double-scaled SYK. This description arises by taking the double-scaled limit of the standard $G, \Sigma$ action of SYK, integrating out the $\Sigma$ field (which appears quadratically in this limit), and changing variables $G(\tau_1, \tau_2) = \frac{1}{2}\text{sgn}(\tau_1 - \tau_2)e^{g(\tau_1, \tau_2)/q}$. The result is that the a nonstandard Liouville-like action

$$I = -\frac{N}{2}\log(2) + \frac{1}{2\lambda}\int_0^\beta \mathrm{d}\tau_1 \int_0^\beta \mathrm{d}\tau_2\left[\frac{1}{4}\partial_1 g\partial_2 g - \mathcal{J}^2 e^g\right]. \tag{H.1}$$

Unlike the usual $G, \Sigma$ action, this one is local in $\tau_1, \tau_2$. An important detail is that the path integral over $\Sigma$ also imposes the boundary conditions

$$g(\tau, \tau) = 0, \qquad g(\tau_1, \tau_2) = g(\tau_2, \tau_1). \tag{H.2}$$

This means that we can restrict to a $g$ variable defined in the domain

$$0 \leq \tau_1 \leq \tau_2 \leq \beta, \tag{H.3}$$

which can be pictured as the shaded region below:

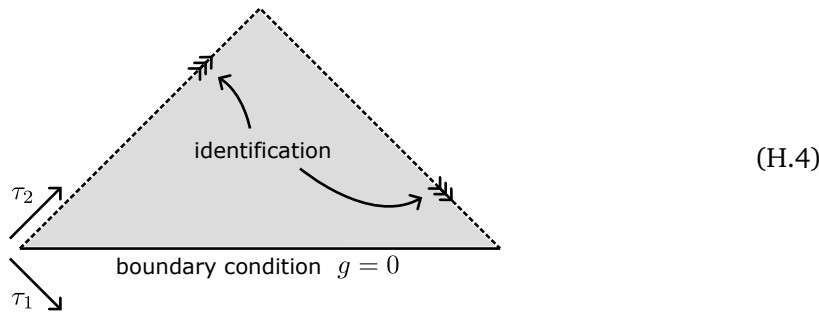

$$\tag{H.4}$$

This space has a boundary at $\tau_1 = \tau_2$ where $g$ is required to vanish. The two sides with dotted lines are identified with each other in an orientation-reversing way, and the end result is a space with the topology of a disk with a crosscap inserted. Restricting to this region, the action is

$$I = -\frac{N}{2}\log(2) + \frac{1}{\lambda}\int_0^\beta d\tau_2 \int_0^{\tau_2} d\tau_1 \left[\frac{1}{4}\partial_1 g \partial_2 g - \mathcal{J}^2 e^g\right]. \tag{H.5}$$

The path integral over $g$ is done with a flat measure, normalized so that the answer is exactly $2^{N/2}$ in the free theory $\mathcal{J}^2 = 0$. We will use brackets $\langle \cdot \rangle_0$ to refer to expectation values in the free theory, normalized so that $\langle 1 \rangle_0 = 1$.

As a first step, we would like to show that this theory reproduces the chord expansion for the partition function. To do so, we study the theory using perturbation theory in $\mathcal{J}^2$, bringing down powers of the $e^g$ operator. To work out this expansion, we need the $\langle g g \rangle$ propagator in the free theory. It has the following convenient expression:

$$\langle g(\tau_1, \tau_2) g(\tau_3, \tau_4) \rangle_0 = \begin{cases} -\lambda, & \text{chord}(\tau_1, \tau_2) \cap \text{chord}(\tau_3, \tau_4) \neq \emptyset, \\ 0, & \text{chord}(\tau_1, \tau_2) \cap \text{chord}(\tau_3, \tau_4) = \emptyset. \end{cases} \tag{H.6}$$

Here by $\text{chord}(\tau_a, \tau_b)$ we mean a straight line connecting the points $\tau_a$ and $\tau_b$ viewed as living on the thermal circle $\tau \sim \tau + \beta$. The propagator is $-\lambda$ if a chord connecting $\tau_1, \tau_2$ intersects a chord connecting $\tau_3, \tau_4$, and it vanishes otherwise. For fixed $\tau_3, \tau_4$, the region where the propagator is nonzero as a function of $\tau_1, \tau_2$ is shown shaded below:

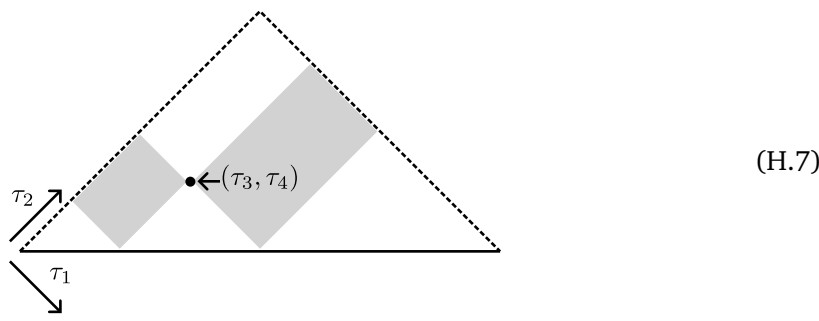

$$\tag{H.7}$$

This propagator respects the boundary conditions (H.2), but we need to check that it satisfies the correct equation for the propagator:

$$\partial_{\tau_1}\partial_{\tau_2}\langle g(\tau_1,\tau_2)g(\tau_3,\tau_4)\rangle_0 = -2\lambda\delta(\tau_{13})\delta(\tau_{24}).\qquad\text{(H.8)}$$

This reduces to checking that we have the correct behavior near $\tau_1 = \tau_3$ and $\tau_2 = \tau_4$. Near this point, we have

$$\langle g(\tau_1,\tau_2)g(\tau_3,\tau_4)\rangle_0 = -\lambda\big[\theta(\tau_{13})\theta(\tau_{24}) + \theta(\tau_{31})\theta(\tau_{42})\big],\qquad\text{(H.9)}$$

and indeed this satisfies (H.8).

Now let's imagine doing perturbation theory in $\mathcal{J}^2$

$$Z = 2^{N/2}\sum_{k=0}^{\infty}\frac{\mathcal{J}^{2k}}{\lambda^k k!}\prod_{i=1}^{k}\Big[\int_0^{\beta}d\tau_1^{(i)}\int_0^{\tau_1^{(i)}}d\tau_2^{(i)}\Big]\langle e^{\sum_{i=1}^{k}g_i}\rangle_0, \qquad g_i \equiv g(\tau_1^{(i)},\tau_2^{(i)}),\qquad\text{(H.10)}$$

$$= 2^{N/2}\sum_{k=0}^{\infty}\frac{\mathcal{J}^{2k}}{\lambda^k k!}\prod_{i=1}^{k}\Big[\int_0^{\beta}d\tau_1^{(i)}\int_0^{\tau_1^{(i)}}d\tau_2^{(i)}\Big]\exp\Big\{\sum_{i<j}\langle g_i g_j\rangle_0 + \frac{1}{2}\langle g_i g_i\rangle_0\Big\}\qquad\text{(H.11)}$$

$$= 2^{N/2}\sum_{k=0}^{\infty}\frac{\mathcal{J}^{2k}}{\lambda^k k!}\prod_{i=1}^{k}\Big[\int_0^{\beta}d\tau_1^{(i)}\int_0^{\tau_1^{(i)}}d\tau_2^{(i)}\Big]q^{\#\text{ chord crossings}}.\qquad\text{(H.12)}$$

This precisely matches the answer from the perspective of the chord diagram expansion of the partition function. In the last step we used $q = e^{-\lambda}$, and we used the prescription $\langle g_i g_i\rangle_0 = 0$ which is needed to give the correct answer for $\langle e^g\rangle_0 = 1$.

We can also compute a correlation function

$$\langle e^{\Delta g}\rangle = \frac{2^{N/2}}{Z}\sum_{k=0}^{\infty}\frac{\mathcal{J}^{2k}}{\lambda^k k!}\prod_{i=1}^{k}\Big[\int_0^{\beta}d\tau_1^{(i)}\int_0^{\tau_1^{(i)}}d\tau_2^{(i)}\Big]\langle e^{\Delta g+\sum_{i=1}^{k}g_i}\rangle_0\qquad\text{(H.13)}$$

$$= \frac{2^{N/2}}{Z}\sum_{k=0}^{\infty}\frac{\mathcal{J}^{2k}}{\lambda^k k!}\prod_{i=1}^{k}\Big[\int_0^{\beta}d\tau_1^{(i)}\int_0^{\tau_1^{(i)}}d\tau_2^{(i)}\Big]\exp\Big\{\Delta\sum_i\langle g g_i\rangle_0 + \sum_{i<j}\langle g_i g_j\rangle_0\Big\}\qquad\text{(H.14)}$$

$$= \frac{2^{N/2}}{Z}\sum_{k=0}^{\infty}\frac{\mathcal{J}^{2k}}{\lambda^k k!}\prod_{i=1}^{k}\Big[\int_0^{\beta}d\tau_1^{(i)}\int_0^{\tau_1^{(i)}}d\tau_2^{(i)}\Big]r^{\#\text{ chords that cross }g\text{ insertion}}q^{\#\text{ chord crossings}},$$

$$\text{(H.15)}$$

where $r = q^{\Delta}$. This again matches the chord rules, and the logic is similar for an arbitrariy insertions of $e^{\Delta g}$ operators. So we see that the chord diagrams are exactly equivalent to doing the $\mathcal{J}^2$ perturbation theory of the Liouville theory. From this point forward we will set $\mathcal{J}^2 = 1$, as in the main text of the paper.

A nice feature of the Liouville perspective is that because the whole action is multiplied by $1/\lambda$, it is clear that small $\lambda$ is a semiclassical limit, with small fluctuations of the $g$ field. One can derive the leading thermodynamics by solving for the saddle point $g_*$ of this action (see [48, 49] for loop computations). The equation of motion (after setting $\mathcal{J}^2 = 1$) is

$$\partial_1\partial_2 g_* = -2e^{g_*}.\qquad\text{(H.16)}$$

The solution corresponding to the thermal state is

$$g_*(\tau_1,\tau_2) = 2\log\frac{\cos\frac{\pi v}{2}}{\cos[\frac{\pi v}{2}(1-\frac{2\tau_{21}}{\beta})]},\qquad 0 < \tau_{21} < \beta,\qquad\text{(H.17)}$$

where in order to solve the equation of motion, $\nu$ must satisfy

$$\frac{\pi\nu}{\beta} = \cos\frac{\pi\nu}{2}. \tag{H.18}$$

By plugging this solution into the action, one can derive the classical energy and entropy, which are conveniently expressed in terms of $\nu$:

$$\langle H\rangle = -\frac{2}{\lambda}\sin\frac{\pi\nu}{2}, \qquad S = \frac{N}{2}\log(2) - \frac{\pi^2\nu^2}{2\lambda}. \tag{H.19}$$

For small but nonzero $\lambda$, one can expand

$$g = g_* + h, \tag{H.20}$$

and study the small fluctuations of $h$. The action for this field is

$$I = I_* + \frac{1}{\lambda}\int_0^\beta d\tau_2 \int_0^{\tau_2} d\tau_1 \left[\frac{1}{4}\partial_1 h \partial_2 h - e^{g_*}\left(\frac{1}{2!}h^2 + \frac{1}{3!}h^3 + \dots\right)\right]. \tag{H.21}$$

The propagator for the $h$ field has an expansion in powers of $\lambda$. The leading term, proportional to $\lambda$, was computed in [8,9]. In the crossed region (where the chords intersect), the result is

$$\langle h(\tau_1,\tau_2)h(\tau_3,\tau_4)\rangle_\lambda = \lambda\left\{-\frac{\tan\frac{\pi\nu}{2}}{2}Y\tan\frac{X}{2}\tan\frac{X'}{2} - \frac{1}{\cos\frac{\pi\nu}{2}}\frac{\cos\frac{Y}{2}}{\cos\frac{X}{2}\cos\frac{X'}{2}}\right.$$
$$\left. + \frac{\tan^2\frac{\pi\nu}{2}}{2\pi\nu\tan\frac{\pi\nu}{2}+4}(2+X\tan\frac{X}{2})(2+X'\tan\frac{X'}{2})\right\}. \tag{H.22}$$

In the uncrossed region (where the chords do not intersect) it is

$$\langle h(\tau_1,\tau_2)h(\tau_3,\tau_4)\rangle_\lambda = \frac{\lambda}{2(\pi\nu\tan\frac{\pi\nu}{2}+2)}\left\{\left(2+X\tan\frac{X}{2}\right)\tan\frac{\pi\nu}{2} - (2+\pi\nu\tan\frac{\pi\nu}{2})\tan\frac{X}{2}\right\}$$
$$\times\left\{\left(2+X'\tan\frac{X'}{2}\right)\tan\frac{\pi\nu}{2} - (2+\pi\nu\tan\frac{\pi\nu}{2})\tan\frac{X'}{2}\right\}. \tag{H.23}$$

The $X,X',Y$ parameters are simple functions of the locations of the points on the thermal circle. For configurations $0 < \tau_1 < \tau_2 < \tau_3 < \tau_4 < \beta$ or $0 < \tau_1 < \tau_3 < \tau_2 < \tau_4 < \beta$, they are given by

$$Y = \pi\nu(1 - \tfrac{2}{\beta}\tau_{3+4-1-2}), \qquad X = \pi\nu(1-\tfrac{2}{\beta}\tau_{21}), \qquad X' = \pi\nu(1-\tfrac{2}{\beta}\tau_{43}). \tag{H.24}$$

All other configurations can be obtained from these two by relabeling points. The free theory $\mathcal{J}^2 = 0$ corresponds to the limit $\nu \to 0$, and one can easily check that these expressions reproduce (H.6) in that limit.

One can try to reproduce the chord algebra from the perspective of the $\lambda$ expansion of the Liouville theory. Suppose that $\tau_1, \tau_2$ are two points on the thermal circle, where we will regard point 1 as the left boundary and point 2 as the right boundary. From the perspective of the chord theory, there is a Hilbert space associated to these two points, and there is an algebra of operators generated by the two-sided size (length) operator $\bar{n}$ and the Hamiltonians $H_L, H_R$ acting on the two points.

This Hilbert space does not seem very natural from the perspective of Liouville theory, but one can nevertheless try to reproduce the algebra of the operators. To do so we need to

translate the size operator and the Hamiltonians into functions of the Liouville field. The size or length operator becomes $g$ itself:

$$\bar{n} \to -\frac{1}{\lambda} g(\tau_1, \tau_2), \tag{H.25}$$

and the two Hamiltonians are translated to

$$H_L \to \frac{1}{\lambda} g^{(1,0)}(\tau_1, \tau_1), \qquad H_R \to \frac{1}{\lambda} g^{(0,1)}(\tau_2, \tau_2). \tag{H.26}$$

Here the superscript on $g^{(a,b)}$ indicates the number of derivatives that we take with respect to its first and second arguments. In taking these derivatives we should stay in the region where the first argument of $g$ is less than the second argument. So, more explicitly, $H_L = \frac{1}{\lambda} \lim_{\epsilon \to 0_+} \partial_\epsilon g(\tau_1 - \epsilon, \tau_1)$ and $H_R = \frac{1}{\lambda} \lim_{\epsilon \to 0_+} \partial_\epsilon g(\tau_2, \tau_2 + \epsilon)$. The above formulas for $H_L, H_R$ are based on the SYK relation $pH = \sum_i [H, \psi^i] \psi^i$ (see 5.76 of [25]). Note that $H_L$ is independent of $\tau_2$ and $H_R$ is independent of $\tau_1$; unlike $\bar{n}$ these are actually one-sided operators.

Given Liouville representations of two chord operators $A \to \mathsf{A}(\tau_1, \tau_2)$ and $B \to \mathsf{B}(\tau_1, \tau_2)$, we can represent their commutator as a Liouville insertion

$$[A, B] \to \mathsf{A}(\tau_1, \tau_2)(\mathsf{B}(\tau_{1+}, \tau_{2-}) - \mathsf{B}(\tau_{1-}, \tau_{2+})), \tag{H.27}$$

where the notation $\tau_{1\pm}$ means $\tau_1 \pm \epsilon$ in the limit $\epsilon \to 0_+$. So in particular, an insertion of $\mathsf{B}(\tau_{1+}, \tau_{2-})$ is slightly earlier on both the L and R boundaries than an insertion of $\mathsf{A}(\tau_1, \tau_2)$.

Let's work out an example by choosing $A = H_R$ and $B = \bar{n}$. Then

$$[H_R, \bar{n}] \to -\frac{1}{\lambda^2} g^{(0,1)}(\tau_2, \tau_2)(g(\tau_{1+}, \tau_{2-}) - g(\tau_{1-}, \tau_{2+})). \tag{H.28}$$

If we have represented the Hamiltonian correctly, then it should act in Liouville variables as $\partial_{\tau_2}$, so the RHS of this expression should actually be equal to $-\lambda^{-1} \partial_{\tau_2} g(\tau_1, \tau_2)$. More precisely, the two expressions should give the same answers when inserted into the Liouville path integral (along with other possible insertions). We will now verify this using the semiclassical small $\lambda$ expansion of the Liouville theory.

At leading order $\lambda^{-2}$, we can evaluate the RHS of (H.28) by plugging in the saddle point value $g = g_*$. Then the two terms in parentheses cancel. There is a similar cancellation if we replace one of the factors of $g$ by the saddle point value $g_*$ and the other by the small fluctuation $h$. But if we replace *both* factors of $g$ by the fluctuation $h$, we can get something nonzero. The biggest nonzero term is at order $\lambda^{-1}$ and it arises from contracting the two factors of $h$ against each other using the propagator worked out above. The two terms correspond to two different regions of the uncrossed correlator. For both terms we have $X = \pi v(1 - \frac{2}{\beta}\epsilon)$, where $\epsilon$ is the small separation between the two arguments in $g^{(0,1)}(\tau_2, \tau_2)$ allows us to take the derivative. For the first term, we have $X' = \pi v(1 - \frac{2}{\beta}\tau_{21})$, and for the second term we have $X' = -\pi v(1 - \frac{2}{\beta}\tau_{21})$. Taking the difference between the two and then taking the derivative with respect to $\epsilon$ and setting $\epsilon = 0$, we get

$$[H_R, \bar{n}] \supset \frac{1}{\lambda} \frac{2\pi v}{\beta} \tan\left[\frac{\pi v}{2}(1 - \frac{2\tau_{21}}{\beta})\right] \tag{H.29}$$

$$= -\frac{1}{\lambda} \partial_{\tau_2} g_*(\tau_1, \tau_2). \tag{H.30}$$

This looks promising but it only gives $g_*$ on the RHS, not $g$. To make up the difference we need an additional term $-\lambda^{-1} \partial_{\tau_2} h(\tau_1, \tau_2)$. To find this we need to go to the next order, allowing

the insertions of $h$ to contract with a cubic interaction term from the expansion of the action (H.21). This gives

$$[H_R, \bar{n}] \supset -\frac{1}{\lambda^2} h^{(0,1)}(\tau_2, \tau_2)(h(\tau_{1+}, \tau_{2-}) - h(\tau_{1-}, \tau_{2+})) \frac{1}{\lambda} \int_0^\beta d\tau_2' \int_0^{\tau_2'} d\tau_1' e^{g_* + h} \frac{1}{3!} h^3. \tag{H.31}$$

Naively this expression will give zero, due to a cancellation between the two terms in parentheses. The fact that it is nonzero is due to the singularities (step function discontinuities) of the $\langle hh \rangle$ propagator along the "lightcones" that form the boundary of the shaded region of (H.7). An important fact is that the discontinuities of the propagator across these lightcones are actually equal to their value in the free theory $\mathcal{J}^2 = 0$. The reason for this is that the propagator (in either $\mathcal{J}^2$ or $\lambda$ perturbation theory) is bounded, and when we connect an external propagator to an interaction vertex, the integral over the location of the interaction vertex will smooth out the discontinuity in the propagator, so perturbative corrections to the propagator will be continuous across the lightcones. This means that the only singular term in the $\langle h^{(0,1)} h \rangle$ propagator comes from the discontinuity of (H.6) across the lightcone, and therefore

$$h^{(0,1)}(\tau_2, \tau_2) h(\tau_1', \tau_2') = -\lambda \delta(\tau_2' - \tau_2) - \lambda \delta(\tau_1' - \tau_2) + \text{nonsingular}. \tag{H.32}$$

One can think of this as the Liouville theory "discovering" that it is describing correlators of fermions in quantum mechanics, where this discontinuity represents the short-distance discontinuity in the fermion propagator, which is determined by the algebra of Majorana fermions and is independent of any interactions.

By similar logic, we find that

$$(h(\tau_{1+}, \tau_{2-}) - h(\tau_{1-}, \tau_{2+})) h(\tau_1', \tau_2) = -\lambda \, \text{sgn}(\tau_1' - \tau_1), \tag{H.33}$$
$$(h(\tau_{1+}, \tau_{2-}) - h(\tau_{1-}, \tau_{2+})) h(\tau_2, \tau_2') = \lambda. \tag{H.34}$$

Using these formulas, (H.31) becomes

$$[H_R, \bar{n}] \supset -\frac{1}{\lambda} \left( \int_{\tau_1}^{\tau_2} d\tau_1' h(\tau_1', \tau_2) e^{g_*} - \int_0^{\tau_1} d\tau_1' h(\tau_1', \tau_2) e^{g_*} + \int_{\tau_2}^\beta d\tau_2' h(\tau_2, \tau_2') e^{g_*} \right) \tag{H.35}$$

$$= \frac{1}{2\lambda} \left( \int_{\tau_1}^{\tau_2} d\tau_1' h^{(1,1)}(\tau_1', \tau_2) - \int_0^{\tau_1} d\tau_1' h^{(1,1)}(\tau_1', \tau_2) + \int_{\tau_2}^\beta d\tau_2' h^{(1,1)}(\tau_2, \tau_2') \right) \tag{H.36}$$

$$= \frac{1}{2\lambda} \left( h^{(0,1)}(\tau_2, \tau_2) - h^{(0,1)}(\tau_1, \tau_2) - h^{(0,1)}(\tau_1, \tau_2) \right.$$
$$\left. + h^{(0,1)}(0, \tau_2) + h^{(1,0)}(\tau_2, \beta) - h^{(1,0)}(\tau_2, \tau_2) \right)$$

$$= -\frac{1}{\lambda} h^{(0,1)}(\tau_1, \tau_2). \tag{H.37}$$

In the second line we used the linearized equation of motion $\partial_{\tau_1} \partial_{\tau_2} h = -2 e^{g_*} h$. Adding this together with (H.30), we find that (H.28) is indeed equivalent to

$$[H_R, \bar{n}] \to -\frac{1}{\lambda} \partial_{\tau_2} g(\tau_1, \tau_2). \tag{H.38}$$

Because we used the linearized equation of motion for $h$ and only expanded down once using the cubic $h^3$ vertex, this computation appears to only be approximate. However, from the SYK perspective (H.38) should be exact, and with the benefit of hindsight we can see why: if the two factors of $h$ in (H.31) are contracted with different interaction vertices, the result will be zero. The effect of including higher interactions $h^k / k!$ combines with nonlinear terms in the equation of motion for $h$ so that one still ends up with (H.36).

# I  Review of the quantum algebra $U_{\sqrt{q}}(\mathfrak{sl}_2)$

Here we collect some facts about the quantum algebra $U_{\sqrt{q}}(\mathfrak{sl}_2)$. The algebra is given by (C.4)

$$KK^{-1} = K^{-1}K = 1, \quad KE_\pm K^{-1} = q^{\pm 1}E_\pm,$$

$$[E_+, E_-] = \frac{K - K^{-1}}{q^{1/2} - q^{-1/2}}. \tag{I.1}$$

In the math literature, two of the generators are often denoted $E = E_+$ and $F = E_-$. In the limit $q \to 1$, if we define $K = 1 - \frac{1-q}{2}H$ then the algebra contracts to the standard $\mathfrak{sl}_2$ algebra $[H, E_\pm] = \pm 2E_\pm$ and $[E_+, E_-] = H$.

The $U_{\sqrt{q}}(\mathfrak{sl}_2)$ algebra has finite dimensional irreducible representations. These are analogous to the spin $j$ representations of $\mathfrak{sl}_2$. These representations are given explicitly via

$$E_+ = \begin{pmatrix} 0 & \alpha_1 & & & \\ & 0 & \alpha_2 & & \\ & & 0 & \ddots & \\ & & & 0 & \alpha_{2j} \\ & & & & 0 \end{pmatrix}, \qquad E_- = \begin{pmatrix} 0 & & & & \\ 1 & 0 & & & \\ & 1 & 0 & & \\ & & \ddots & 0 & \\ & & & 1 & 0 \end{pmatrix}, \tag{I.2}$$

$$K^\pm = \mathrm{diag}(q^{\pm j}, q^{\pm(j-1)}, \cdots, q^{\mp j}).$$

There is a Casimir operator defined in (C.9), and for these representations it has the value

$$\Omega = q^{-j} + q^j, \quad \dim R = 2j + 1. \tag{I.3}$$

For $|q| \neq 1$ these are the only finite dimensional, irreducible representations.

The algebra (I.1) can be promoted to a Hopf algebra by including a coproduct $D$

$$D(E_+) = E_+ \otimes K + 1 \otimes E_+, \quad D(E_-) = E_- \otimes 1 + K^{-1} \otimes E_-, \quad D(K) = K \otimes K. \tag{I.4}$$

One can check that $D$ satisfies the relations (I.1), e.g., it is an algebra homomorphism. There is also an "antipode" $S$ which is the analog of the group inverse:

$$S(K) = K^{-1}, \quad S(E_+) = -E_+K^{-1}, \quad S(E_-) = -KE_-. \tag{I.5}$$

$S$ is an algebra anti-automorphism.

As an aside, let us remark on that in this paper, we only need the quantum algebra $U_{\sqrt{q}}(\mathfrak{sl}_2)$. This should not be confused with the quantum group sometimes denoted $SL_q(2)$, see [46]. For a discussion of $SL_q(2)$ in the context of double-scaled SYK, see [33] and [32].

In addition to the finite dimensional irreps, there are also irreps that are analogous to the discrete series and the continuous series of the ordinary algebra $\mathfrak{sl}_2$. For fixed and very large $\bar{n}$, the 1-particle states transform as the discrete series irrep of $U_{\sqrt{q}}(\mathfrak{sl}_2)$, see Appendix (A.2). The continuous series irrep also makes a brief appearance in the 6j symbol of $U_{\sqrt{q}}(\mathfrak{sl}_2)$.

# J  Fake effects in low temperature SYK at general $p$

The order $1/N$ term in the four point function of the SYK model was studied at low temperature but general $p$ in [7]. The leading term at low temperature, proportional to $\beta\mathcal{J}$, agrees with JT gravity. The first subleading term, at order $(\beta\mathcal{J})^0$, has a piece that has the same form as in the large $p$ limit (but with a $p$-dependent coefficient) and also a piece that comes from a sum

over operators that depend on $p$ – this part vanishes in the large $p$ limit. The total shift in the Lyapunov exponent away from the low temperature value $\lambda_L = 2\pi/\beta$ is

$$\frac{\delta\lambda_L^{\text{total}}}{\lambda_L} = -\frac{1}{k_R'(-1)}\frac{\alpha_K}{\beta\mathcal{J}}, \tag{J.1}$$

and the contribution from the "large $p$" piece is

$$\frac{\delta\lambda_L^{\text{fake}}}{\lambda_L} = -\frac{1}{|k_c'(2)|}\frac{\alpha_K}{\beta\mathcal{J}}. \tag{J.2}$$

The ratio is

$$\frac{\delta\lambda_L^{\text{fake}}}{\delta\lambda_L^{\text{total}}} = \frac{k_R'(-1)}{|k_c'(2)|} = \frac{p((p-6)p+6) - 2\pi(p-2)(p-1)\cot\left(\frac{2\pi}{p}\right)}{p((p-6)p+6) - 2\pi(p-2)(p-1)\csc\left(\frac{2\pi}{p}\right)}. \tag{J.3}$$

For $p = 4$ this is approximately 0.175. Note that the fake contribution to the correction to the Lyapunnov exponent is also consistent with the leading correction to the conformal form of the two point function, which involves the same coefficient [7]

$$\left(\frac{\langle\psi(\tau)\psi(0)\rangle}{\langle\psi(\tau)\psi(0)\rangle_{\text{conformal}}}\right)^p = 1 - \frac{\alpha_K}{|k_c'(2)|\beta\mathcal{J}}\left(2 + \frac{\pi-|\theta|}{\tan\frac{|\theta|}{2}}\right) + \dots, \tag{J.4}$$

where $\theta = 2\pi\tau/\beta$.

The fact that $\delta\lambda_L^{\text{fake}}$ is only part of the correction to the Lyapunov exponent at finite $p$ means that the model does not saturate the strengthened chaos bound of [18], see [50].

Note, however, that while (J.3) is relatively small, the ratio $\lambda_{\text{fake}}/\lambda_{\text{total}}$ is within $\sim 1\%$ of unity [50] over all temperatures (see their Figure 9b). By this measure, fake effects at $\beta\mathcal{J} \sim 1$ dominate over "stringy" effects even at $p = 4$.

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
