# Peer review of "A symmetry algebra in double-scaled SYK"

_SciPost Physics, doi:SciPost Phys. 15, 234 (2023)_

## Round 1 · Referee Report · Anonymous (Referee 1) · 2023-9-19

Strengths

1- The paper is concerned with exploring important questions related to the bulk dual of the SYK model at finite temperature, and derives a concrete result about the sub-maximal chaos exponent.

2- It generalizes the near horizon symmetries of JT gravity.

Weaknesses

None

Report

This paper by Lin and Stanford gives a bulk derivation of the sub-maximal Lyapunov exponent 2πv/ β, at finite temperature, using the “chord Hilbert space” constructed in the scaling limit N → ∞, p → ∞ fixed λ ≡ 2p2/N, of the SYK model. The answer agrees with the known result calculated some time ago by Maldacena and Stanford in the limit of large p, i.e. λ → 0.

On way to the above result they discovered a symmetry sub-algebra of the chord algebra that reduces to the SL2 algebra of near horizon symmetries, in the limit λ → 0, at finite temperature. This result would imply a maximal Lyapunov exponent! The puzzle is resolved by demonstrating that the bulk dual of the large p SYK model is defined on an extended “fake” disk, on which the SL2 has a natural action, leading to a sub-maximal Lyapunov exponent.

The chord (bi)-algebra at finite λ and its representation theory is discussed in detail. They find that the symmetry sub-algebra has finite dimensional unitary representations related to worm holes with an integral chord number.

The paper is very well written and discusses the chord Hilbert space and its operator algebra in detail and raises a host of interesting questions related to a discrete generalization of the geometric JT gravity and its symmetry algebra and generalization to higher dimensions.

Requested changes

None

---

## Round 1 · Referee Report · Daniel Harlow (Referee 2) · 2023-11-17

Report

This very nice paper explains how to extend the gravitational algebra of JT gravity plus matter to an exact statement about double-scaled SYK at finite q. The technical explanations are all clear and to the point, and I think this paper should be published as is. I have only one question for the authors, mostly for my own education: in sections 4.2-4.4 they introduce a "fake disk" picture for understanding their algebra in the limit that $\lambda\to 0$. They suggest that the fake regions of the disk can be interpreted as a failure of "lattice-ish" modes to decouple, similar to fermion doubling. My question is the following: does this mean that there is some mistake in the usual analysis of the SYK model at the disk level at finite $p$? Are there some states that were missed in the usual analysis? Or is this just a re-interpretation of things which have some other more convoluted explanation in the usual formalism?
  • validity: -
  • significance: -
  • originality: -
  • clarity: -
  • formatting: -
  • grammar: -

Author:  Henry Lin  on 2023-12-06  [id 4173]

(in reply to Report 2 by Daniel Harlow on 2023-11-17)
Category:
answer to question

Hi Daniel, in response to your question: no, we did not find any mistakes in the canonical texts on finite p SYK. We just give a "geometric" interpretation of some pieces of the 4-pt function of finite p SYK. Since we have a bulk Hilbert space formalism, we can say more clearly what (a subset of) the states that are produced when we collide 2 particles, etc. In the previous path integral approaches to the 4-pt function, one could try to interpret some terms as coming from "states" in some intermediate channel but it wasn't totally clear how to describe these states as elements in a Hilbert space.

---

## Editorial Decision

published